# Scalable Valuation of Human Feedback through Provably Robust Model Alignment

**Masahiro Fujisawa**[*,†,1,2,4], **Masaki Adachi**[*,2,3], **Michael A. Osborne**[3]
[1] The University of Osaka, [2] Lattice Lab, Toyota Motor Corporation,
[3] Machine Learning Research Group, University of Oxford, [4] RIKEN AIP
[*] Equal Contribution   [†] Corresponding Author
fujisawa@ist.osaka-u.ac.jp, {masaki, mosb}@robots.ox.ac.uk

## Abstract

Despite the importance of aligning language models with human preferences, crowd-sourced human feedback is often noisy—for example, preferring less desirable responses—posing a fundamental challenge to alignment. A truly robust alignment objective should yield identical model parameters even under severe label noise, a property known as *redescending*. We prove that no existing alignment methods satisfy this property. To address this, we propose Hölder-DPO, the first principled alignment loss with a provable redescending property, enabling estimation of the clean data distribution from noisy feedback. The aligned model estimates the likelihood of clean data, providing a theoretically grounded metric for dataset valuation that identifies the location and fraction of mislabels. This metric is gradient-free, enabling scalable and automated human feedback valuation without costly manual verification or clean validation dataset. Hölder-DPO achieves state-of-the-art robust alignment performance while accurately detecting mislabels in controlled datasets. Finally, applied to Anthropic HH-RLHF dataset, it reveals substantial noise levels and removing these mislabels significantly improves alignment performance across methods. The code is available[1].

## 1 Introduction

Aligning large language models (LLMs; [1, 95, 92]) with human preferences is essential for value alignment and mitigating safety risks [65, 9]. Direct Preference Optimization (DPO; [84]) has emerged as a key method, fine-tuning LLMs using pairwise comparisons—e.g., between preferred and rejected responses [25, 18]. This framework supports alignment with a broad range of rankable values, such as helpfulness [8, 51], and summarization quality [90, 97]. Human feedback is typically gathered via crowdsourcing (e.g., ChatGPT [1]), but is often noisy—e.g., preferring undesirable responses—which can significantly deteriorate model performance [105, 24, 29]. In fact, the Anthropic HH-RLHF dataset [34] reportedly contain over 25% inconsistent feedback [88, 20, 105]. Plus, recent work [107, 42, 15] shows that models trained on smaller, high-quality datasets can outperform those trained on larger, noisier ones. However, maintaining data quality often requires costly manual review (e.g., inter-annotator agreement), highlighting the need for scalable, automated assessment methods.

To mitigate the impact of noisy human feedback, researchers have proposed robustifed DPO variants. Among various heuristics [76, 6, 100], two methods claim provable guarantees: Provably Robust DPO (R-DPO) [24] and Distributionally Robust DPO (Dr. DPO) [105]. However, their notion of robustness is limited to bounding the parameter estimation error—bounds that degrade as the fraction of mislabelled data $\epsilon$ increases. In contrast, truly robust methods should estimate the clean data distribution precisely under heavy contamination. This stronger form of robustness corresponds to

---

[1] https://github.com/ma921/HolderDPO

39th Conference on Neural Information Processing Systems (NeurIPS 2025).

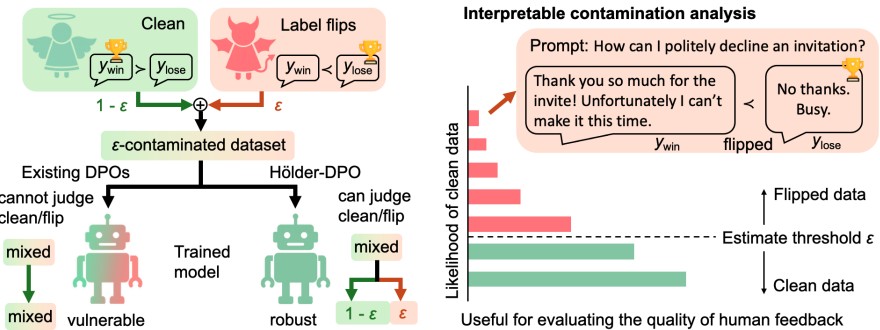

Figure 1: While existing DPO variants are vulnerable to $\epsilon$-contamination, Hölder-DPO is provably robust. It also ranks data points by clean-data likelihood, enabling mislabel identification.

the *redescending* property [74]—the influence of extreme outliers diminishes to zero—a concept in robust statistics defined via the influence function (IF; [45, 27]). We analyze existing DPO variants through IFs and prove that none satisfy the redescending property. If this property holds, the aligned model approximates the clean data distribution—enabling principled mislabel detection, as mislabels naturally exhibit lower likelihoods being clean data points. Crucially, identifying mislabelled data has benefits beyond robustness: it supports annotator evaluation, incentivises higher-quality feedback, and enables scalable dataset cleaning, reducing reliance on costly manual verification.

To address this challenge, we propose Hölder-DPO, the first alignment loss with a provable redescending property, stably learning clean data distribution even under heavy contamination (e.g., $\epsilon = 0.4$). Furthermore, our loss also enables the aligned model itself to serve as a mislabel valuation metric (see Figure 1). Ours is a *first-of-its-kind* principled metric that provides a theoretically grounded estimator of both the likelihood and fraction of (mis-)labelled data. Furthermore, our method is highly scalable—it avoids costly gradient-based computations required by prior work [60, 22] and eliminates the need for manual verification [58], thereby enabling fully automated feedback cleaning. Applied to the Anthropic HH dataset [34], it reveals substantial noise ($\epsilon \approx 0.25$), and removing the identified mislabels significantly improves alignment performance across methods.

## 2 Preliminaries

### 2.1 Language Model Alignment by Direct Preferential Optimization

**Human feedback and Bradley-Terry model.** Let $x \in \mathcal{X}$ be a prompt sampled from a distribution $p(x)$ over a finite context space $\mathcal{X}$. The policy model $\pi(y \mid x)$ generates a pair of responses $(y_1, y_2)$. A human annotator then provides a preference between them, identifying the preferred response as $y_{\text{win}}$, and the rejected response as $y_{\text{lose}}$, i.e., $y_{\text{win}} \succ y_{\text{lose}}$. Although the underlying reward function $r^*(x, y)$, which ranks responses, is unobservable, Bradley–Terry (BT) model [12] provides a standard framework for modelling pairwise comparisons based solely on observed preferences:

$$p^*(y_1 \succ y_2 \mid x, y_1, y_2) := \sigma\left[\exp(r^*(x, y_1)) - \exp(r^*(x, y_2))\right], \tag{1}$$

where $\sigma(a) := 1/(1 + \exp(-a))$ is the logistic function. We define a pairwise data point as $s := \{x, y_{\text{win}}, y_{\text{lose}}\}$, and denote the preference dataset as $\mathcal{D} := \{s^{(i)}\}_{i=1}^N$. Each data point $s^{(i)}$ is sampled independently from $p_{\mathcal{D}}(s)$, where $p_{\mathcal{D}}(s) \propto p^*(y_{\text{win}} \succ y_{\text{lose}} \mid x, y_1, y_2) \cdot \pi(y_1, y_2 \mid x) \cdot p(x)$.

**Direct Preference Optimization (DPO).** The goal of preference-based model alignment is to find parameters $\theta$ such that the policy $\pi_\theta$ accurately approximates either the latent reward function $r^*$ or the preference distribution $p^*$. To achieve this, Rafailov et al. [84] proposed DPO, which efficiently solves the alignment task by minimizing the following optimization problem $\mathcal{L}(s, \pi_\theta)$:

$$\underset{\theta}{\arg\min}\ \mathbb{E}_{p_{\mathcal{D}}(s)}[-\log \sigma(g_\theta(s))], \ \text{where } g_\theta(s) = \beta \log \frac{\pi_\theta(y_{\text{win}} \mid x)}{\pi_{\text{ref}}(y_{\text{win}} \mid x)} - \beta \log \frac{\pi_\theta(y_{\text{lose}} \mid x)}{\pi_{\text{ref}}(y_{\text{lose}} \mid x)}, \tag{2}$$

where $\pi_{\text{ref}}$ is the reference model, and $\beta > 0$ is a hyperparameter controlling alignment strength. The reference model is typically obtained by supervised fine-tuning (SFT) on the prompts $x$ with the preferred response $y_{\text{win}}$ only [84, 86]. The corresponding gradient is given by:

$$\nabla_\theta \mathcal{L}(s, \pi_\theta) := \sigma(-g_\theta(s)) \cdot (\nabla_\theta \log \pi_\theta(y_{\text{win}} \mid x) - \nabla_\theta \log \pi_\theta(y_{\text{lose}} \mid x)), \tag{3}$$

where $g_\theta(s) := \hat{r}_\theta(x, y_{\text{win}}) - \hat{r}_\theta(x, y_{\text{lose}})$ and $\hat{r}_\theta(x, y) = \beta \log(\pi_\theta(y|x)/\pi_{\text{ref}}(y|x))$ denotes the implicit reward function induced by $\pi_\theta$.

## 2.2 Data Contamination Model for Label Flips and Redescending Property

**Data contamination model.** Noise in human preference datasets [35, 105, 99] can arise from two sources: noise in the responses $(y_1, y_2)$ [43, 105], such as irrelevant or incoherent samples, and noise in preference feedback $y_{\text{win}} \succ y_{\text{lose}}$ [35, 87, 28, 23, 105], due to incorrect preference annotations (also known as label flips). As Wu et al. [105] proved that vanilla DPO is robust against noise in responses, we focuses on robustness to label flips. We model the label flip process as follows: We denote a flipped data point as $s_{\text{flip}} := \{x, y_{\text{win}}^{\text{flip}}, y_{\text{lose}}^{\text{flip}}\}$, where $y_{\text{win}}^{\text{flip}} = y_{\text{lose}}$ and $y_{\text{lose}}^{\text{flip}} = y_{\text{win}}$ represent a flipped preference label pair. Such a flipped point $s_{\text{flip}}$ is sampled from the distribution $p(s_{\text{flip}}) \propto p_{\text{flip}}(y_{\text{win}}^{\text{flip}} \succ y_{\text{lose}}^{\text{flip}} \mid x, y_1, y_2) \cdot \pi(y_1, y_2 \mid x) \cdot p(x)$. Contaminated dataset $\widetilde{\mathcal{D}}$ is given by:

**Definition 1 ($\epsilon$-contamination model).** Let $0 \leq \epsilon < 1/2$ be contamination ratio. The generative distribution of $\widetilde{\mathcal{D}}$ under the $\epsilon$-contamination model is defined as $p_{\widetilde{\mathcal{D}}}^{(\epsilon)}(\tilde{s}) = (1 - \epsilon)p_{\mathcal{D}}(s) + \epsilon p(s_{\text{flip}})$, and denote $\tilde{s}$ as a contaminated data point, which can be either clean $s$ or flipped $s_{\text{flip}}$.

Definition 1 is widely adopted in robust statistics [48, 44, 49] and in the machine learning community [38, 33, 75]. We assume access only to the contaminated dataset $p_{\widetilde{\mathcal{D}}}^{(\epsilon)}(\tilde{s})$, without access to a clean validation set $p_{\mathcal{D}}(s)$—a prerequisite in many existing methods [24, 58]. As such, the contamination problem cannot be reduced to a standard classification or distribution shift task. Instead, we leverage a tail contamination assumption to enable estimation of the contamination ratio $\epsilon$:

**Assumption 1 (Tail contamination [32]).** Suppose that $p_{\mathcal{D}} = \sigma(g_{\theta^*}(s))$ for some $\theta^* \in \Theta$, meaning that the true data distribution is exactly recovered by the model. Let $\gamma > 0$. Then, in a neighborhood of $\theta = \theta^*$, $\mathbb{E}_{p(s_{\text{flip}})}[\sigma(g_\theta(s_{\text{flip}}))^\gamma]$ is sufficiently small, i.e., $\mathbb{E}_{p(s_{\text{flip}})}[\sigma(g_\theta(s_{\text{flip}}))^\gamma] \approx 0$ when $\theta \approx \theta^*$.

Intuitively, if we had access to the ground-truth clean-data likelihood $\sigma(g_{\theta^*}(s))$, it would assign high values to clean data points and negligibly small values to flipped ones. This principle is well-established in robust statistics [32, 53], and implicitly adapted in prior work [57, 13, 4, 109]. Moreover, this assumption unlocks heavy contamination scenarios, in contrast to typical assumptions that consider only infinitesimally small perturbations, i.e., $\epsilon \approx 0$ [41, 61, 22]. This distinction is particularly important for noisy human preference datasets (e.g., $\epsilon \approx 0.25$ for HH-RLHF [88, 20]).

**Redescending property.** We evaluate robustness to the data contamination through the learned LLM parameters. Let $\theta^*$ denote the optimal parameters learned from the clean dataset $p_{\mathcal{D}}$, and $\theta^*(\epsilon)$ denote those learned from the $\epsilon$-contaminated dataset $p_{\widetilde{\mathcal{D}}}^{(\epsilon)}$. Intuitively, the impact of contamination can be measured by $\theta^*(\epsilon) - \theta^*$, and its first-order approximation, known as IF [44], is commonly used:

$$\text{IF}(s_{\text{flip}}, \theta, p_{\mathcal{D}}) := \left.\frac{\partial \theta^*(\epsilon)}{\partial \epsilon}\right|_{\epsilon=0}. \tag{4}$$

See Appendix C for further details. Using the IF, we define the robustness condition as follows:

**Definition 2 (Redescending property [74]).** Let $s_{\text{flip}} = (x, y_{\text{win}}^{\text{flip}}, y_{\text{lose}}^{\text{flip}})$ be a preference pair. An objective is said to be robust to $\epsilon$-contamination if the following condition is satisfied:

$$\lim_{\hat{r}_\theta(x, y_{\text{lose}}^{\text{flip}}) \to \infty} \|\text{IF}(s_{\text{flip}}, \theta, p_{\mathcal{D}})\| = 0,$$

where $\|\cdot\|$ denotes the Euclidean norm.

The limit $\hat{r}_\theta(x, y_{\text{lose}}^{\text{flip}}) \to \infty$ occurs when the data point $s_{\text{flip}}$ attains the lowest reward, i.e., $g_\theta(s_{\text{flip}}) \to -\infty$. Yet, the label flip forces the model to learn this least preferable sample as best, representing the most adversarial data-poisoning scenario that a single label flip can induce. Intuitively, if the IF remains zero even under this adversarial condition, the learning objective can be considered robust: the effect of the worst possible data point on the model parameters vanishes, i.e., $\theta^*(\epsilon) \approx \theta^{*2}$.

## 2.3 Related work

**Robust DPOs and RLHFs.** A broad range of DPO variants have been proposed to improve the robustness of the alignment objective [76, 6, 100, 24, 105]. Reinforcement-learning-based approaches also constitute a popular family of alignment methods [26, 8, 78, 55], with several recent works introducing robustified objectives [72, 14, 82]. However, none provides theoretical guarantees for the redescending property or for detecting mislabelled data. We prove their limitations in Section 3.

---

[2]Up to higher-order approximation error $\mathcal{O}(\epsilon^2)$

**Dataset valuation and cleaning.** Dataset valuation has been studied using IFs [41, 61, 22, 64, 63, 108, 52], Shapley values [102], and unrolling [7]. Various heuristic methods have also been proposed for identifying and filtering mislabelled data, including prompt-based filtering [77, 91, 68, 103, 70], attribution-based approaches [101, 36, 11, 79], and perplexity-based filtering [58]. In contrast, our method is the first to provide theoretical guarantees for dataset filtering under heavy contamination.

## 3 Existing DPO Variants Are Neither Robust Nor Able to Identify Mislabels

We first derive the IFs of existing DPO variants and show that none satisfy the redescending property.

**Theorem 2** (**IF for DPO variants**). *Let $\theta^*$ be the optimum learnt from the clean dataset $p_{\mathcal{D}}$, and $\theta^*(\epsilon)$ learnt from the $\epsilon$-contaminated dataset $p_{\mathcal{D}}^{(\epsilon)}$. Let $\mathcal{L}_{\mathrm{gen}}(\pi_\theta; \pi_{\mathrm{ref}})$ be a generic DPO loss function corresponding to a specific DPO variant. Assume that the Hessian $\nabla_\theta^2 \mathcal{L}_{gen}(s, \pi_\theta)|_{\theta=\theta^*}$ is positive definite.[3] Then, the $s_{flip}$-dependent component of the IF for this DPO variants is given by:*

$$\mathrm{IF}(s_{flip}, \theta, p_{\mathcal{D}}) \propto \mathbb{E}_{p(s_{flip})}[\nabla_\theta \mathcal{L}_{\mathrm{gen}}(s_{flip}, \pi_{\theta^*})], \tag{5}$$

*where $\nabla_\theta \mathcal{L}_{\mathrm{gen}}(s_{flip}, \pi_{\theta^*})$ corresponds, for example, to Eq. (3) when $\mathcal{L}_{\mathrm{gen}}(s_{flip}, \pi_{\theta^*}) = \mathcal{L}(s_{flip}, \pi_{\theta^*})$.*

**Theorem 3** (informal). *The following objectives do not satisfy the redescending property:*

| algorithm | objective $\mathcal{L}_{\mathrm{gen}}(\pi_\theta; \pi_{\mathrm{ref}})$ | bounded ? | redescending? | can detect $s_{flip}$? |
|---|---|---|---|---|
| DPO [84] | $\mathbb{E}_{p_{\mathcal{D}}^{(\epsilon)}}[-\log \sigma(g_\theta(\tilde{s}))]$ | ✓ | ✗ | ✗ |
| IPO [6] | $\mathbb{E}_{p_{\mathcal{D}}^{(\epsilon)}}\left[\left(\frac{g_\theta(\tilde{s})}{\beta} - \frac{1}{2\beta}\right)^2\right]$ | ✗ | ✗ | ✗ |
| C-DPO [76] | $(1-c)\mathbb{E}_{p_{\mathcal{D}}^{(\epsilon)}}[-\log \sigma(g_\theta(\tilde{s}))] - c\mathbb{E}_{p_{\mathcal{D}}^{(\epsilon)}}[-\log \sigma(-g_\theta(\tilde{s}))]$ | ✓ | ✗ | ✗ |
| R-DPO [24] | $\frac{1-c}{1-2c}\mathbb{E}_{p_{\mathcal{D}}^{(\epsilon)}}[-\log \sigma(g_\theta(\tilde{s}))] - \frac{c}{1-2c}\mathbb{E}_{p_{\mathcal{D}}^{(\epsilon)}}[-\log \sigma(-g_\theta(\tilde{s}))]$ | ✓ | ✗ | ✗ |
| Dr. DPO [105] | $-\beta' \log \mathbb{E}_{p_{\mathcal{D}}^{(\epsilon)}}\left[\exp\left(\frac{\log \sigma(g_\theta(\tilde{s}))}{\beta'}\right)\right]$ | ✓ | ✗ | ✗ |

See Appendices E and G for full proofs and detailed expressions of the IFs for each DPO variant. We now provide an intuitive interpretation of Theorem 3. The IF can be viewed as the gradient of the generic objective $\mathcal{L}_{\mathrm{gen}}$ (see Eq. (5)), which can be decomposed into two components. For example, in the vanilla DPO case (Eq. (3)), we have:

$$\nabla_\theta \mathcal{L}(s, \pi_\theta) = \underbrace{\sigma(-g_\theta(s))}_{\text{IF weights.}} \underbrace{g_{\mathrm{grad}}(s)}_{\text{direction}}.$$

Here, the gradient term $g_{\mathrm{grad}}(s)$ determines the direction of the update, while the IF weight $\sigma(-g_\theta(s))$ acts as a scaling factor that controls the strength of the update for each data point. If the IF weight tends

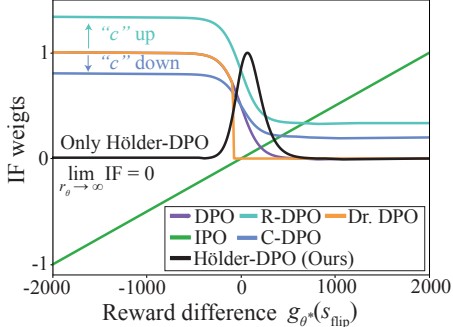

Figure 2: IF analysis reveals only Hölder-DPO satisfies the redescending property.

to zero in the worst-case scenario (i.e., $g_\theta(s_{\mathrm{flip}}) \to -\infty$), the method is robust, as it effectively ignores highly confident but misaligned (flipped) samples. Figure 2 visualises the behavior of IF weights across different objectives under extreme reward perturbations. Both DPO and Dr. DPO exhibit the same asymptotic trend, with $\lim_{r_\theta(x, y_{\mathrm{lose}}^{\mathrm{flip}}) \to \infty} \mathrm{IF} = a$, where $a$ is a non-zero constant. R-DPO shifts this limit upward with a constant $c$, while C-DPO pushes it downward, but neither approaches zero. In contrast, our proposed Hölder-DPO, introduced in the next section, completely nullifies the influence of outliers, demonstrating strong robustness (redescending), not merely boundedness.

## 4 Proposed method: Hölder-DPO

In response to the issues identified in §3, we introduce our robust DPO framework, *Hölder-DPO*, the first algorithm that has a provable redescending property and can identify mislabels $s_{\mathrm{flip}}$.

---

[3]This is a local assumption around the optimal parameters $\theta^*$. Such local assumptions are standard in IF analysis for non-convex deep learning models, as global convexity rarely holds (e.g., [56, 22]).

## 4.1 Hölder-DPO Is Provably Robust

The objective of vanilla DPO in Eq. (2) can be rewritten with KL divergence:

$$\operatorname*{argmin}_{\theta} \underbrace{\mathbb{E}_{p_{\tilde{\mathcal{D}}}^{(\epsilon)}(\tilde{s})}[\log p_{\tilde{\mathcal{D}}}^{(\epsilon)}(\tilde{s})]}_{\text{ignorable const.}} + \underbrace{\mathbb{E}_{p_{\tilde{\mathcal{D}}}^{(\epsilon)}(\tilde{s})}[-\log \sigma(g_\theta(\tilde{s}))]}_{\text{Original DPO loss}} = \operatorname*{argmin}_{\theta} D_{\mathrm{KL}}[p_{\tilde{\mathcal{D}}}^{(\epsilon)}(\tilde{s}) \,\|\, \sigma(g_\theta(\tilde{s}))]. \quad (6)$$

Thus, vanilla DPO can be understood as matching $\sigma(g_\theta(\tilde{s})) \approx p_{\tilde{\mathcal{D}}}^{(\epsilon)}(\tilde{s})$ in terms of KL divergence. However, the KL divergence is well-known to be highly sensitive to data contamination [10, 32], and accordingly, DPO is not robust (Theorem 3). This fact motivates replacing the KL divergence with a more robust alternative. In this paper, we employ the *Hölder divergence* [53], a generalization of the KL divergence that encompasses several robust divergences, defined as follows:

**Definition 3** (Hölder divergence [53]). Let $\phi : \mathbb{R}_+ \to \mathbb{R}$ be a continuous function such that $\phi(h) \geq -h^{1+\gamma}$ for all $h \geq 0$ and $\gamma > 0$, and $\phi(1) = -1$. Then, the Hölder divergence is defined as

$$D_{\mathrm{H}}[p(\omega) \,\|\, q(\omega)] := \begin{cases} S_\gamma(p \,\|\, q) - S_\gamma(p \,\|\, p) & (\gamma > 0), \\ D_{\mathrm{KL}}[p(\omega) \,\|\, q(\omega)] & (\gamma = 0), \end{cases} \quad (7)$$

where $S_\gamma(p \,\|\, q) := \phi\left(\frac{\mathbb{E}_p[q^\gamma(\omega)]}{\mathbb{E}_q[q^\gamma(\omega)]}\right) \cdot \mathbb{E}_q[q^\gamma(\omega)]$, $S_\gamma(p \,\|\, p) := -\mathbb{E}_p[p^\gamma(\omega)]$, and $p$, $q$ are non-negative, non-zero functions.

We then propose to perform LLM alignment by solving the following optimization problem:

$$\operatorname*{argmin}_{\theta} D_{\mathrm{H}}[p_{\tilde{\mathcal{D}}}^{(\epsilon)}(\tilde{s})) \,\|\, \sigma(g_\theta(\tilde{s}))]. \quad (8)$$

**Remark 1.** Prob. (8) generalises the KL divergence and encompasses two robust divergences.

| $\gamma$ | $\phi(h) =$ | divergence | equivalence |
|---|---|---|---|
| $\gamma = 0$ | N/A | KL | vanilla DPO (Eq. (6)) |
| $\gamma > 0$ | $\gamma - (1+\gamma)h$ | density-powered (DP; [10]) | N/A |
| $\gamma > 0$ | $-h^{1+\gamma}$ | pseudo-spherical (PS) score [40, 39] | $\gamma$-divergence [32, 53] |

When $\gamma > 0$, $S_\gamma(p_{\tilde{\mathcal{D}}}^{(\epsilon)} \,\|\, p_{\tilde{\mathcal{D}}}^{(\epsilon)})$ is a negligible constant, Prob. (8) can be further reformulated as

$$\operatorname*{argmin}_{\theta} S_\gamma(p_{\tilde{\mathcal{D}}}^{(\epsilon)}(\tilde{s}) \,\|\, \sigma(g_\theta(\tilde{s}))). \quad (9)$$

We refer to the new objective, Prob. (9), as *Hölder-DPO*. We now analyze its robustness via IF:

**Theorem 4** (IF for Hölder-DPO). *Suppose that $\theta^*(\epsilon) = \operatorname{argmin}_\theta S_\gamma(\tilde{p}_{\mathcal{D}}^{(\epsilon)} \,\|\, \sigma(g_\theta))$ and $\theta^* = \operatorname{argmin}_\theta S_\gamma(p_{\mathcal{D}} \,\|\, \sigma(g_\theta))$. Let $\phi(h)$ be twice-differentiable, and let $0 < \sigma(g_\theta(s))$ and $0 < \gamma < \infty$. Assume that $\phi(h)$ satisfies $\phi'(h) \neq 0$ for $h > 0$ and $\phi(h) \neq c \cdot h$ for any constant $c$[4] and the Hessian $\nabla_\theta^2 \mathcal{L}(s, \pi_\theta)|_{\theta=\theta^*}$ is positive definite. Then, the IF of the Hölder-DPO, excluding terms independent of $s_{flip}$, is given by:*

$$\mathrm{IF}_{\mathrm{H-DPO}}(s_{flip}, \theta, p_{\mathcal{D}}) \propto \mathbb{E}_{p(s_{flip})}[F_{\theta^*}^{(\gamma)}(s_{flip})], \quad (10)$$

*where $F_{\theta^*}^{(\gamma)}(s_{flip}) := \sigma(g_{\theta^*}(s_{flip}))^\gamma \, \nabla_\theta \mathcal{L}(s_{flip}, \pi_{\theta^*})$.*

**Corollary 1** (Hölder-DPO is robust). *Suppose that the policy gradient $\nabla_\theta \log \pi_\theta(y \mid x)$ is bounded by $C$ and satisfies $L$-Lipchitz in $\theta$, where $0 < C < \infty$ and $0 < L < \infty$. Then, under Theorem 4, the IF of Hölder-DPO satisfies the redescending property in Definition 2.*

The full proof and the complete form of Eq. (10) are summarised in Appendices E.3 and E.4. This robustness result is expected, as both the DP divergence and the $\gamma$-divergence are well known to be robust to data contamination across various settings [10, 32, 38, 33]. The key difference between the IFs of DPO variants (Theorem 2) and our method (Theorem 4) lies in the term $\sigma(g_{\theta^*}(s_{flip}))^\gamma$. This subtle change induces diminishing IF weights, as shown in Figure 2, and ensures the redescending.

---

[4]These assumptions are satisfied by the DP divergence and the PS score, as formally shown in Lemma 2.

## 4.2 Hölder-DPO Can Estimate Contamination Ratio $\epsilon$ and Detect Contaminated Data $s_{\text{flip}}$

We now move to the most novel part of our algorithm: contaminated data detection, which is the primary motivation for employing Hölder divergence in addition to its robustness and generality across three divergences. Specifically, we show that our Hölder-DPO inherits the contamination ratio ($\epsilon$) estimation capability of the Hölder divergence [54].

**Model Extension.** Estimating the contamination ratio $\epsilon$ without access to clean validation dataset may sound surprising—how is it possible? The key enabler is the idea of *model extension* [54]. Due to space limitations, we present only the high-level intuition here and defer technical details to Appendix F.2. The crucial insight is that if the learning objective satisfies redescending property—that is, it nullifies the effect of contamination—the divergence can be approximated as:

$$\underset{\theta}{\arg\min}\, S_\gamma[\tilde{p}_{\mathcal{D}}^{(\epsilon)} \,\|\, \sigma(g_\theta(s))] = \underset{\theta}{\arg\min}\, S_\gamma[(1-\epsilon)p_{\mathcal{D}}(s) + \epsilon p(s_{\text{flip}}) \,\|\, \sigma(g_\theta(s))], \quad \text{(Definition 1)}$$

$$\approx \underset{\theta}{\arg\min}\, S_\gamma[(1-\epsilon)p_{\mathcal{D}}(s) \,\|\, \sigma(g_\theta(s))]. \quad \text{(Corollary 1)}$$

Thus, our Hölder-DPO objective can be interpreted as matching $(1-\epsilon)p_{\mathcal{D}}(s) \approx \sigma(g_\theta(s))$. However, the scaling factor $(1-\epsilon)$ remains on the target distribution side. The model extension idea [54] addresses this issue by introducing an additional scaling parameter $\xi$ into the model:

$$\{\theta^*(\epsilon),\ \xi^*\} \approx \underset{\theta,\xi}{\arg\min}\, S_\gamma[(1-\epsilon)p_{\mathcal{D}}(s) \,\|\, \xi\sigma(g_\theta(s))].$$

Intuitively, if $\xi \approx 1-\epsilon$, the objective approximately reduces to the clean form: $S_\gamma(p_{\mathcal{D}}(s) \,\|\, \sigma(g_\theta(s)))$. In other words, the aligned model $\sigma(g_{\theta^*(\epsilon)}(s))$ approximates the distribution of the clean dataset, i.e., $p_{\mathcal{D}}(s) \approx \sigma(g_{\theta^*(\epsilon)}(s))$. Furthermore, the optimised scaling parameter $\xi$ provides an estimate of the contamination ratio $\epsilon$, using $\epsilon \approx 1-\xi$.

**Estimator of $\epsilon$.** The contamination ratio estimate $\hat{\epsilon}$ has the following closed-form:

**Proposition 1 (Contamination ratio estimator).** *For any fixed $\theta$, the optimal solution to the inner optimization problem* $\arg\min_\xi S_\gamma(p_{\tilde{\mathcal{D}}}^{(\epsilon)} \,\|\, \xi\sigma(g_\theta(s)))$ *is given in closed form by* $\xi^* = \mathbb{E}_{\tilde{p}_{\mathcal{D}}^{(\epsilon)}}[\sigma(g_\theta(\tilde{s}))^\gamma] \,/\, \int \sigma(g_\theta(\tilde{s}))^{1+\gamma}\mathrm{d}\tilde{s}$. *Then, the contamination ratio $\epsilon$ can be estimated by:*

$$\hat{\epsilon} = \min\{0, 1-\hat{\xi}^*\}, \quad \text{where} \quad \hat{\xi}^* := \frac{\frac{1}{N}\sum_{i=1}^N \bar{\sigma}(g_\theta(\tilde{s}^{(i)}))^\gamma}{\sum_{i=1}^N \bar{\sigma}(g_\theta(\tilde{s}^{(i)}))^{1+\gamma}}, \quad \bar{\sigma}(g_\theta(\tilde{s}^{(i)})) := \frac{\sigma(g_\theta(\tilde{s}^{(i)}))}{\sum_{i=1}^N \sigma(g_\theta(\tilde{s}^{(i)}))}. \quad (11)$$

Surprisingly, the contamination ratio $\hat{\epsilon}$ can be estimated using the closed-form solution. As a result, Hölder-DPO enables contamination estimation without solving a complex optimization problem. A full derivation and proof are provided in Appendix F.2.

**Choice of $\phi$.** Given two divergence options, DP or PS (see Remark 1), which should we choose? Kanamori and Fujisawa [54] addressed this question in the classical setting and showed that only DP—i.e., $\phi(h) = \gamma - (1+\gamma)h$—satisfies both the redescending property (Corollary 1) and contamination ratio estimation (Proposition 1). We proved that the same theoretical result holds in the model alignment context: neither the PS score nor KL divergence satisfies both properties, while DP does. (see Appendix F.4). Thus, our final objective reduces to the following empirical estimator:

$$\underset{\theta}{\arg\min}\, \widehat{S}_{\text{DP}}\left(p_{\tilde{\mathcal{D}}}^{(\epsilon)} \,\|\, \xi\sigma(g_\theta(\tilde{s}))\right) := \underset{\theta}{\arg\min}\, -\frac{(1+\gamma)}{N}\sum_{i=1}^N \left[\sigma(g_\theta(\tilde{s}^{(i)}))^\gamma\right] + \frac{\gamma}{N}\sum_{i=1}^N \sigma(g_\theta(\tilde{s}^{(i)}))^{1+\gamma}. \quad (12)$$

The details of the derivation of the DP divergence estimator are provided in Appendix D.

## 4.3 Algorithm to Detect Data Contamination

Algorithm 1 summarises the overall procedure for detecting contaminated data points. As shown in Line 1, our proposed Hölder-DPO serves as a plug-and-play replacement for the standard DPO loss. Simply training with the objective in Eq. (12) yields a robust model alignment algorithm. For further analysis, Line 2 estimates the contamination ratio $\hat{\epsilon}$ without

---

**Algorithm 1** Contamination detection

**Require:** $\widetilde{\mathcal{D}}, \pi_\theta, \pi_{\text{ref}}, \gamma$
1: Fine-tune via Prob. (12) $\rightarrow \theta^*(\epsilon)$.
2: Estimate contamination ratio $\hat{\epsilon}$ via Eq. (11).
3: Identify mislabels in $\widetilde{\mathcal{D}}$ by sorting the estimated clean-data likelihood $\sigma(g_{\theta^*(\epsilon)}(\tilde{s}^{(i)}))$ and take least $\lfloor N\hat{\epsilon}\rfloor$ samples.
4: **return** $\theta^*(\epsilon), \hat{\epsilon}, \mathcal{D}_z$

---

requiring access to a clean validation dataset—unlike prior work that depends on one [24, 58]. We then perform dataset valuation using the estimated clean-data likelihood $\sigma(g_{\theta^*(\epsilon)}(\tilde{s}^{(i)}))$, identifying the estimated mislabelled subset.

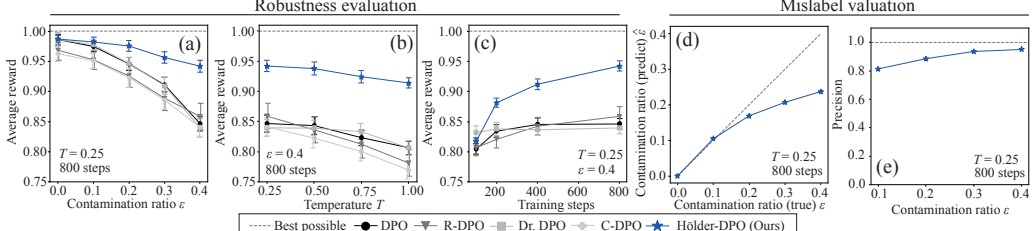

Figure 3: Controlled sentiment generation task using `GPT2-large`. Error bars indicate the standard deviation over 10 trials with different random seeds. Hölder-DPO consistently outperforms all baselines in average reward under varying (a) contamination ratios $\epsilon$, (b) generation temperatures, and (c) training steps. In addition, only Hölder-DPO offers reliable (d) contamination ratio estimation $\hat{\epsilon}$ and (e) precision of mislabel identification, measured by precision as a binary classifier.

## 5 Experiments

We benchmarked Hölder-DPO against two robust DPO variants: R-DPO [24] and Dr. DPO [105]. We also compared to several heuristic baselines, including cDPO [76] and vanilla DPO [84]. As shown in Theorem 3, none of these baselines satisfy the redescending property, though they may offer empirical improvements in some settings. All experiments are conducted on a single NVIDIA A100 GPU (40 GB VRAM, 83.48 GB RAM). For SFT of the reference model, we train for one epoch using context–response pairs $(x, y_{\text{win}})$. Each DPO variant is then trained for five epochs. We implement all methods using the Transformers [96, 104], TRL [98], and PyTorch [80] libraries. We set $\beta = 0.1$ and $\gamma = 2.0$, and all other hyperparameters follow the TRL defaults. Complete implementation details, including Hugging Face URLs to models and datasets used, are provided in Appendix H.

### 5.1 Controlled Sentiment Generation: Robustness Evaluation

**Setup.** We evaluate alignment robustness on a sentiment-controlled text generation task using the IMDb dataset [71], which contains 25,000 polarised movie reviews. Following Chowdhury et al. [24], we extract the first 20 tokens from each review as a context prompt. For each prompt, we generate four responses using the SFT-ed `GPT2-large` [83]. These responses are used to construct six preference pairs per prompt. To provide sentiment rankings, we employ `SiEBERT` [69, 46] as the latent (ground-truth) reward model $r^*(y, x)$. To ensure clean dataset, we filter out preference pairs that do not satisfy $r^*(y_{\text{win}}, x) - r^*(y_{\text{lose}}, x) > 0.1$, yielding 12,000 clean pairs. Of these, 10,000 are used for training and 2,000 for evaluation. To simulate label noise, we randomly flipped $(y_{\text{win}}, y_{\text{lose}})$ in the training set with contamination ratio $\epsilon \in \{0., 0.1, 0.2, 0.3, 0.4\}$. We then fine-tuned `GPT2-large` using various DPO variants. Alignment quality is measured by the average reward assigned by `SiEBERT` to responses generated by the aligned model on the 2,000 evaluation prompts.

**Alignment quality.** Figure 3(a) shows that Hölder-DPO consistently outperforms all baselines in average reward, with only a minor drop as the contamination level $\epsilon$ increases. This supports our theoretical claim: Hölder-DPO, uniquely characterised by the redescending property, achieves robustness even under severe noise (e.g., when $\epsilon \geq 0.3$). A mild linear decay remains, likely due to the reduced proportion of effectively clean data points, which limits estimation accuracy. Figures 3(b) and 3(c) further demonstrate Hölder-DPO's robustness across generation temperatures and training steps. Notably, while the baseline methods plateau early during training, Hölder-DPO continues to improve, suggesting stronger resistance to overfitting on noisy preferences.

**Valuation quality.** Figures 3(d) and 3(e) evaluate mislabel valuation. As shown in Proposition 1, this estimation capability is unique to Hölder-DPO and is not supported by other baselines (see Theorem 3). The detection task is entirely unsupervised—no ground-truth labels indicate whether a preference pair is clean or flipped. Remarkably, Hölder-DPO accurately estimates the contamination ratio up to $\epsilon = 0.2$, after which the estimates begin to saturate. This is expected, given that contamination ratio estimation relies on the optimised $\sigma(g_{\theta^*(\epsilon)})$, and the average reward slightly deteriorates as $\epsilon$ increases (see

Table 1: Effect of $\gamma$ on Hölder-DPO under $\epsilon = 0.4$ contamination.

| $\gamma$ | average reward | $\hat{\epsilon}$ | precision |
|---|---|---|---|
| 1 | 0.8975 | 0.2503 | 0.6433 |
| 2 | 0.9420 | 0.2373 | **0.9286** |
| 4 | 0.9446 | 0.3083 | 0.8902 |
| 6 | **0.9561** | **0.4123** | 0.8328 |
| 8 | 0.9470 | 0.4380 | 0.7860 |

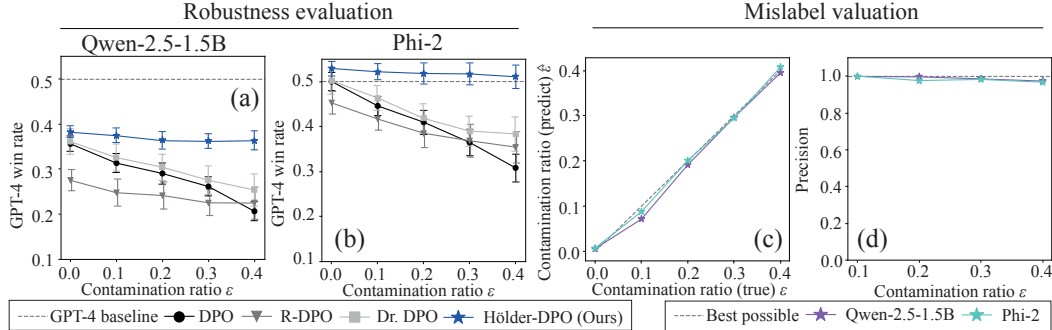

Figure 4: Helpful assistant dialogue generation on the Golden HH dataset. Error bars denote standard deviation over 10 trials with different random seeds. Hölder-DPO consistently achieves the highest `GPT-4` win rate across both base models: (a)`Qwen-2.5-1.5B` and (b)`Phi-2`. Notably, Hölder-DPO is the only method that outperforms `GPT-4`-generated prompts even when using the smaller 2.8B `Phi-2` model. It also delivers near-perfect (d) contamination ratio estimation $\hat{\epsilon}$ and (e) precision in mislabelled data detection, regardless of the base model.

Figure 3(a)). Nonetheless, it reliably ranks datasets by relative noise levels. For mislabel detection, precision—i.e., the proportion of true mislabels among those predicted as mislabelled—increases with higher $\epsilon$, which we attribute to underestimation of $\epsilon$, reducing false positives and improving detection precision.

**Impact of $\gamma$ choice.** We vary the key hyperparameter $\gamma$ to assess its effect. As shown in Table 1, there is a trade-off: increasing $\gamma$ leads to a more accurate contamination ratio estimate $\hat{\epsilon}$ but lowers mislabel detection precision. Intuitively, $\gamma$ controls how strongly potential outliers are down-weighted. While the optimal $\gamma$ is task-dependent and relates to the true (yet unknown) $\epsilon$, Hölder-DPO consistently outperforms all baselines with a fixed default of $\gamma = 2$, suggesting that it is effective without extensive tuning. See Appendix H for further ablation studies on hyperparameters, e.g., batch size.

## 5.2 Single-turn Dialogue: Generated Responce Quality Evaluation

Given a user prompt, the task is to generate responses that are more helpful and less harmful. In this setting, we do not have access to ground-truth rewards (e.g., from a sentiment model like `SiEBERT`). Instead, we adopt the `GPT-4` win rate [84, 105], the standard metric for evaluating alignment when ground truth is unavailable. This metric follows three steps: (A) generate a response $y_{\text{gen}}$ using the aligned model for a given prompt $x$; (B) retrieve the preferred response $y_{\text{win}}$ from the test dataset for the same prompt $x$; (C) ask `GPT-4` to compare $(y_{\text{win}}, y_{\text{gen}})$ and determine which is better. The win rate is defined as the fraction of prompts where $y_{\text{gen}}$ is preferred. For dataset, we evaluate on the Golden HH [15], a manually curated version of Anthropic HH [34], in which low-quality responses have been replaced with `GPT-4`-generated outputs. Since $y_{\text{win}}$ corresponds to `GPT-4` generations, the benchmark is both cleaner and more challenging than the original Anthropic HH. The dataset contains 42,500 training and 2,310 test examples. To evaluate robustness, we randomly flipped preference labels with varying noise levels $\epsilon \in \{0, 0.1, 0.2, 0.3, 0.4\}$. We tested two base models: `Qwen-2.5-1.5B` [106] and `Phi-2` [50], and fine-tuned using DeepSpeed Stage 3 [3] for memory efficiency, with an effective batch size of 32[5]. As shown in Figures 4(a) and 4(b), Hölder-DPO consistently outperforms all baselines across both base models. Notably, Hölder-DPO with `Phi-2` is the only method to exceed a 50% win rate, slightly outperforming `GPT-4`-generated responses. Figures 4(c) and 4(d) further show that mislabel detection is nearly perfect. Remarkably, these results are achieved in a fully unsupervised setting, underscoring the effectiveness of our approach.

**Can Hölder-DPO scale to larger models?** We further evaluated Hölder-DPO on larger language models—`Mistral-8B` [93] and `NeMo-12B` [94]—both capable of multilingual interaction. Experiments were conducted on the OASST1 dataset [89, 59], which contains 18,000 multilingual messages across 35 languages in a

Table 2: GPT-4 win rate on OASST1 ($\epsilon = 0.4$).

| loss | Ministral-8B | NeMo-12B |
|---|---|---|
| DPO | 0.5801 | 0.6021 |
| R-DPO | 0.5737 | 0.5992 |
| Dr. DPO | 0.6058 | 0.6196 |
| Hölder-DPO | **0.6314** | **0.6473** |

[5]Batch size of 4 with gradient accumulation of 8.

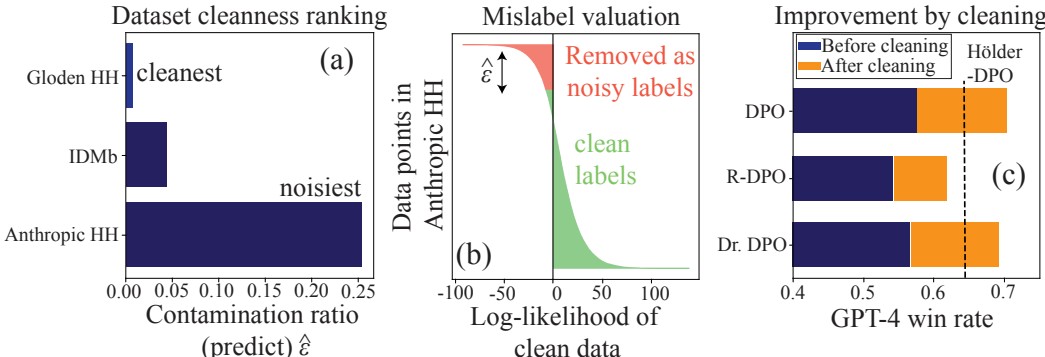

Figure 5: Dataset valuation using `Phi-2`. (a) Hölder-DPO estimates substantial contamination ($\hat{\epsilon} \approx 0.25$) in popular Anthropic HH dataset. (b) Distribution of log-likelihoods of clean data points in the dataset. (c) Improvement in `GPT-4` win rate across methods after removing detected noisy data points from the training set—surpassing even Hölder-DPO trained on the original (noisy) dataset.

multi-turn dialogue format. Preference labels in the training set were flipped with a contamination ratio of $\epsilon = 0.4$. To accommodate limited GPU resources, we apply low-rank adaptation (LoRA; [47]) via the PEFT library [73], along with model quantization using bitsandbytes [30, 31]. Despite these constraints, Hölder-DPO consistently outperforms all baselines, as shown in Table 2.

### 5.3 Real-world Noisy Datasets: Dataset Valuation and Cleaning

To go beyond controlled settings with synthetic noise, we next applied Hölder-DPO to real-world datasets where the true contamination ratio $\epsilon$ is unknown. Using our contamination ratio estimator $\hat{\epsilon}$ in Eq. (11), we performed dataset valuation and cleaning on the Anthropic HH dataset [34], which is widely used but known to contain significant label noise [88, 20, 105]. We adopted `Phi-2` as the base model and apply Hölder-DPO directly to the raw dataset.

**Dataset valuation.** Figure 5(a) shows the estimated contamination levels for three datasets without synthetic label flipping, reflecting inherent label noise. As expected, the Anthropic HH dataset exhibits the highest estimated contamination, with $\hat{\epsilon} \approx 0.25$. This aligns with prior findings reporting substantial noise in Anthropic HH [88, 20, 105, 24]. Figure 5(b) displays the distribution of log-likelihoods, with illustrative examples provided in Appendix H.3. Manual inspection reveals a consistent pattern: for prompts that LLMs are expected to refuse (e.g., harmful or unethical requests), both responses in the removed examples often provide information, leading to inconsistent rankings. These findings show that Hölder-DPO not only serves as a robust alignment objective but also offers interpretable and actionable insights for dataset valuation.

**Data cleaning.** We also evaluated the effectiveness of Hölder-DPO for data cleaning. Here, Hölder-DPO was used as a pre-processing step to identify and remove noisy preference labels, yielding a filtered training dataset. We then compared the `GPT-4` win rates before and after cleaning. As shown in Figure 5(c), removing samples identified as mislabelled consistently improves performance across all methods. Notably, both DPO and Dr. DPO trained on the cleaned dataset outperform Hölder-DPO trained on the original (noisy) dataset. This is expected: while Hölder-DPO is designed to be robust under label noise ($\epsilon > 0$), once the data is cleaned ($\epsilon \approx 0$), robust objectives can be overly conservative, thus KL-based methods like DPO can align more effectively. These results suggest that Hölder-DPO can serve not only as a robust training objective, but also as a valuable pre-filtering tool for noisy preference datasets.

## 6 Discussion

**Role of Hölder-DPO; pre-filtering or new objective?** Hölder-DPO is a versatile framework whose optimal role depends on the application context. When maximising final performance is prioritised over training cost, a two-stage pipeline—first filtering with Hölder-DPO and then fine-tuning using a vanilla DPO objective—yields superior results (see Figure 5(c)). Conversely, when computational efficiency is the priority, a single-stage approach—directly fine-tuning with the Hölder-

DPO objective—offers a cost-effective solution while maintaining strong alignment quality. This flexibility enables practitioners to choose the two strategies according to their constraints.

**How to set $\gamma$?** As shown in Table 1, the value of $\gamma$ has a strong influence on performance. Its optimal setting is likely task- and dataset-dependent. Ideally, dynamically tuning $\gamma$ during training would yield the best results. Nevertheless, our experiments show that a default value of 2.0 performs robustly across diverse settings, and we recommend using it as the default choice. When fine-tuning $\gamma$ is desired, a promising approach is to employ a human-in-the-loop process. As illustrated in Figure 5(b), the Hölder-DPO model can rank data points. This ranking can serve as a diagnostic tool to guide manual inspection: for instance, increasing $\gamma$ if clear mislabels are still being tolerated, or decreasing it if correct preferences are penalised as mislabels.

**What if $\epsilon > 0.5$?** When the noise ratio becomes extreme (e.g., $\epsilon = 0.9$), Hölder-DPO can still identify the 10% of clean samples by isolating them as outliers. In such cases, the roles can be reversed—the identified minority can be treated as the clean subset, while the majority is discarded. This reversal can be validated with minimal human effort by inspecting a few of the top-ranked samples flagged by our method. Hence, Hölder-DPO remains applicable even in highly noisy settings (e.g., $\epsilon > 0.6$). However, detecting mislabeled data becomes fundamentally difficult when $\epsilon \approx 0.5$. We argue that this challenge is not a limitation of our method per se, but rather a theoretical identifiability barrier: when clean and noisy data are equally represented, distinguishing between them is statistically impossible without access to a clean reference set. Our approach inherently assumes a bimodal distribution, where one mode—typically the majority—corresponds to clean data.

**What kind of label noise does the tail assumption cover?** While our tail-contamination assumption formally presumes i.i.d. noise, it is in fact more general than it may appear. Hölder-DPO is effective against a wide range of label noise types—symmetric, asymmetric, and even certain systematic annotator biases—so long as the resulting noisy preference pairs are recognized by the learned model as low-likelihood events (i.e., outliers). This generality represents a clear advantage over prior methods that can only handle symmetric label flips (e.g., [67]). The main limitation arises when the noise is too structured to be identified as an outlier by the model. For instance, if all preference labels associated with a single prompt or narrow topic are consistently flipped, such errors cease to form a distinct "tail" in the data distribution. Instead, they produce a coherent but incorrect signal within that subdomain—one the model may interpret as a genuine, domain-specific preference. Addressing such structured noise requires modeling dependencies across samples rather than treating each observation as independently corrupted. Exploring this direction is a key avenue for future work. Concurrent research has begun tackling these challenges using latent-variable formulations [16].

# 7 Conclusion and Limitation

**Conclusion.** We analyzed the robustness of DPO variants through the lens of the redescending property and showed that none of the existing methods satisfy it. We then introduced Hölder-DPO— the first algorithm with a provable redescending property—alongside a principled data valuation method for estimating contamination and identifying mislabelled data. Our experiments demonstrate that Hölder-DPO improves alignment quality and accurately detects mislabels without access to a clean validation set. Notably, it offers the first theoretically grounded approach to dataset valuation, revealing significant noise in the widely used Anthropic HH dataset. Moreover, filtering out the detected noisy examples further improves performance across alignment methods, positioning Hölder-DPO as an effective pre-training filter. Taken together, our theoretical and empirical results advance scalable, automatable alignment without relying on clean validation data.

**Limitations.** A key limitation lies in the selection of the hyperparameter $\gamma$ and see Section 6 for the potential solutions. Another limitation is that our theoretical guarantees are also based on an i.i.d. label flip model and see also Section 6 for the details. Furthermore, our objective relies on a coarse approximation of the DP divergence (see Appendix D), and exploring more precise approximation methods is another important avenue. Finally, we implicitly assume that majority opinion represents the clean data; see Appendix A for the broader implications.

## Acknowledgments

We sincerely appreciate the anonymous reviewers for their insightful feedback. We thank Dr. Siu Lun Chau, Juliusz Ziomek, and Yuki Tachibana for their valuable discussions. This research was primarily conducted while MF was a visiting researcher at the University of Oxford. MF was previously supported by the RIKEN Special Postdoctoral Researcher Program and JST ACT-X (Grant Number: JPMJAX210K), Japan. MF is currently supported by KAKENHI (Grant Number: 25K21286). MA was supported by the Clarendon Fund, the Oxford Kobe Scholarship, the Watanabe Foundation, and Toyota Motor Corporation.

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

# Part I

# Appendix

## Table of Contents

## A  Broader impact statement

Robust model alignment and reliable human feedback valuation are critical for the safe deployment of large language models (LLMs). While our work may have various societal implications, we do not identify any specific risks that require immediate attention.

Conceptually, our definition of "robustness" aligns with a utilitarian perspective: we assume that the majority opinion in the training dataset represents the true target distribution, and that minority opinions deviating from this majority are noise to be filtered out. This assumption is often reasonable in applications like toxicity removal or instruction following, where the normative standard tends to be implicitly shared by the majority.

However, we acknowledge that this framework does not universally apply. In contexts such as opinion formation or deliberative dialogue, disregarding minority voices is inappropriate and potentially harmful. In such cases, distributionally robust methods (e.g., [85, 21]) that account for underrepresented groups are more suitable. Fundamentally, these challenges relate to the broader problem of aggregating heterogeneous preferences into a single model—a problem well-known in

social choice theory. According to the impossibility theorem [5], no aggregation rule (or "social welfare functional") can simultaneously satisfy all desirable rationality axioms. As such, there is no universally optimal solution, and algorithmic choices must be guided by task-specific priorities (see, e.g., [2]).

## B   Additional Discussion of Related Work

**Theoretical Role of Influence Functions.**   While influence functions (IFs) have been widely used for dataset valuation and model interpretation [41, 61], their use in deriving *robustness guarantees* remains less explored. Classical works in robust statistics [49, 44] provide foundational tools for analyzing model behavior under infinitesimal contamination. Our work draws a conceptual bridge between these classical formulations and the practical challenges in LLM alignment, utilizing IFs to derive sufficient conditions for redescending robustness and contamination detection.

**Limitations of Prior Robust DPOs.**   Recent works propose various robust DPO formulations, including DRO-based [105], noise-aware [23], and filtering-enhanced methods [66]. However, many of these approaches either break the connection to reward learning or rely on strong assumptions or additional supervision. Our Hölder-DPO maintains a clean theoretical formulation with a robustness guarantee, while being computationally efficient and easy to implement.

**Empirical Noise Levels and Filtering Limitations.**   Empirical studies report that preference datasets often contain 20–40% noise [35, 62], with performance degrading sharply under modest increases in noise. Common mitigation strategies, such as regularization [35] and teacher-based filtering [35, 17], suffer from limited generalization and inefficiency against symmetric noise. Our approach requires neither external LLMs nor manual heuristics, offering a lightweight alternative grounded in divergence-based theory.

**Broader Applicability.**   Our framework naturally extends to settings like:

- **Group-specific Alignment:** Supporting heterogeneous user preferences [85, 19].
- **Personalized Objectives:** Aligning LLMs to individual user intents [81].
- **DPO with Divergence Constraints:** Leveraging $f$-divergences to balance alignment and diversity [100].

## C   Influence Function to Measure the Impact of the data contamination for DPO

Given the application of DPO and under the first-order optimality conditions, the model alignment results for $p_{\mathcal{D}}(s)$ and $p_{\mathcal{D}}^{(\epsilon)}(\tilde{s})$ are expressed as:

$$\theta^* = \operatorname*{argmin}_{\theta} \mathbb{E}_{p_{\mathcal{D}}(s)}[-\log \sigma(g_\theta(s))] \quad \text{and} \quad \theta^*(\epsilon) = \operatorname*{argmin}_{\theta} \mathbb{E}_{p_{\mathcal{D}}^{(\epsilon)}(\tilde{s})}[-\log \sigma(g_\theta(\tilde{s}))]. \tag{13}$$

From our definition of contamination data in Section 2.2, its influence on model alignment can be quantified by measuring the deviation in the optimized parameters, given by $\|\theta^*(\epsilon) - \theta^*\|$. To evaluate this quantity, we apply a Taylor expansion with respect to $\epsilon$ around $\epsilon = 0$ for $\theta^*(\epsilon)$, yielding:

$$\theta^*(\epsilon) = \theta^* + \epsilon \cdot \left.\frac{\partial \theta^*(\epsilon)}{\partial \epsilon}\right|_{\epsilon=0} + \mathcal{O}(\epsilon^2) \Rightarrow \theta^*(\epsilon) - \theta^* = \epsilon \cdot \left.\frac{\partial \theta^*(\epsilon)}{\partial \epsilon}\right|_{\epsilon=0} + \mathcal{O}(\epsilon^2).$$

From this result, we can approximately evaluate the influence of $s_{\text{flip}}$ as follows:

$$\mathrm{IF}(z, \theta, p_{\mathcal{D}}) \coloneqq \left.\frac{\partial \theta^*(\epsilon)}{\partial \epsilon}\right|_{\epsilon=0}, \tag{14}$$

which is called as the *influence function* (IF) in the robust statistic context [44]. In the next section, we provide a detailed discussion on the robustness of DPO to contamination data through an analysis of this IF.

# D Computation of Hölder-DPO Objective

Here, we describe the computation of the Hölder-DPO objective introduced in Eq. (8). Given the dataset $\mathcal{D} = \{s^{(i)}\}_{i=1}^N$ with $s^{(i)} \sim p_\mathcal{D}$, our optimization objective is:

$$\underset{\theta}{\operatorname{argmin}}\, S_\gamma(p_\mathcal{D}\|\sigma(g_\theta)) \Rightarrow \underset{\theta}{\operatorname{argmin}}\, \phi\left(\frac{\mathbb{E}_{p_\mathcal{D}}[\sigma(g_\theta(s))^\gamma]}{\int \sigma(g_\theta(s))^{1+\gamma}\mathrm{d}s}\right) \cdot \left(\int \sigma(g_\theta(s))^{1+\gamma}\mathrm{d}s\right).$$

The numerator $\mathbb{E}_{p_\mathcal{D}}[\sigma(g_\theta(s))^\gamma]$ is directly estimated from the empirical distribution. However, the integral in the denominator is generally intractable. To address this, we approximate it using the empirical measure $\mathrm{d}\hat{g}(s) \coloneqq \frac{1}{N}\sum_{i=1}^N \delta(s - s^{(i)})$, yielding:

$$\int \sigma(g_\theta(s))^{1+\gamma}\mathrm{d}\hat{g}(s) = \frac{1}{N}\sum_{i=1}^N \sigma(g_\theta(s^{(i)}))^{1+\gamma},$$

where $\delta$ is the Dirac's delta function.

Thus, our estimator can be expressed as:

$$\widehat{S}_\gamma(p_\mathcal{D}\|\sigma(g_\theta)) = \phi\left(\frac{\frac{1}{N}\sum_{i=1}^N \sigma(g_\theta(s^{(i)}))^\gamma}{\frac{1}{N}\sum_{i=1}^N \sigma(g_\theta(s^{(i)}))^{1+\gamma}}\right) \cdot \left(\frac{1}{N}\sum_{i=1}^N \sigma(g_\theta(s^{(i)}))^{1+\gamma}\right).$$

For the special case where $\phi(h) = \gamma - (1+\gamma)h$, corresponding to the scaled density power divergence, the objective simplifies to:

$$\widehat{S}_{\mathrm{DP}}(p_\mathcal{D}\|\sigma(g_\theta)) = -\frac{(1+\gamma)}{N}\sum_{i=1}^N \sigma(g_\theta(s^{(i)}))^\gamma + \frac{\gamma}{N}\sum_{i=1}^N \sigma(g_\theta(s^{(i)}))^{1+\gamma}.$$

An alternative approach for constructing a more precise estimator is to employ importance sampling. However, in the context of LLM fine-tuning, selecting an appropriate proposal distribution is nontrivial. While this issue lies beyond the scope of the present work, it constitutes an important direction for future research. Importantly, our empirical results confirm that this objective still retains the key robustness properties of DP divergence: it remains resistant to label-flipped dataset and enables accurate estimation of the contamination ratio in practice (see Section 5).

# E Proofs

## E.1 Proof for Theorem 2 (The Case of DPO)

**Theorem 5** (IF for DPO). *Suppose $\theta^*$ denotes the optimal parameters learned from clean dataset $p_\mathcal{D}$, and $\theta^*(\epsilon)$ denotes those learned from $\epsilon$-contaminated dataset $p_{\widetilde{\mathcal{D}}}^{(\epsilon)}$. Assume that the Hessian $\nabla_\theta^2 \mathcal{L}(s, \pi_\theta)|_{\theta=\theta^*}$ is positive definite. Then, the $s_{\text{flip}}$-dependent component of the IF for DPO is given by:*

$$\mathrm{IF}_{\mathrm{DPO}}(x, \theta, p_\mathcal{D}) \propto \mathbb{E}_{p(s_{\text{flip}})}[\nabla_\theta \mathcal{L}(s_{\text{flip}}, \pi_{\theta^*})], \tag{15}$$

*where $\nabla_\theta \mathcal{L}(s_{\text{flip}}, \pi_{\theta^*})$ is in Eq. (3).*

*Proof.* The gradient of Eq. (2) under $p_{\widetilde{\mathcal{D}}}^{(\epsilon)}$ is given by

$$\nabla_\theta \widetilde{\mathcal{L}}(\tilde{s}, \pi_\theta) = -\beta \mathbb{E}_{p_{\widetilde{\mathcal{D}}}^{(\epsilon)}}\left[\sigma(-g_\theta(\tilde{s}))\left(\nabla_\theta \log \pi_\theta(\tilde{y}_{\text{win}} \mid \tilde{x}) - \nabla_\theta \log \pi_\theta(\tilde{y}_{\text{lose}} \mid \tilde{x})\right)\right],$$

where $\tilde{s} = \{\tilde{x}, \tilde{y}_{\text{win}}, \tilde{y}_{\text{lose}}\} \sim p_{\widetilde{\mathcal{D}}}^{(\epsilon)}$.

From the definition of $\theta^*(\epsilon)$, we have $0 = \nabla_\theta \widetilde{\mathcal{L}}(\tilde{s}, \pi_\theta)|_{\theta=\theta^*(\epsilon)}$. By taking the derivation of this term w.r.t. $\epsilon$, we obtain

$$
\begin{aligned}
0 &= \frac{\partial}{\partial \epsilon} \nabla_\theta \widetilde{\mathcal{L}}(\tilde{s}, \pi_\theta) \Big|_{\theta=\theta^*(\epsilon)} \\
&= -\beta \frac{\partial}{\partial \epsilon} \mathbb{E}_{p_{\widetilde{\mathcal{D}}}^{(\epsilon)}} \Big[ \underbrace{\sigma(-g_{\theta^*(\epsilon)}(\tilde{s})) \Big( \nabla_\theta \log \pi_{\theta^*(\epsilon)}(\tilde{y}_{\mathrm{win}} \mid \tilde{x}) - \nabla_\theta \log \pi_{\theta^*(\epsilon)}(\tilde{y}_{\mathrm{lose}} \mid \tilde{x}) \Big)}_{=:F_{\theta^*(\epsilon)}(\tilde{s})} \Big] \\
&= -\beta \Big\{ \int \Big\{ \frac{\partial}{\partial \epsilon} p_{\widetilde{\mathcal{D}}}^{(\epsilon)}(\tilde{s}) \Big\} F_{\theta^*(\epsilon)}(\tilde{s}) \mathrm{d}\tilde{s} + \mathbb{E}_{p_{\widetilde{\mathcal{D}}}^{(\epsilon)}} \Big[ \frac{\partial}{\partial \epsilon} F_{\theta^*(\epsilon)}(\tilde{s}) \Big] \Big\} \\
&= -\beta \Big\{ \int \Big\{ \frac{\partial}{\partial \epsilon} p_{\widetilde{\mathcal{D}}}^{(\epsilon)}(\tilde{s}) \Big\} F_{\theta^*(\epsilon)}(\tilde{s}) \mathrm{d}\tilde{s} + \mathbb{E}_{p_{\widetilde{\mathcal{D}}}^{(\epsilon)}} \Big[ \frac{\partial \theta^*(\epsilon)}{\partial \epsilon} \frac{\partial F_{\theta^*(\epsilon)}(\tilde{s})}{\partial \theta^*(\epsilon)} \Big] \Big\} \\
&= -\beta \Big\{ \int \Big\{ \frac{\partial}{\partial \epsilon} p_{\widetilde{\mathcal{D}}}^{(\epsilon)}(\tilde{s}) \Big\} F_{\theta^*(\epsilon)}(\tilde{s}) \mathrm{d}\tilde{s} + \mathbb{E}_{p_{\widetilde{\mathcal{D}}}^{(\epsilon)}} \Big[ \frac{\partial \theta^*(\epsilon)}{\partial \epsilon} H_{\theta^*(\epsilon)}(\tilde{s}) \Big] \Big\},
\end{aligned}
$$
(16)

where $H_{\theta^*(\epsilon)}(\tilde{s}) := \frac{\partial F_{\theta^*(\epsilon)}(\tilde{s})}{\partial \theta^*(\epsilon)}$.

From the definition of $p_{\widetilde{\mathcal{D}}}^{(\epsilon)}(\tilde{s})$, we obtain

$$
\int \Big\{ \frac{\partial}{\partial \epsilon} p_{\widetilde{\mathcal{D}}}^{(\epsilon)}(\tilde{s}) \Big\} F_{\theta^*(\epsilon)}(\tilde{s}) \mathrm{d}\tilde{s} = \mathbb{E}_{p(s_{\mathrm{flip}})} \Big[ F_{\theta^*(\epsilon)}(s_{\mathrm{flip}}) \Big] - \mathbb{E}_{p_{\mathcal{D}}} \Big[ F_{\theta^*(\epsilon)}(s) \Big],
$$

since $\frac{\partial}{\partial \epsilon} p_{\widetilde{\mathcal{D}}}^{(\epsilon)}(\tilde{s}) = p(s_{\mathrm{flip}}) - p_{\mathcal{D}}(s)$, where $F_{\theta^*}(s_{\mathrm{flip}}) := \sigma(-g_{\theta^*}(s_{\mathrm{flip}}))(\nabla_\theta \log \pi_{\theta^*}(y_{\mathrm{win}}^{\mathrm{flip}} \mid z) - \nabla_\theta \log \pi_{\theta^*}(y_{\mathrm{lose}}^{\mathrm{flip}} \mid z))$. By taking $\epsilon \to 0$, we have

$$
\Big( \int \Big\{ \frac{\partial}{\partial \epsilon} p_{\widetilde{\mathcal{D}}}^{(\epsilon)}(\tilde{s}) \Big\} F_{\theta^*(\epsilon)}(\tilde{s}) \mathrm{d}\tilde{s} \Big) \Big|_{\epsilon=0} = \mathbb{E}_{p(s_{\mathrm{flip}})} \Big[ F_{\theta^*}(s_{\mathrm{flip}}) \Big],
$$

since $\theta^{(*)}(\epsilon) \to \theta^{(*)}$ and thus $\mathbb{E}_{p_{\mathcal{D}}}[F_{\theta^*}(s)] = \nabla_\theta \mathcal{L}(\pi_\theta; \pi_{\mathrm{ref}})|_{\theta=\theta^*} = 0$ from the first-order optimal condition in Eq. (13).

Furthermore, we also obtain

$$
\mathbb{E}_{p_{\widetilde{\mathcal{D}}}^{(\epsilon)}} \Big[ \frac{\partial \theta^*(\epsilon)}{\partial \epsilon} H_{\theta^*(\epsilon)}(\tilde{s}) \Big] \Big|_{\epsilon=0} = \mathbb{E}_{p_{\mathcal{D}}} \Big[ \frac{\partial \theta^*(\epsilon)}{\partial \epsilon} H_{\theta^*}(s) \Big],
$$

where $H_{\theta^*}(s) := \frac{\partial F_{\theta^*}(s)}{\partial \theta^*}$.

Then, Eq. (16) under $\epsilon \to 0$ can be rewritten as

$$
0 = \Big( \frac{\partial}{\partial \epsilon} \nabla_\theta \widetilde{\mathcal{L}}(\pi_\theta; \pi_{\mathrm{ref}}) \Big|_{\theta=\theta^*(\epsilon)} \Big) \Big|_{\epsilon=0} = -\beta \Big\{ \mathbb{E}_{p(s_{\mathrm{flip}})} \Big[ F_{\theta^*}(s_{\mathrm{flip}}) \Big] + \mathbb{E}_{p_{\mathcal{D}}} \Big[ \frac{\partial \theta^*(\epsilon)}{\partial \epsilon} \Big|_{\epsilon=0} H_{\theta^*}(s) \Big] \Big\}.
$$

By solving the above equality w.r.t. $\frac{\partial \theta^*(\epsilon)}{\partial \epsilon}$, we obtain

$$
\frac{\partial \theta^*(\epsilon)}{\partial \epsilon} \Big|_{\epsilon=0} = -\Big( \mathbb{E}_{p_{\mathcal{D}}} [H_{\theta^*}(s)] \Big)^{-1} \mathbb{E}_{p(s_{\mathrm{flip}})} \Big[ F_{\theta^*}(s_{\mathrm{flip}}) \Big].
$$

This completes the proof. □

### E.2 Proof for Theorem 3 (The Case of DPO)

**Corollary 2 (DPO is not robust).** *Suppose that the policy gradient $\nabla_\theta \log \pi_\theta(y \mid x)$ is bounded by $C$ and satisfies L-Lipchitz in $\theta$, where $0 < C < \infty$ and $0 < L < \infty$. Then, under Theorem 2, the IF of DPO does not satisfy the redescending property in Definition 2, i.e., $\lim_{\hat{r}_{\theta^*}(x, y_{\mathrm{lose}}^{\mathit{flip}}) \to \infty} \|\mathrm{IF}_{\mathrm{DPO}}(x, \theta, p_{\mathcal{D}})\| \neq 0$.*

*Proof.* From the positive definite assumption on the Hessian in Theorem 5 and the $L$-Lipschitz assumption on the gradient, it follows that $\mathbb{E}_{p_\mathcal{D}}[H_{\theta^*}(s)]$ is a positive definite matrix. Let $L' = \lambda_{\min}(\mathbb{E}_{p_\mathcal{D}}[H_{\theta^*}(s)]) > 0$ be its minimum eigenvalue. Then the norm of its inverse is bounded: $\|(\mathbb{E}_{p_\mathcal{D}}[H_{\theta^*}(s)])^{-1}\| = 1/L'$. Furthermore, from the assumption that $\|\nabla_\theta \log \pi_\theta(y \mid x)\| \leq C$ ($0 < C < \infty$), we have $\|\nabla_\theta \log \pi_\theta(y_{\text{win}} \mid x) - \nabla_\theta \log \pi_\theta(y_{\text{lose}} \mid x)\| \leq 2C$ from the triangle inequality. Taking the limit and applying Jensen's inequality and the bounded convergence theorem, we have:

$$\lim_{\hat{r}_{\theta^*}(x, y_{\text{lose}}^{\text{flip}}) \to \infty} \|\text{IF}_{\text{DPO}}(x, \theta, p_\mathcal{D})\|$$

$$\leq \lim_{\hat{r}_{\theta^*}(x, y_{\text{lose}}^{\text{flip}}) \to \infty} \left\| \left( \mathbb{E}_{p_\mathcal{D}}[H_{\theta^*}(s)] \right)^{-1} \right\| \cdot \lim_{\hat{r}_{\theta^*}(x, y_{\text{lose}}^{\text{flip}}) \to \infty} \left\| \mathbb{E}_{p(s_{\text{flip}})} \left[ F_{\theta^*}(s_{\text{flip}}) \right] \right\|$$

$$\leq (1/L') \cdot \lim_{\hat{r}_{\theta^*}(x, y_{\text{lose}}^{\text{flip}}) \to \infty} \mathbb{E}_{p(s_{\text{flip}})}[\sigma(-g_{\theta^*}(s_{\text{flip}}))] \cdot 2C$$

$$= (1/L') \cdot 1 \cdot 2C = 2C/L'.$$

The fact $0 < 2C/L' < \infty$ according to $0 < C < \infty$ and $0 < L' < \infty$ completes the proof. $\square$

### E.3 Proof for Theorem 4

Before showing Theorem 4, we introduce the following proposition regarding the gradient of $S_\gamma(p_\mathcal{D} \| \sigma(g_\theta))$.

**Proposition 2.** *Suppose that $\nabla_\theta g_\theta(s)$ is bounded. Then, under $p_\mathcal{D}$, the gradient of $S_\gamma(p_\mathcal{D} \| \sigma(g_\theta))$ w.r.t. $\theta$ is obtained as*

$$\nabla_\theta S_\gamma(p_\mathcal{D} \| \sigma(g_\theta)) = \gamma \phi'(h_\theta) \mathbb{E}_{p_\mathcal{D}}[\sigma(g_\theta(s))^\gamma (1 - \sigma(g_\theta(s))) \nabla_\theta g_\theta(s)]$$

$$+ (1 + \gamma) \cdot \left( \phi(h_\theta) - h_\theta \cdot \phi'(h_\theta) \right) \cdot \left( \int \sigma(g_\theta(s))^{1+\gamma} (1 - \sigma(g_\theta(s))) \nabla_\theta g_\theta(s) \mathrm{d}s \right),$$

*where $\phi'(h_\theta) := \nabla_{h_\theta} \phi(h_\theta)$ and $h_\theta := \frac{\mathbb{E}_{p_\mathcal{D}}[\sigma(g_\theta(s))^\gamma]}{\int \sigma(g_\theta(s))^{1+\gamma} \mathrm{d}s}$.*

*Proof.* We start by introducing the following shorthand notation:

$$A(\theta) := \mathbb{E}_{p_\mathcal{D}}[\sigma(g_\theta(s))^\gamma], \quad B(\theta) := \int \sigma(g_\theta(s))^{1+\gamma} \mathrm{d}s,$$

and $h_\theta := \frac{A(\theta)}{B(\theta)}$. Then, $S_\gamma(p_\mathcal{D} \| \sigma(g_\theta))$ can be expressed as $S_\gamma(p_\mathcal{D} \| \sigma(g_\theta)) = \phi(h_\theta) \cdot B(\theta)$. Taking the derivative with respect to $\theta$ using the product rule, we have

$$\nabla_\theta S_\gamma(p_\mathcal{D} \| \sigma(g_\theta)) = \nabla_\theta \left( \phi(h_\theta) \cdot B(\theta) \right) = \phi'(h_\theta) \cdot \nabla_\theta h_\theta \cdot B(\theta) + \phi(h_\theta) \cdot \nabla_\theta B(\theta). \quad (17)$$

We first evaluate $\nabla_\theta h_\theta$, which can be calculated as

$$\nabla_\theta h_\theta = \frac{B(\theta) \nabla_\theta A(\theta) - A(\theta) \nabla_\theta B(\theta)}{B(\theta)^2}.$$

By substituting this into Eq. (17), we obtain

$$\nabla_\theta S_\gamma(p_\mathcal{D} \| \sigma(g_\theta)) = \phi'(h_\theta) \cdot \frac{B(\theta) \nabla_\theta A(\theta) - A(\theta) \nabla_\theta B(\theta)}{B(\theta)} + \phi(h_\theta) \cdot \nabla_\theta B(\theta)$$

$$= \phi'(h_\theta) \cdot \left( \nabla_\theta A(\theta) - h_\theta \nabla_\theta B(\theta) \right) + \phi(h_\theta) \cdot \nabla_\theta B(\theta)$$

$$= \phi'(h_\theta) \cdot \nabla_\theta A(\theta) + \left( \phi(h_\theta) - h_\theta \cdot \phi'(h_\theta) \right) \cdot \nabla_\theta B(\theta).$$

We next evaluate $\nabla_\theta A(\theta) = \nabla_\theta \mathbb{E}_{p_\mathcal{D}}[\sigma(g_\theta(s))^\gamma]$. Since $p_\mathcal{D}$ does not depend on $\theta$, from the chain rule, we can see $\nabla_\theta \mathbb{E}_{p_\mathcal{D}}[\sigma(g_\theta(s))^\gamma] = \mathbb{E}_{p_\mathcal{D}}[\nabla_\theta \sigma(g_\theta(s))^\gamma]$, where

$$\nabla_\theta \sigma(g_\theta(s))^\gamma = \gamma \sigma(g_\theta(s))^{\gamma-1} \nabla_\theta \sigma(g_\theta(s))$$

$$= \gamma \sigma(g_\theta(s))^{\gamma-1} \cdot \left( \sigma(g_\theta(s))(1 - \sigma(g_\theta(s))) \nabla_\theta g_\theta(s) \right)$$

$$= \gamma \sigma(g_\theta(s))^\gamma (1 - \sigma(g_\theta(s))) \nabla_\theta g_\theta(s).$$

As for $\nabla_\theta B(\theta)$, we obtain

$$\nabla_\theta B(\theta) = \nabla_\theta \int \sigma(g_\theta(s))^{1+\gamma} \mathrm{d}s = \int \nabla_\theta \sigma(g_\theta(s))^{1+\gamma} \mathrm{d}s$$

$$= (1 + \gamma) \int \sigma(g_\theta(s))^{1+\gamma} (1 - \sigma(g_\theta(s))) \nabla_\theta g_\theta(s) \mathrm{d}s,$$

where the second equality comes from the dominated convergence theorem due to the fact that the derivative of $\sigma(g_\theta)$ is bounded and continuous under the bounded $\nabla_\theta g(s)$.

By substituting the results of $\nabla_\theta A(\theta)$ and $\nabla_\theta B(\theta)$ into Eq. (17), we have

$$\nabla_\theta S_\gamma(p_\mathcal{D} \| \sigma(g_\theta)) = \gamma \phi'(h_\theta) \mathbb{E}_{p_\mathcal{D}}[\sigma(g_\theta(s))^\gamma (1 - \sigma(g_\theta(s))) \nabla_\theta g_\theta(s)]$$

$$+ (1 + \gamma) \cdot \left( \phi(h_\theta) - h_\theta \cdot \phi'(h_\theta) \right) \cdot \left( \int \sigma(g_\theta(s))^{1+\gamma} (1 - \sigma(g_\theta(s))) \nabla_\theta g_\theta(s) \mathrm{d}s \right).$$

This completes the proof. $\qquad\square$

Furthermore, we show the following lemma to precisely derive the IF for Hölder-DPO.

**Lemma 1.** *Suppose that $\theta^* = \mathrm{argmin}_\theta S_\gamma(p_\mathcal{D} \| \sigma(g_\theta))$. Let $0 < \gamma < \infty$, and let $0 < \sigma(g_\theta(s))$. Assume that $\phi(h)$ satisfies $\phi'(h) \neq 0$ for $h > 0$ and $\phi(h) \neq c \cdot h$ for any constant c. Then, the first-order optimal condition of Hölder-DPO, $0 = \nabla_\theta S_\gamma(p_\mathcal{D} \| \sigma(g_\theta)) |_{\theta=\theta^*}$, holds if and only if $\mathbb{E}_{p_\mathcal{D}}[F_{\theta^*}^{(\gamma)}(s)] = 0$ and $\int F_{\theta^*}^{(1+\gamma)}(s) \mathrm{d}s = 0$, where $F_{\theta^*}^{(\gamma)}(s) := \sigma(g_\theta(s))^\gamma (1 - \sigma(g_\theta(s))) \nabla_\theta g_\theta(s)$.*

*Proof.* From Proposition 2, the first-order optimal condition is given by:

$$0 = \nabla_\theta S_\gamma(p_\mathcal{D} \| \sigma(g_\theta)) \bigg|_{\theta=\theta^*}$$

$$= \underbrace{\gamma \phi'(h\theta)}_{C_1} \underbrace{\mathbb{E}p_\mathcal{D}[F_\theta^{(\gamma)}(s)]}_{X} + \underbrace{(1 + \gamma)(\phi(h\theta) - \phi'(h_\theta)h_\theta)}_{C_2} \underbrace{\left( \int F^{(1+\gamma)} \theta(s) \mathrm{d}s \right)}_{Y}.$$

This is a linear combination $C_1 X + C_2 Y = 0$. The "if and only if" statement holds if we can show that both coefficients $C_1$ and $C_2$ are non-zero.

First, we show $h_{\theta^*} > 0$. By definition, $h_{\theta^*} = A(\theta^*)/B(\theta^*)$, where $A(\theta^*) = \mathbb{E}_{p_\mathcal{D}}[\sigma(g_{\theta^*}(s))^\gamma]$ and $B(\theta^*) = \int \sigma(g_{\theta^*}(s))^{1+\gamma} \mathrm{d}s$. From the premise $0 < \sigma(g_\theta(s))$ and $\gamma > 0$, we have $\sigma(g_{\theta^*}(s))^\gamma > 0$ and $\sigma(g_{\theta^*}(s))^{1+\gamma} > 0$. Therefore, $A(\theta^*) > 0$ and $B(\theta^*) > 0$, which implies $h_{\theta^*} > 0$. Given $\gamma > 0$, $h_{\theta^*} > 0$, and our assumption $\phi'(h) \neq 0$ for $h > 0$, the first coefficient $C_1 = \gamma \phi'(h_{\theta^*}) \neq 0$.

The second coefficient $C_2$ is zero if and only if $\phi(h_{\theta^*}) - h_{\theta^*} \phi'(h_{\theta^*}) = 0$. This condition holds if $\phi(h)$ is a homogeneous function of degree 1, i.e., $\phi(h) = c \cdot h$. By our assumption $\phi(h) \neq c \cdot h$, this implies $\phi(h_{\theta^*}) - h_{\theta^*} \phi'(h_{\theta^*}) \neq 0$, and thus $C_2 \neq 0$. Since both coefficients $C_1$ and $C_2$ are non-zero, the optimality condition $C_1 X + C_2 Y = 0$ holds if and only if $X = \mathbb{E}_{p_\mathcal{D}}[F_{\theta^*}^{(\gamma)}(s)] = 0$ and $Y = \int F_{\theta^*}^{(1+\gamma)}(s) \mathrm{d}s = 0$. This completes the proof. $\qquad\square$

We remark that the conditions of $\phi(h)$, $\phi'(h) \neq 0$ for $h > 0$ and $\phi(h) \neq c \cdot h$ for any constant $c$., introduced in Lemma 1, are satisfied by the DP divergence and the PS score (which is closely related to the $\gamma$-divergence). The following lemma formally verifies this. This fact indicates that constructing $\phi$ so as to satisfy the above condition is one of keys to guarantee robustness.

**Lemma 2.** *Let $\gamma > 0$. The $\phi(h)$ functions for the DP-divergence and the PS-divergence (Remark 1) satisfy the assumptions required in Lemma 1, namely $\phi'(h) \neq 0$ and $\phi(h) - h\phi'(h) \neq 0$ for all $h > 0$.*

*Proof.* **For DP-divergence:** Let $\phi(h) = \gamma - (1 + \gamma)h$.

(i) The derivative is $\phi'(h) = -(1 + \gamma)$. Since $\gamma > 0$, $\phi'(h)$ is a non-zero constant, thus $\phi'(h) \neq 0$ for all $h > 0$.

(ii) We check the second coefficient term from Lemma 1:
$$\phi(h) - h\phi'(h) = \big(\gamma - (1 + \gamma)h\big) - h\big(-(1 + \gamma)\big)$$
$$= \gamma - (1 + \gamma)h + (1 + \gamma)h$$
$$= \gamma$$
Since $\gamma > 0$, we have $\phi(h) - h\phi'(h) \neq 0$.

**For PS score:** Let $\phi(h) = -h^{1+\gamma}$.

(i) The derivative is $\phi'(h) = -(1 + \gamma)h^\gamma$. Since $\gamma > 0$ and we evaluate for $h > 0$, $h^\gamma > 0$. Thus, $\phi'(h)$ is strictly negative, and $\phi'(h) \neq 0$.

(ii) We check the second coefficient term:
$$\phi(h) - h\phi'(h) = \big(-h^{1+\gamma}\big) - h\big(-(1 + \gamma)h^\gamma\big)$$
$$= -h^{1+\gamma} + (1 + \gamma)h^{1+\gamma}$$
$$= (-1 + 1 + \gamma)h^{1+\gamma}$$
$$= \gamma h^{1+\gamma}$$
Since $\gamma > 0$ and $h > 0$, we have $\gamma h^{1+\gamma} > 0$, which implies $\phi(h) - h\phi'(h) \neq 0$.

In both cases, the assumptions hold. This completes the proof. $\square$

Now we show the full proof of Theorem 4.

**Theorem 4** (IF for Hölder-DPO). *Suppose that $\theta^*(\epsilon) = \arg\min_\theta S_\gamma(\tilde{p}_\mathcal{D}^{(\epsilon)} \,\|\, \sigma(g_\theta))$ and $\theta^* = \arg\min_\theta S_\gamma(p_\mathcal{D} \,\|\, \sigma(g_\theta))$. Let $\phi(h)$ be twice-differentiable, and let $0 < \sigma(g_\theta(s))$ and $0 < \gamma < \infty$. Assume that $\phi(h)$ satisfies $\phi'(h) \neq 0$ for $h > 0$ and $\phi(h) \neq c \cdot h$ for any constant $c$[6] and the Hessian $\nabla_\theta^2 \mathcal{L}(s, \pi_\theta)|_{\theta=\theta^*}$ is positive definite. Then, the IF of the Hölder-DPO, excluding terms independent of $s_{\text{flip}}$, is given by:*

$$\text{IF}_{\text{H-DPO}}(s_{\text{flip}}, \theta, p_\mathcal{D}) \propto \mathbb{E}_{p(s_{\text{flip}})}[F_{\theta^*}^{(\gamma)}(s_{\text{flip}})], \tag{10}$$

*where $F_{\theta^*}^{(\gamma)}(s_{\text{flip}}) := \sigma(g_{\theta^*}(s_{\text{flip}}))^\gamma \, \nabla_\theta \mathcal{L}(s_{\text{flip}}, \pi_{\theta^*})$.*

*Proof.* From Proposition 2, the gradient of $S_\gamma(\tilde{p}_\mathcal{D}^{(\epsilon)} \| \sigma(g_\theta))$ w.r.t. $\theta$ is

$$\nabla_\theta S_\gamma(\tilde{p}_\mathcal{D}^{(\epsilon)} \| \sigma(g_\theta)) = \gamma\phi'(\tilde{h}_\theta)\mathbb{E}_{p_\mathcal{D}^{(\epsilon)}}[F_\theta^{(\gamma)}(\tilde{s})] + (1 + \gamma) \cdot \left(\phi(\tilde{h}_\theta) - \tilde{h}_\theta \cdot \phi'(\tilde{h}_\theta)\right) \cdot \left(\int F_\theta^{(1+\gamma)}(\tilde{s})\mathrm{d}\tilde{s}\right),$$

where $F_\theta^{(\gamma)}(\tilde{s}) := \sigma^\gamma(g_\theta(\tilde{s}))\,(1 - \sigma(g_\theta(\tilde{s})))\,\nabla_\theta g_\theta(\tilde{s}) = \sigma^\gamma \nabla_\theta \mathcal{L}(\tilde{s}, \pi_\theta)$, $F_\theta^{(1+\gamma)}(\tilde{s}) := \sigma(g_\theta(\tilde{s})) \cdot F_\theta^{(\gamma)}(\tilde{s})$, $\tilde{h}_\theta := \mathbb{E}_{p_\mathcal{D}^{(\epsilon)}}[\sigma(g_\theta(\tilde{s}))^\gamma]/(\int \sigma(g_\theta(\tilde{s}))^{1+\gamma}\mathrm{d}\tilde{s})$, and $\tilde{s} = \{\tilde{x}, \tilde{y}_{\text{win}}, \tilde{y}_{\text{lose}}\} \sim \tilde{p}_\mathcal{D}^{(\epsilon)}$.

From the definition of $\theta^*(\epsilon)$, we have

$$0 = \nabla_\theta S_\gamma(\tilde{p}_\mathcal{D}^{(\epsilon)} \| \sigma(g_\theta))\Big|_{\theta=\theta^*(\epsilon)}$$

$$= \gamma\phi'(\tilde{h}_{\theta^*(\epsilon)})\mathbb{E}_{\tilde{p}_\mathcal{D}^{(\epsilon)}}[F_{\theta^*(\epsilon)}^{(\gamma)}(\tilde{s})] + (1 + \gamma)\left(\phi(\tilde{h}_{\theta^*(\epsilon)}) - \phi'(\tilde{h}_{\theta^*(\epsilon)})\tilde{h}_{\theta^*(\epsilon)}\right)\left(\int F_{\theta^*(\epsilon)}^{(1+\gamma)}(\tilde{s})\mathrm{d}\tilde{s}\right),$$

---

[6]These assumptions are satisfied by the DP divergence and the PS score, as formally shown in Lemma 2.

where $F_{\theta^*(\epsilon)}^{(\gamma)}(\tilde{s}) := \sigma(g_{\theta^*(\epsilon)}(\tilde{s}))^\gamma \left(1 - \sigma(g_{\theta^*(\epsilon)}(\tilde{s}))\right) \{\nabla_\theta g_\theta(\tilde{s})|_{\theta=\theta^*(\epsilon)}\}$ and $F_{\theta^*(\epsilon)}^{(1+\gamma)}(\tilde{s}) :=$
$\sigma(g_{\theta^*(\epsilon)}(\tilde{s})) \cdot F_{\theta^*(\epsilon)}^{(\gamma)}(\tilde{s})$.

By taking the derivation of this term w.r.t. $\epsilon$ in the above, we obtain

$$
0 = \gamma \left\{ \underbrace{\frac{\partial}{\partial \epsilon} \phi'(\tilde{h}_{\theta^*(\epsilon)}) \mathbb{E}_{\tilde{p}_{\mathcal{D}}^{(\epsilon)}}[F_{\theta^*(\epsilon)}^{(\gamma)}(\tilde{s})]}_{(\mathrm{I})} \right\}
$$
$$
+ (1+\gamma) \left\{ \underbrace{\frac{\partial}{\partial \epsilon} \left( \phi(\tilde{h}_{\theta^*(\epsilon)}) - \phi'(\tilde{h}_{\theta^*(\epsilon)}) \tilde{h}_{\theta^*(\epsilon)} \right) \left( \int F_{\theta^*(\epsilon)}^{(1+\gamma)}(\tilde{s}) \mathrm{d}\tilde{s} \right)}_{(\mathrm{II})} \right\}. \tag{18}
$$

For the term (I), we have

$$
\frac{\partial}{\partial \epsilon} \phi'(h_{\theta^*(\epsilon)}) \mathbb{E}_{p_{\tilde{\mathcal{D}}}^{(\epsilon)}}[F_{\theta^*(\epsilon)}^{(\gamma)}(\tilde{s})]
$$
$$
= \phi''(h_{\theta^*(\epsilon)}) \cdot \frac{\partial h_{\theta^*(\epsilon)}}{\partial \epsilon} \mathbb{E}_{p_{\tilde{\mathcal{D}}}^{(\epsilon)}}[F_{\theta^*(\epsilon)}^{(\gamma)}(\tilde{s})]
$$
$$
+ \phi'(h_{\theta^*(\epsilon)}) \left\{ \int \left\{ \frac{\partial}{\partial \epsilon} p_{\tilde{\mathcal{D}}}^{(\epsilon)}(\tilde{s}) \right\} F_{\theta^*(\epsilon)}^{(\gamma)}(\tilde{s}) \mathrm{d}\tilde{s} + \mathbb{E}_{\tilde{p}_{\mathcal{D}}^{(\epsilon)}} \left[ \frac{\partial}{\partial \epsilon} F_{\theta^*(\epsilon)}^{(\gamma)}(\tilde{s}) \right] \right\}
$$
$$
= \phi''(h_{\theta^*(\epsilon)}) \cdot \frac{\partial h_{\theta^*(\epsilon)}}{\partial \epsilon} \cdot \mathbb{E}_{\tilde{p}_{\mathcal{D}}^{(\epsilon)}}[F_{\theta^*(\epsilon)}^{(\gamma)}(\tilde{s})]
$$
$$
+ \phi'(h_{\theta^*(\epsilon)}) \left\{ \mathbb{E}_{p(s_{\mathrm{flip}})}[F_{\theta^*(\epsilon)}^{(\gamma)}(s_{\mathrm{flip}})] - \mathbb{E}_{p_{\mathcal{D}}}[F_{\theta^*(\epsilon)}^{(\gamma)}(s)] + \mathbb{E}_{\tilde{p}_{\mathcal{D}}^{(\epsilon)}} \left[ \frac{\partial \theta^*(\epsilon)}{\partial \epsilon} \cdot \frac{\partial F_{\theta^*(\epsilon)}^{(\gamma)}(\tilde{s})}{\partial \theta^*(\epsilon)} \right] \right\}
$$
$$
= \phi''(h_{\theta^*(\epsilon)}) \cdot \frac{\partial \theta^*(\epsilon)}{\partial \epsilon} \cdot \frac{\partial h_{\theta^*(\epsilon)}}{\partial \theta^*(\epsilon)} \cdot \mathbb{E}_{\tilde{p}_{\mathcal{D}}^{(\epsilon)}}[F_{\theta^*(\epsilon)}^{(\gamma)}(\tilde{s})]
$$
$$
+ \phi'(h_{\theta^*(\epsilon)}) \left\{ \mathbb{E}_{p(s_{\mathrm{flip}})}[F_{\theta^*(\epsilon)}^{(\gamma)}(s_{\mathrm{flip}})] - \mathbb{E}_{p_{\mathcal{D}}}[F_{\theta^*(\epsilon)}^{(\gamma)}(s)] + \mathbb{E}_{\tilde{p}_{\mathcal{D}}^{(\epsilon)}} \left[ \frac{\partial \theta^*(\epsilon)}{\partial \epsilon} \cdot H_{\theta^*(\epsilon)}^{(\gamma)}(\tilde{s}) \right] \right\}
$$
$$
= \frac{\partial \theta^*(\epsilon)}{\partial \epsilon} \left\{ \phi''(h_{\theta^*(\epsilon)}) \cdot \frac{\partial h_{\theta^*(\epsilon)}}{\partial \theta^*(\epsilon)} \cdot \mathbb{E}_{\tilde{p}_{\mathcal{D}}^{(\epsilon)}}[F_{\theta^*(\epsilon)}^{(\gamma)}(\tilde{s})] + \phi'(h_{\theta^*(\epsilon)}) \cdot \mathbb{E}_{\tilde{p}_{\mathcal{D}}^{(\epsilon)}} \left[ H_{\theta^*(\epsilon)}^{(\gamma)}(\tilde{s}) \right] \right\}
$$
$$
+ \phi'(h_{\theta^*(\epsilon)}) \left\{ \mathbb{E}_{p(s_{\mathrm{flip}})}[F_{\theta^*(\epsilon)}^{(\gamma)}(s_{\mathrm{flip}})] - \mathbb{E}_{p_{\mathcal{D}}}[F_{\theta^*(\epsilon)}^{(\gamma)}(s)] \right\},
$$

where $H_{\theta^*(\epsilon)}^{(\gamma)}(\tilde{s}) := \partial F_{\theta^*(\epsilon)}^{(\gamma)} / \partial \theta^*(\epsilon)$. By taking $\epsilon \to 0$, we have

$$
\frac{\partial}{\partial \epsilon} \phi'(h_{\theta^*(\epsilon)}) \mathbb{E}_{\tilde{p}_{\mathcal{D}}^{(\epsilon)}}[F_{\theta^*(\epsilon)}^{(\gamma)}(\tilde{s})] \bigg|_{\epsilon=0}
$$
$$
= \frac{\partial \theta^*(\epsilon)}{\partial \epsilon} \bigg|_{\epsilon=0} \cdot \left\{ \phi''(h_{\theta^*}) \cdot \frac{\partial h_{\theta^*}}{\partial \theta^*} \cdot \mathbb{E}_{p_{\mathcal{D}}}[F_{\theta^*}^{(\gamma)}(s)] + \phi'(h_{\theta^*}) \cdot \mathbb{E}_{p_{\mathcal{D}}} \left[ H_{\theta^*}^{(\gamma)}(s) \right] \right\}
$$
$$
+ \phi'(h_{\theta^*}) \left\{ \mathbb{E}_{p(s_{\mathrm{flip}})}[F_{\theta^*(\epsilon)}^{(\gamma)}(s_{\mathrm{flip}})] - \mathbb{E}_{p_{\mathcal{D}}}[F_{\theta^*}^{(\gamma)}(s)] \right\}. \tag{19}
$$

For the term (II), we obtain

$$\frac{\partial}{\partial\epsilon}\left(\phi(h_{\theta^*(\epsilon)})-\phi'(h_{\theta^*(\epsilon)})h_{\theta^*(\epsilon)}\right)\left(\int F_{\theta^*(\epsilon)}^{(1+\gamma)}(\tilde{s})\mathrm{d}\tilde{s}\right)$$

$$=\left\{\frac{\partial}{\partial\epsilon}\left(\phi(h_{\theta^*(\epsilon)})-\phi'(h_{\theta^*(\epsilon)})h_{\theta^*(\epsilon)}\right)\right\}\left(\int F_{\theta^*(\epsilon)}^{(1+\gamma)}(\tilde{s})\mathrm{d}\tilde{s}\right)$$

$$+\left(\phi(h_{\theta^*(\epsilon)})-\phi'(h_{\theta^*(\epsilon)})h_{\theta^*(\epsilon)}\right)\left(\int\frac{\partial}{\partial\epsilon}F_{\theta^*(\epsilon)}^{(1+\gamma)}(\tilde{s})\mathrm{d}\tilde{s}\right)$$

$$=\left(-\phi''(h_{\theta^*(\epsilon)})\cdot\frac{\partial h_{\theta^*(\epsilon)}}{\partial\epsilon}\cdot h_{\theta^*(\epsilon)}\right)\left(\int F_{\theta^*(\epsilon)}^{(1+\gamma)}(\tilde{s})\mathrm{d}\tilde{s}\right)$$

$$+\left(\phi(h_{\theta^*(\epsilon)})-\phi'(h_{\theta^*(\epsilon)})h_{\theta^*(\epsilon)}\right)\left(\int\frac{\partial\theta^*(\epsilon)}{\partial\epsilon}H_{\theta^*(\epsilon)}^{(1+\gamma)}(\tilde{s})\mathrm{d}\tilde{s}\right)$$

$$=\frac{\partial\theta^*(\epsilon)}{\partial\epsilon}\left\{\left(-\phi''(h_{\theta^*(\epsilon)})\cdot\frac{\partial h_{\theta^*(\epsilon)}}{\partial\theta^*(\epsilon)}\cdot h_{\theta^*(\epsilon)}\right)\left(\int F_{\theta^*(\epsilon)}^{(1+\gamma)}(\tilde{s})\mathrm{d}\tilde{s}\right)\right.$$

$$\left.+\left(\phi(h_{\theta^*(\epsilon)})-\phi'(h_{\theta^*(\epsilon)})h_{\theta^*(\epsilon)}\right)\left(\int H_{\theta^*(\epsilon)}^{(1+\gamma)}(\tilde{s})\mathrm{d}\tilde{s}\right)\right\},$$

where we use the dominated convergence theorem to exchange the integral and derivative in the second equality and $H_{\theta^*(\epsilon)}^{(1+\gamma)}(\tilde{s}):=\partial F_{\theta^*(\epsilon)}^{(1+\gamma)}/\partial\theta^*(\epsilon)$. By taking $\epsilon\to0$, we obtain

$$\frac{\partial}{\partial\epsilon}\left(\phi(h_{\theta^*(\epsilon)})-\phi'(h_{\theta^*(\epsilon)})h_{\theta^*(\epsilon)}\right)\left(\int F_{\theta^*(\epsilon)}^{(1+\gamma)}(\tilde{s})\mathrm{d}\tilde{s}\right)\Bigg|_{\epsilon=0}$$

$$=\frac{\partial\theta^*(\epsilon)}{\partial\epsilon}\Bigg|_{\epsilon=0}\cdot\left\{\left(-\phi''(h_{\theta^*})\cdot\frac{\partial h_{\theta^*}}{\partial\theta^*}\cdot h_{\theta^*}\right)\left(\int F_{\theta^*}^{(1+\gamma)}(s)\mathrm{d}s\right)\right.$$

$$\left.+\left(\phi(h_{\theta^*})-\phi'(h_{\theta^*})h_{\theta^*}\right)\left(\int H_{\theta^*}^{(1+\gamma)}(s)\mathrm{d}s\right)\right\}. \tag{20}$$

Substituting Eqs. (19) and (20) into Eq. (18) gives us:

$$-\gamma\phi'(h_{\theta^*})\left\{\mathbb{E}_{p(s_{\text{flip}})}[F_{\theta^*}^{(\gamma)}(s_{\text{flip}})]-\mathbb{E}_{p_{\mathcal{D}}}[F_{\theta^*}^{(\gamma)}(s)]\right\}$$

$$=\frac{\partial\theta^*(\epsilon)}{\partial\epsilon}\Bigg|_{\epsilon=0}\cdot\left\{\gamma\phi''(h_{\theta^*})\cdot\frac{\partial h_{\theta^*}}{\partial\theta^*}\cdot\mathbb{E}_{p_{\mathcal{D}}}[F_{\theta^*}^{(\gamma)}(s)]+\gamma\phi'(h_{\theta^*})\cdot\mathbb{E}_{p_{\mathcal{D}}}\left[H_{\theta^*}^{(\gamma)}(s)\right]\right.$$

$$+(1+\gamma)\left(-\phi''(h_{\theta^*})\cdot\frac{\partial h_{\theta^*}}{\partial\theta^*}\cdot h_{\theta^*}\right)\left(\int F_{\theta^*}^{(1+\gamma)}(s)\mathrm{d}s\right)$$

$$\left.+(1+\gamma)\left(\phi(h_{\theta^*})-\phi'(h_{\theta^*})h_{\theta^*}\right)\left(\int H_{\theta^*}^{(1+\gamma)}(s)\mathrm{d}s\right)\right\}.$$

From Lemma 1, we further obtain

$$-\gamma\phi'(h_{\theta^*})\mathbb{E}_{p(s_{\text{flip}})}[F_{\theta^*}^{(\gamma)}(s_{\text{flip}})]=\frac{\partial\theta^*(\epsilon)}{\partial\epsilon}\Bigg|_{\epsilon=0}\cdot C_{\theta^*}^{(\gamma)}(s),$$

where $C_{\theta^*}^{(\gamma)}(s)$ is the constant term w.r.t. the contaminated input $s_{\text{flip}}$ defined as

$$C_{\theta^*}^{(\gamma)}(s):=\gamma\phi'(h_{\theta^*})\cdot\mathbb{E}_{p_{\mathcal{D}}}\left[H_{\theta^*}^{(\gamma)}(s)\right]+(1+\gamma)\left(\phi(h_{\theta^*})-\phi'(h_{\theta^*})h_{\theta^*}\right)\left(\int H_{\theta^*}^{(1+\gamma)}(s)\mathrm{d}s\right).$$

We finally obtain

$$\frac{\partial\theta^*(\epsilon)}{\partial\epsilon}\Bigg|_{\epsilon=0}=-\left(C_{\theta^*}^{(\gamma)}(s)\right)^{-1}\cdot\gamma\phi'(h_{\theta^*})\mathbb{E}_{p(s_{\text{flip}})}[F_{\theta^*}^{(\gamma)}(s_{\text{flip}})]. \tag{21}$$

From the assumption, $(C_{\theta^*}^{(\gamma)}(s))^{-1}$ exists because $H_{\theta^*}^{(\gamma)}(s)$ is positive definite from the fact that the gradient of Hölder-DPO is $\sigma(g_{\theta^*}(s_{\text{flip}}))^\gamma\nabla_\theta\mathcal{L}(s_{\text{flip}},\pi_{\theta^*})$, where $\nabla_\theta\mathcal{L}(s_{\text{flip}},\pi_{\theta^*})$ is the gradient of DPO loss whose Hessian is assumed as positive definite. This completes the proof. $\qquad\square$

### E.4 Proof for Corollary 1

**Corollary 1** (**Hölder-DPO is robust**). *Suppose that the policy gradient $\nabla_\theta \log \pi_\theta(y \mid x)$ is bounded by $C$ and satisfies $L$-Lipchitz in $\theta$, where $0 < C < \infty$ and $0 < L < \infty$. Then, under Theorem 4, the IF of Hölder-DPO satisfies the redescending property in Definition 2.*

*Proof.* From Theorem 4, the IF for Hölder-DPO is given by:

$$\text{IF}_{\text{H-DPO}} = -\left( C_{\theta^*}^{(\gamma)}(s) \right)^{-1} \cdot \gamma \phi'(h_{\theta^*}) \mathbb{E}_{p(s_{\text{flip}})}[F_{\theta^*}^{(\gamma)}(s_{\text{flip}})].$$

First, we analyze the denominator $C_{\theta^*}^{(\gamma)}(s)$. Following Theorem 5, we can see that the Hessian-related term $C_{\theta^*}^{(\gamma)}(s)$ is positive definite. Let $L' = \lambda_{\min}(C_{\theta^*}^{(\gamma)}(s)) > 0$ be its minimum eigenvalue. Then the norm of its inverse is bounded: $\|(C_{\theta^*}^{(\gamma)}(s))^{-1}\| \le 1/L'$.

Next, we analyze the numerator, $\mathbb{E}_{p(s_{\text{flip}})}[F_{\theta^*}^{(\gamma)}(s_{\text{flip}})]$, where $F_{\theta^*}^{(\gamma)}(s_{\text{flip}}) := \sigma(g_{\theta^*}(s_{\text{flip}}))^\gamma(1 - \sigma(g_{\theta^*}(s_{\text{flip}})))\nabla_\theta g_{\theta^*}(s_{\text{flip}})$. From the assumption that $\|\nabla_\theta \log \pi_\theta\| \le C$, the term $\|\nabla_\theta g_{\theta^*}\|$ is also bounded by $C' = 2\beta C$.

We now take the limit required by Definition 2:

$$\lim_{\hat{r}_{\theta^*}(x,y_{\text{lose}}^{\text{flip}})\to\infty} \|\text{IF}_{\text{H-DPO}}\|$$

$$\le \lim_{\hat{r}_{\theta^*}(x,y_{\text{lose}}^{\text{flip}})\to\infty} \left\| \left( C_{\theta^*}^{(\gamma)}(s) \right)^{-1} \cdot \gamma\phi'(h_{\theta^*}) \right\| \cdot \lim_{\hat{r}_{\theta^*}(x,y_{\text{lose}}^{\text{flip}})\to\infty} \left\| \mathbb{E}_{p(s_{\text{flip}})}[F_{\theta^*}^{(\gamma)}(s_{\text{flip}})] \right\|$$

$$\le K \cdot \lim_{\hat{r}_{\theta^*}(x,y_{\text{lose}}^{\text{flip}})\to\infty} \mathbb{E}_{p(s_{\text{flip}})} \left[ \underbrace{\sigma(g_{\theta^*}(s_{\text{flip}}))^\gamma}_{\to 0} \cdot \underbrace{(1 - \sigma(g_{\theta^*}(s_{\text{flip}})))}_{\to 1} \cdot \underbrace{\|\nabla_\theta g_{\theta^*}\|}_{\le C'} \right],$$

where $K = (1/L') \cdot |\gamma\phi'(h_{\theta^*})|$ is a finite non-zero constant.

By the bounded convergence theorem, we can move the limit inside the expectation:

$$\lim_{\hat{r}_{\theta^*}(x,y_{\text{lose}}^{\text{flip}})\to\infty} \|\text{IF}_{\text{H-DPO}}\| \le K \cdot \mathbb{E}_{p(s_{\text{flip}})} \left[ \lim_{\hat{r}_{\theta^*}(x,y_{\text{lose}}^{\text{flip}})\to\infty} \left( \sigma(g_{\theta^*})^\gamma \cdot (1 - \sigma(g_{\theta^*})) \cdot \|\nabla_\theta g_{\theta^*}\| \right) \right]$$

$$\le K \cdot \mathbb{E}_{p(s_{\text{flip}})} \left[ 0 \cdot 1 \cdot C' \right]$$

$$= 0.$$

The IF of Hölder-DPO converges to $0$, satisfying the redescending property. This completes the proof. $\qquad\square$

## F Contamination ratio estimation and outlier detection

Estimating the contamination ratio $\epsilon$ and identifying the contamination data based on this are crucial for achieving appropriate model alignment. In this section, we show that our Hölder-DPO can be extended to incorporate Enlarged models, enabling these objectives to be realized.

### F.1 Model extension approach to both parameter estimation and contamination rate estimation

Here, we reorganize the framework using *model extension* proposed by Kanamori and Fujisawa [54] for simultaneously estimating model parameters and contamination rates in the context of the DPO.

According to Eq. (6), the essence of DPO-based approaches, including our Hölder-DPO, lies in minimizing a divergence $D$ between $p_{\mathcal{D}}(s)$ and $\sigma(g_\theta(s))$ with respect to $\theta$, i.e., estimating $p_{\mathcal{D}}(s)$ through the model parameters $\theta$. When $p_{\mathcal{D}}(s)$ is contaminated as defined in Definition 1, this optimization problem can be reformulated as:

$$\theta^*(\epsilon) = \underset{\theta}{\arg\min}\, D[\tilde{p}_{\mathcal{D}}^{(\epsilon)}\|\sigma(g_\theta(s))] = \underset{\theta}{\arg\min}\, D[(1-\epsilon)p_{\mathcal{D}}(s) + \epsilon\delta_{s_{\text{flip}}}\|\sigma(g_\theta(s))].$$

If a divergence that automatically mitigates the impact of $\delta_{s_\text{flip}}$ is chosen, the optimization reduces to:

$$\theta^*(\epsilon) = \operatorname*{argmin}_\theta D[\tilde{p}_\mathcal{D}^{(\epsilon)} \| \sigma(g_\theta(s))] \approx \operatorname*{argmin}_\theta D[(1-\epsilon)p_\mathcal{D}(s) \| \sigma(g_\theta(s))],$$

indicating that $\sigma(g_\theta(s))$ estimates $(1-\epsilon)p_\mathcal{D}(s)$, rather than the original target $p_\mathcal{D}(s)$.

To address this gap, we consider the following extended model: $m_\eta(s) = \xi\sigma(g_\theta(s))$, where $\eta = (\xi, \theta)$. In this case, the DPO-based model alignment under the robust divergence can be formulated as:

$$\theta^*(\epsilon) = \operatorname*{argmin}_\theta \min_\xi D[\tilde{p}_\mathcal{D}^{(\epsilon)} \| m_\eta(s)] \approx \operatorname*{argmin}_\theta \min_\xi D[(1-\epsilon)p_\mathcal{D}(s) \| \xi\sigma(g_\theta(s))].$$

With this formulation, $\xi$ is expected to play the role of determining the ratio of clean data $1-\epsilon$ and $\sigma(g_\theta(s))$ is expected to serve as an estimator of the clean data distribution $p_\mathcal{D}(s)$.

## F.2 Contamination ratio estimation

### F.2.1 Model extension to estimate $\theta$ and $\epsilon$:

Recall from Eq. (6) that DPO-based methods, including our Hölder-DPO, estimate the preference data distribution $p_\mathcal{D}$ by minimizing $D[p_\mathcal{D} \| \sigma(g_\theta)]$, where $D$ denotes a *generic* divergence measuring the discrepancy between $p_\mathcal{D}$ and the model output $\sigma(g_\theta)$. When the data is contaminated as $p_{\tilde{\mathcal{D}}}^{(\epsilon)}(\tilde{s}) = (1-\epsilon)p_\mathcal{D}(s) + \epsilon p(s_\text{flip})$, the objective becomes minimizing $D[p_{\tilde{\mathcal{D}}}^{(\epsilon)} \| \sigma(g_\theta)]$. If $D$ approximately nullifies the contribution of the contamination term $p(s_\text{flip})$, i.e., $D[p(s_\text{flip}) \| \sigma(g_\theta(s_\text{flip}))] \approx 0$ [7], then minimizing $D[p_{\tilde{\mathcal{D}}}^{(\epsilon)} \| \sigma(g_\theta)]$ aligns $\sigma(g_\theta)$ with $(1-\epsilon)p_\mathcal{D}$. This results in a *mismatch in scale* relative to $p_\mathcal{D}$ (see Appendix F.1 for details). To correct for this mismatch, we *extend the model* to $m_\eta = \xi \cdot \sigma(g_\theta)$ with $\eta = (\xi, \theta)$, introducing a scaling parameter $\xi > 0$ to explicitly account for the *clean-data proportion* $(1-\epsilon)$. The revised objective $\operatorname{argmin}_\eta D[p_{\tilde{\mathcal{D}}}^{(\epsilon)} \| m_\eta]$ enables $\sigma(g_\theta)$ to serve as an estimator of $p_\mathcal{D}(s)$, while $\xi$ absorbs the $(1-\epsilon)$ scaling.

### F.2.2 Extended model for estimating the contamination ratio

In this section, we discuss the case when we conduct our Hölder-DPO under the extended model $m_\eta$. According to the discussion in Section 4.1, the optimization problem of Hölder-DPO with $m_\eta$ can be formulated as

$$\operatorname*{argmin}_\theta \min_\xi D_H[\tilde{p}_\mathcal{D}^{(\epsilon)} \| m_\eta(\tilde{s})] = \operatorname*{argmin}_\theta \min_\xi S_\gamma(\tilde{p}_\mathcal{D}^{(\epsilon)} \| m_\eta(\tilde{s})). \tag{22}$$

Recalling Definition 3 and the discrete probabilistic nature of $\sigma$, we can see that

$$
\begin{aligned}
S_\gamma(\tilde{p}_\mathcal{D}^{(\epsilon)} \| m_\eta(\tilde{s})) &= \phi\left(\frac{\mathbb{E}_{\tilde{p}_\mathcal{D}^{(\epsilon)}}[m_\eta^\gamma(\tilde{s})]}{\int m_\eta^{1+\gamma}(\tilde{s})\mathrm{d}\tilde{s}}\right)\left(\int m_\eta^{1+\gamma}(\tilde{s})\mathrm{d}\tilde{s}\right) \\
&= \phi\left(\frac{\mathbb{E}_{\tilde{p}_\mathcal{D}^{(\epsilon)}}[\sigma(g_\theta(\tilde{s}))^\gamma]}{\xi \cdot \int \sigma(g_\theta(\tilde{s}))^{1+\gamma}\mathrm{d}\tilde{s}}\right)\left(\xi^{1+\gamma} \cdot \int \sigma(g_\theta(\tilde{s}))^{1+\gamma}\mathrm{d}\tilde{s}\right) \\
&\geq -\left(\frac{\mathbb{E}_{\tilde{p}_\mathcal{D}^{(\epsilon)}}[\sigma(g_\theta(\tilde{s}))^\gamma]}{\xi \cdot \int \sigma(g_\theta(\tilde{s}))^{1+\gamma}\mathrm{d}\tilde{s}}\right)^{1+\gamma}\left(\xi^{1+\gamma} \cdot \int \sigma(g_\theta(\tilde{s}))^{1+\gamma}\mathrm{d}\tilde{s}\right) \\
&= -\frac{\mathbb{E}_{\tilde{p}_\mathcal{D}^{(\epsilon)}}[\sigma(g_\theta(\tilde{s}))^\gamma]}{\underbrace{\left(\int \sigma(g_\theta(\tilde{s}))^{1+\gamma}\mathrm{d}\tilde{s}\right)^{\frac{\gamma}{1+\gamma}}}_{=:S_\text{PS}(\tilde{p}_\mathcal{D}^{(\epsilon)} \| \sigma(g_\theta(\tilde{s})))}} \\
&= S_\text{PS}(\tilde{p}_\mathcal{D}^{(\epsilon)} \| \sigma(g_\theta(\tilde{s}))) = -\exp\left\{-\gamma(1+\gamma) \cdot S_\text{log}(\tilde{p}_\mathcal{D}^{(\epsilon)} \| \sigma(g_\theta(\tilde{s})))\right\}, \tag{23}
\end{aligned}
$$

---

[7] DP- and $\gamma$-divergences satisfy this property under Assumption 1 (see Appendix F.4).

where the third inequality comes from the fact that $\phi(h) \geq -h^{1+\gamma}$ for all $h \geq 0$, the term $S_{\mathrm{PS}}(\tilde{p}_{\mathcal{D}}^{(\epsilon)} \| \sigma(g_\theta(\tilde{s})))$ in the forth line is called as the pseudo-spherical (PS) score, and $S_{\log}(\tilde{p}_{\mathcal{D}}^{(\epsilon)} \| \sigma(g_\theta(\tilde{s})))$ is the $\gamma$-score associated with $\gamma$-divergence defined as

$$S_{\log}(\tilde{p}_{\mathcal{D}}^{(\epsilon)} \| \sigma(g_\theta(\tilde{s}))) := -\frac{1}{\gamma} \log \left( \mathbb{E}_{\tilde{p}_{\mathcal{D}}^{(\epsilon)}}[\sigma(g_\theta(\tilde{s}))^\gamma] \right) + \frac{1}{1+\gamma} \log \int \sigma(g_\theta(\tilde{s}))^{1+\gamma} \mathrm{d}\tilde{s}. \tag{24}$$

From the Eq. (23), we can see that the lower bound of $S_\gamma(\tilde{p}_{\mathcal{D}}^{(\epsilon)} \| m_\eta(\tilde{s}))$ is independent of $\xi$. Therefore, the optimal solution of $\xi^* = \min_\xi S_\gamma(\tilde{p}_{\mathcal{D}}^{(\epsilon)} \| m_\eta(\tilde{s}))$ is obtained as the equality condition of Eq. (23), that is,

$$\xi^* = \frac{\mathbb{E}_{\tilde{p}_{\mathcal{D}}^{(\epsilon)}}[\sigma(g_\theta(\tilde{s}))^\gamma]}{\int \sigma(g_\theta(\tilde{s}))^{1+\gamma} \mathrm{d}\tilde{s}} \quad (\forall \theta), \tag{25}$$

where we used the fact that $\phi(1) = -1$.

At the end of this section, we verify that $\xi^*$ in Eq. (25) serves as an estimate of the contamination rate. When robust model alignment using Hölder-DPO effectively reduce the effects of contamination and the value of $\sigma_\theta(g_\theta(\tilde{s}))$ closely approximates the target distribution $p_{\mathcal{D}}(s)$, i.e., $\sigma_\theta(g_\theta(\tilde{s})) \approx p_{\mathcal{D}}(s)$, we have:

$$\xi^* \approx \frac{\mathbb{E}_{\tilde{p}_{\mathcal{D}}^{(\epsilon)}}[p_{\mathcal{D}}(s)^\gamma]}{\mathbb{E}_{p_{\mathcal{D}}}[p_{\mathcal{D}}(s)^\gamma]} = \frac{(1-\epsilon)\mathbb{E}_{p_{\mathcal{D}}}[p_{\mathcal{D}}(s)^\gamma] + \epsilon \mathbb{E}_{p(s_{\mathrm{flip}})}[p_{\mathcal{D}}(s_{\mathrm{flip}})^\gamma]}{\mathbb{E}_{p_{\mathcal{D}}}[p_{\mathcal{D}}(s)^\gamma]}$$

$$= (1-\epsilon) + \frac{\epsilon}{\mathbb{E}_{p_{\mathcal{D}}}[p_{\mathcal{D}}(s)^\gamma]} \cdot \mathbb{E}_{p(s_{\mathrm{flip}})}[p_{\mathcal{D}}(s_{\mathrm{flip}})^\gamma].$$

When the contamination distribution $\delta_{s_{\mathrm{flip}}}$ is located at the tail of the target distribution $p_D(s)$, meaning that the probability of a sample drawn from $\delta_{s_{\mathrm{flip}}}$ under $p_{\mathcal{D}}(s)$ is sufficiently small, we can see $\mathbb{E}_{\delta_z(s)}[p_{\mathcal{D}}(s)^\gamma] \approx 0$. Then, we have $\xi^* \approx (1-\epsilon)$ and thus the contamination rate can be estimated by $\min\{0, 1 - \xi^*\}$.

### F.3 Estimator of $\xi^*$

Since the exact form of $p_{\mathcal{D}}^{(\epsilon)}$ is unknown and computing the integral with respect to $\sigma(g_\theta)$ in Eq. (22) is intractable, following the same strategy in our objective, we estimate $\xi$ empirically as:

$$\hat{\xi} = \frac{\frac{1}{N} \sum_{i=1}^N \sigma(g_\theta(\tilde{s}^{(i)}))^\gamma}{\frac{1}{N} \sum_{i=1}^N \sigma(g_\theta(\tilde{s}^{(i)}))^{1+\gamma}} = \frac{\sum_{i=1}^N \sigma(g_\theta(\tilde{s}^{(i)}))^\gamma}{\sum_{i=1}^N \sigma(g_\theta(\tilde{s}^{(i)}))^{1+\gamma}}, \tag{26}$$

where the final expression is derived via empirical approximation.

While $\xi^*$ can be estimated via Eq. (26), the resulting estimator $\hat{\xi}^*$ is not necessarily guaranteed to lie within the valid range $[0, 1]$. In fact, the following lemma demonstrates that the estimator $\hat{\xi}^*$ is not operate properly within the valid interval $[0, 1]$.

**Lemma 3.** *Suppose that $0 < \sigma(g_\theta(\tilde{s}^{(i)})) \leq 1$ for all $i = 1, \ldots, N$. Let $0 < \gamma < \infty$. Then, the estimator $\hat{\xi}^*$ defined in Eq. (26) satisfies $\hat{\xi}^* > 1$.*

*Proof.* If $\hat{\xi}^* \leq 1$, we have

$$\sum_{i=1}^N \sigma(g_\theta(\tilde{s}^{(i)}))^\gamma \leq \sum_{i=1}^N \sigma(g_\theta(\tilde{s}^{(i)}))^{1+\gamma}.$$

However, since $0 < \sigma(g_\theta(\tilde{s}^{(i)})) \leq 1$ for all $i$, we have

$$\sum_{i=1}^N \sigma(g_\theta(\tilde{s}^{(i)}))^\gamma \geq \sum_{i=1}^N \sigma(g_\theta(\tilde{s}^{(i)}))^{1+\gamma} \Rightarrow \hat{\xi} \geq 1,$$

which is contradicted by the above condition. This completes the proof. $\qquad \square$

One possible approach to mitigate this issue is to introduce a scaling parameter $v$ and define $\bar{\xi} := v\xi$ as the scaling term in the extended model $m_\eta$, where $\eta = (\xi, \theta)$. Note that $v$ is not optimized; rather, it serves solely as a hyperparameter for scaling the estimated rate $\xi^*$. In this case, by following the same discussion on Section F.2.2, we obtain, for any $v$ and $\theta$,

$$\hat{\xi}^* = v^{-1} \cdot \frac{\sum_{i=1}^N \sigma(g_\theta(\tilde{s}^{(i)}))^\gamma}{\sum_{i=1}^N \sigma(g_\theta(\tilde{s}^{(i)}))^{1+\gamma}}.$$

Since the above expression holds for any $v$, one can select $v$ to ensure that $\hat{\xi} \in [0, 1]$. However, manually tuning $v$ introduces arbitrariness and risks biasing the contamination estimate. Given that the original $\hat{\xi}$ is derived from model likelihoods $\sigma(g_\theta(\tilde{s}^{(i)}))$, it is desirable for the scaling factor to be informed by model-based quantities. The following lemma shows that scaling by the mean model likelihood yields a principled correction to $\hat{\xi}$.

**Lemma 4.** *Suppose that $0 < \sigma(g_\theta(\tilde{s}^{(i)})) \leq 1$ for all $i = 1, \ldots, N$. Let $0 < \gamma < \infty$. Then, we have $0 < v \cdot \hat{\xi} \leq 1$ when we set $v^{-1} = \frac{1}{N}\sum_{i=1}^N \sigma(g_\theta(\tilde{s}^{(i)}))$.*

*Proof.* Because of $0 < \sigma(g_\theta(\tilde{s}^{(i)})) \leq 1$ for all $i$, we have $v \cdot \hat{\xi} > 0$. We can reorganize $\{\sigma(g_\theta(\tilde{s}^{(i)}))\}_{i=1}^N$ and $\{\sigma(g_\theta(\tilde{s}^{(i)}))^\gamma\}_{i=1}^N$ so as to be the similarly-ordered-sequences (both are increasing functions of $\sigma(g_\theta(\tilde{s}^{(i)}))$). Then, for two similarly ordered, non-negative sequences $\{\sigma(g_\theta(\tilde{s}^{(i)}))\}_{i=1}^N$ and $\{\sigma(g_\theta(\tilde{s}^{(i)}))^\gamma\}_{i=1}^N$, we obtain

$$\frac{1}{N}\sum_{i=1}^N \sigma(g_\theta(\tilde{s}^{(i)}))^{1+\gamma} = \frac{1}{N}\sum_{i=1}^N \sigma(g_\theta(\tilde{s}^{(i)})) \cdot \sigma(g_\theta(\tilde{s}^{(i)}))^\gamma$$

$$\geq \left(\frac{1}{N}\sum_{i=1}^N \sigma(g_\theta(\tilde{s}^{(i)}))\right)\left(\frac{1}{N}\sum_{i=1}^N \sigma(g_\theta(\tilde{s}^{(i)}))^\gamma\right),$$

where the final inequality comes from Chebyshev's sum inequality. Dividing both sides by $\frac{1}{N}\sum_{i=1}^N \sigma(g_\theta(\tilde{s}^{(i)}))^{1+\gamma}$ completes the proof. $\square$

From this lemma, we adopt the following estimator for the clean data proportion:

$$\hat{\xi}^* = \left(\frac{1}{N}\sum_{i=1}^N \sigma(g_\theta(\tilde{s}^{(i)}))\right) \cdot \frac{\sum_{i=1}^N \sigma(g_\theta(\tilde{s}^{(i)}))^\gamma}{\sum_{i=1}^N \sigma(g_\theta(\tilde{s}^{(i)}))^{1+\gamma}} = \frac{\frac{1}{N}\sum_{i=1}^N \bar{\sigma}(g_\theta(\tilde{s}^{(i)}))^\gamma}{\sum_{i=1}^N \bar{\sigma}(g_\theta(\tilde{s}^{(i)}))^{1+\gamma}}, \qquad (27)$$

where the normalized likelihood is defined as

$$\bar{\sigma}(g_\theta(\tilde{s}^{(i)})) := \frac{\sigma(g_\theta(\tilde{s}^{(i)}))}{\sum_{i=1}^N \sigma(g_\theta(\tilde{s}^{(i)}))}.$$

Recalling that $\hat{\xi}$ estimates the *clean* data ratio, Eq. (27) behaves as desired. Consider a simple case where the model $\sigma(g_\theta)$ perfectly distinguishes between clean and contaminated data, i.e., $\sigma(g_\theta(s_{\text{flip}})) = 1$ for clean samples and $\sigma(g_\theta(s_{\text{flip}})) = 0$ for flipped (noisy) ones. Suppose that $M$ out of $N$ total samples are contaminated. Then:

$$\frac{\sum_{i=1}^N \sigma(g_\theta(\tilde{s}^{(i)}))^\gamma}{\sum_{i=1}^N \sigma(g_\theta(\tilde{s}^{(i)}))^{1+\gamma}} = \frac{N-M}{N-M} = 1, \quad \frac{1}{N}\sum_{i=1}^N \sigma(g_\theta(\tilde{s}^{(i)})) = \frac{N-M}{N}.$$

Multiplying these two terms yields $\hat{\xi}^* = \frac{N-M}{N}$, which exactly recovers the true clean-data proportion.

## F.4 Choice of $\phi$

To implement Hölder-DPO in practice, one must specify a concrete choice of the function $\phi$. For robust LM alignment, a natural choice is the DP divergence with $\phi(h) = \gamma - (1+\gamma)h$, or alternatively, the PS score (equivalently, the $\gamma$-score) with $\phi(h) = -h^{1+\gamma}$. The following lemma shows that, when the goal is to achieve *both robustness and contamination ratio estimation* simultaneously, the DP divergence is the preferable choice among these two options.

**Lemma 5.** *Under Assumption 1 and Definition 1. Then, under the extended model $m_\eta$, the objective $S_\gamma(p_{\tilde{\mathcal{D}}}^{(\epsilon)}\|m_\eta(\tilde{s}))$ can be approximated around $\theta = \theta^*$ as*

$$S_\gamma(p_{\tilde{\mathcal{D}}}^{(\epsilon)}\|m_\eta(\tilde{s})) \approx \begin{cases} (1-\epsilon)S_{\mathrm{PS}}(p_{\mathcal{D}}(s)\|\sigma(g_\theta(s))) & \text{if } \phi(h) = -h^{1+\gamma}, \\ S_{\mathrm{DP}}((1-\epsilon)p_{\mathcal{D}}\|m_\eta(s)) & \text{if } \phi(h) = \gamma - (1+\gamma)h, \end{cases}$$

*where $S_{\mathrm{DP}}$ and $S_{\mathrm{PS}}$ denote the DP divergence and the PS score, respectively, as defined in Appendix F.4.*

Under Lemma 5, the DPO objective based on the PS score reduces to $(1-\epsilon)\,S_{\mathrm{PS}}(p_{\mathcal{D}}\|\sigma(g_\theta))$ in a neighborhood of $\theta^*$. Consequently, provided that the optimized parameter $\theta$ lies within this neighborhood, $\min_\eta S_\gamma(p_{\tilde{\mathcal{D}}}^{(\epsilon)}\|m_\eta)$ recovers $\min_\theta S_{\mathrm{PS}}(p_{\mathcal{D}}\|\sigma(g_\theta))$, independently of the contamination ratio $\epsilon$. Because the scale parameter $\xi$ disappears from this reduced objective, the contamination proportion cannot be identified in the PS-score variant. By contrast, with the DP-divergence-based DPO, the reduced objective $S_{\mathrm{DP}}((1-\epsilon)p_{\mathcal{D}}\|m_\eta)$ retains $\xi$, enabling the optimization to *jointly recover* both the target parameter $\theta^*$ and the clean-data proportion $1-\epsilon$. In summary, once a robust solution near $\theta^*$ is obtained, the optimized model $\sigma(g_\theta)$ closely approximates $p_{\mathcal{D}}$, while the scaling parameter $\xi$ serves as an accurate estimator of $1-\epsilon$. For further details, see Appendix F.4. Whether this property holds in practice depends on the ability to estimate $\theta$ robustly under $p_{\tilde{\mathcal{D}}}^{(\epsilon)}$; hence, the theoretical robustness guarantee of Hölder-DPO presented in Section 4.1 plays a crucial role.

When applying Hölder-DPO, it is necessary to select an appropriate function $\phi$. For the purpose of performing robust DPO, one may consider using the DP divergence with $\phi(h) = \gamma - (1+\gamma)h$, or the $\gamma$-score obtained by setting $\phi(h) = -h^{1+\gamma}$. However, if the goal is to simultaneously *achieve robustness and estimate the contamination ratio*, the following discussion shows that using the DP divergence is preferable.

When we set $\phi(h) = -h^{1+\gamma}$, the Hölder-DPO objective function can be decomposed as:

$$\begin{aligned}
S_\gamma(\tilde{p}_{\mathcal{D}}^{(\epsilon)}\|m_\eta(\tilde{s})) &= \underbrace{-\frac{\mathbb{E}_{p_{\tilde{\mathcal{D}}}^{(\epsilon)}}[m_\eta(\tilde{s})^\gamma]}{\left(\mathbb{E}_{m_\eta}[m_\eta(\tilde{s})^\gamma]\right)^{\frac{\gamma}{1+\gamma}}}}_{=:S_{\mathrm{PS}}(\tilde{p}_{\mathcal{D}}^{(\epsilon)}\|m_\eta(\tilde{s}))} \\
&= -(1-\epsilon)\frac{\mathbb{E}_{p_{\mathcal{D}}}[m_\eta(s)^\gamma]}{\left(\mathbb{E}_{m_\eta}[m_\eta(s)^\gamma]\right)^{\frac{\gamma}{1+\gamma}}} - \epsilon\frac{\mathbb{E}_{p(s_{\mathrm{flip}})}[m_\eta(s_{\mathrm{flip}})^\gamma]}{\left(\mathbb{E}_{m_\eta}[m_\eta(s_{\mathrm{flip}})^\gamma]\right)^{\frac{\gamma}{1+\gamma}}} \\
&= -(1-\epsilon)\frac{\mathbb{E}_{p_{\mathcal{D}}}[\sigma(g_\theta(s))^\gamma]}{\left(\mathbb{E}_{\sigma(g_\theta)}[\sigma(g_\theta(s))^\gamma]\right)^{\frac{\gamma}{1+\gamma}}} - \epsilon\frac{\mathbb{E}_{p(s_{\mathrm{flip}})}[\sigma(g_\theta(s_{\mathrm{flip}}))^\gamma]}{\left(\mathbb{E}_{\sigma(g_\theta)}[\sigma(g_\theta(s_{\mathrm{flip}}))^\gamma]\right)^{\frac{\gamma}{1+\gamma}}} \\
&= (1-\epsilon)S_{\mathrm{PS}}(p_{\mathcal{D}}(s)\|\sigma(g_\theta(s))) - \epsilon\frac{\mathbb{E}_{p(s_{\mathrm{flip}})}[\sigma(g_\theta(s_{\mathrm{flip}}))^\gamma]}{\left(\mathbb{E}_{\sigma(g_\theta)}[\sigma(g_\theta(s_{\mathrm{flip}}))^\gamma]\right)^{\frac{\gamma}{1+\gamma}}}.
\end{aligned}$$

Under Assumption 1, the optimal solution of $\mathrm{argmin}_\theta S_\gamma(\tilde{p}_{\mathcal{D}}^{(\epsilon)}\|m_\eta(\tilde{s}))$ will be close to that of $\mathrm{argmin}_\theta S_{\mathrm{PS}}(p_{\mathcal{D}}(s)\|\sigma(g_\theta(s)))$. This implies that Hölder-DPO with $\phi(h) = -h^{1+\gamma}$ is *robust to heavy contamination*, since it does not require the contamination ratio $\epsilon$ to be small.

However, this objective function ignores the parameter $\xi$, which was introduced in the extended model to estimate the contamination ratio. This is because

$$\begin{aligned}
S_{\mathrm{PS}}(\tilde{p}_{\mathcal{D}}^{(\epsilon)}\|m_\eta(\tilde{s})) &= -\frac{\mathbb{E}_{p_{\tilde{\mathcal{D}}}^{(\epsilon)}}[m_\eta(\tilde{s})^\gamma]}{\left(\mathbb{E}_{m_\eta}[m_\eta(\tilde{s})^\gamma]\right)^{\frac{\gamma}{1+\gamma}}} \\
&= -\frac{\mathbb{E}_{p_{\tilde{\mathcal{D}}}^{(\epsilon)}}[\sigma(g_\theta(\tilde{s}))^\gamma]}{\left(\int \sigma(g_\theta(\tilde{s}))^{1+\gamma}\mathrm{d}\tilde{s}\right)^{\frac{\gamma}{1+\gamma}}} \\
&= S_{\mathrm{PS}}(\tilde{p}_{\mathcal{D}}^{(\epsilon)}\|\sigma(g_\theta(\tilde{s}))),
\end{aligned}$$

which implies that the parameter $\xi$ in Eq. (25) does not influence the optimization, and therefore cannot serve as an estimator of $(1-\epsilon)$. In fact, even when using the enlarged model $m_\eta$, we have,

for all $\xi > 0$,

$$\underset{\eta = \{\xi, \theta\}}{\arg\min} \, S_{\mathrm{PS}}(\tilde{p}_{\mathcal{D}}^{(\epsilon)} \| m_\eta(\tilde{s})) = \underset{\theta}{\arg\min} \, S_{\mathrm{PS}}(\tilde{p}_{\mathcal{D}}^{(\epsilon)} \| \sigma(g_\theta(\tilde{s}))),$$

which confirms that $\xi$ has no effect on the solution.

On the other hand, the enlarged model becomes effective when we use the DP divergence. When we set $\phi(h) = \gamma - (1 + \gamma)h$, the Hölder-DPO objective becomes:

$$
\begin{aligned}
S_\gamma(\tilde{p}_{\mathcal{D}}^{(\epsilon)} \| m_\eta(\tilde{s})) &= \underbrace{\gamma \mathbb{E}_{m_\eta}[m_\eta(\tilde{s})^\gamma] - (1 + \gamma)\mathbb{E}_{p_{\mathcal{D}}^{(\epsilon)}}[m_\eta(\tilde{s})^\gamma]}_{=: S_{\mathrm{DP}}(\tilde{p}_{\mathcal{D}}^{(\epsilon)} \| m_\eta(\tilde{s}))} \\
&= \gamma \mathbb{E}_{m_\eta}[m_\eta(s)^\gamma] - (1 - \epsilon)(1 + \gamma)\mathbb{E}_{p_{\mathcal{D}}}[m_\eta(s)^\gamma] - \epsilon(1 + \gamma)\mathbb{E}_{p(s_{\mathrm{flip}})}[m_\eta(s_{\mathrm{flip}})^\gamma] \\
&= S_{\mathrm{DP}}((1 - \epsilon)p_{\mathcal{D}} \| m_\eta(s)) - \epsilon(1 + \gamma)\xi \mathbb{E}_{p(s_{\mathrm{flip}})}[\sigma(g_\theta(\tilde{s}))^\gamma].
\end{aligned}
$$

If $\mathbb{E}_{p(s_{\mathrm{flip}})}[\sigma(g_\theta(s_{\mathrm{flip}}))^\gamma] \approx 0$ around $\theta = \theta^*$, then the optimal solution of $\arg\min_\eta S_\gamma(\tilde{p}_{\mathcal{D}}^{(\epsilon)} \| m_\eta(\tilde{s}))$ is close to that of $\arg\min_\eta S_{\mathrm{DP}}((1 - \epsilon)p_{\mathcal{D}} \| m_\eta(s))$. Recalling that the DP divergence is strictly proper over the set of non-negative functions [53], this implies that minimizing the DP divergence with the extended model allows for estimation of both the target parameter $\theta^*$ and the clean-data ratio $1 - \epsilon$.

## G   IF Analysis for the DPO variants (summarized in Theorem 3)

### G.1   rDPO do not satisfy the redescending property

The objective of rDPO [23] is as follows:

$$\widetilde{\mathcal{L}}_{\mathrm{rDPO}}(\pi_\theta; \pi_{\mathrm{ref}}) := \frac{(1 - c)\mathbb{E}_{p_{\mathcal{D}}^{(\epsilon)}}[-\log \sigma(g_\theta(\tilde{s}))] - c\mathbb{E}_{p_{\mathcal{D}}^{(\epsilon)}}[-\log \sigma(-g_\theta(\tilde{s}))]}{1 - 2c}, \tag{28}$$

where $0 \le c < 1/2$.

We first show the IF for the rDPO.

**Theorem 6.** *Suppose $\theta^*$ denotes the optimal parameters learned from the clean dataset $p_{\mathcal{D}}$, and $\theta^*(\epsilon)$ denotes those learned from the $\epsilon$-contaminated dataset $p_{\mathcal{D}}^{(\epsilon)}$. Let the Hessian $H_{\theta^*}^{(rDPO)}(s) := \nabla_\theta^2 \mathcal{L}_{\mathrm{rDPO}}(s, \pi_\theta)|_{\theta = \theta^*}$ is positive definite. Then, the IF for the rDPO is given by:*

$$\mathrm{IF}_{\mathrm{rDPO}}(x, \theta, p_{\mathcal{D}}) = -\left( \mathbb{E}_{p_{\mathcal{D}}} \left[ H_{\theta^*}^{(rDPO)}(s) \right] \right)^{-1} \mathbb{E}_{p(s_{\mathrm{flip}})}[F_{\theta^*}^{(rDPO)}(s_{\mathrm{flip}})], \tag{29}$$

*where $F_{\theta^*}^{(rDPO)}(s_{\mathit{flip}}) := \xi_{\theta^*}(s_{\mathit{flip}})\left( \nabla_\theta \log \pi_{\theta^*}(y_{\mathit{win}}^{\mathit{flip}} \mid x) - \nabla_\theta \log \pi_{\theta^*}(y_{\mathit{lose}}^{\mathit{flip}} \mid x) \right)$ and $\xi_{\theta^*}(s_{\mathit{flip}}) := \frac{1-c}{1-2c}\sigma(-g_{\theta^*}(s_{\mathit{flip}})) + \frac{c}{1-2c}\sigma(g_{\theta^*}(s_{\mathit{flip}}))$.*

*Proof.* The gradient of Eq. (28) under $p_{\mathcal{D}}^{(\epsilon)}$ is given by

$$\nabla_\theta \widetilde{\mathcal{L}}_{\mathrm{rDPO}}(\pi_\theta; \pi_{\mathrm{ref}}) = -\beta \mathbb{E}_{p_{\mathcal{D}}^{(\epsilon)}}\left[ \xi_{\theta^*(\epsilon)}(\tilde{s})\left( \nabla_\theta \log \pi_\theta(\tilde{y}_{\mathrm{win}} \mid \tilde{x}) - \nabla_\theta \log \pi_\theta(\tilde{y}_{\mathrm{lose}} \mid \tilde{x}) \right) \right],$$

where

$$\xi_{\theta^*(\epsilon)}(\tilde{s}) := \frac{1 - c}{1 - 2c}\sigma(-g_{\theta^*(\epsilon)}(\tilde{s})) + \frac{c}{1 - 2c}\sigma(g_{\theta^*(\epsilon)}(\tilde{s})).$$

From the definition of $\theta^*(\epsilon)$, we have $0 = \nabla_\theta \widetilde{\mathcal{L}}_{\mathrm{rDPO}}(\pi_\theta; \pi_{\mathrm{ref}})|_{\theta=\theta^*(\epsilon)}$. By taking the derivation of this term w.r.t. $\epsilon$, we obtain

$$
\begin{aligned}
0 &= \frac{\partial}{\partial \epsilon} \nabla_\theta \widetilde{\mathcal{L}}_{\mathrm{rDPO}}(\pi_\theta; \pi_{\mathrm{ref}}) \bigg|_{\theta=\theta^*(\epsilon)} \\
&= -\beta \frac{\partial}{\partial \epsilon} \mathbb{E}_{p_{\widetilde{\mathcal{D}}}^{(\epsilon)}} \bigg[ \underbrace{\xi_{\theta^*(\epsilon)}(\tilde{s}) \Big( \nabla_\theta \log \pi_\theta(\tilde{y}_{\mathrm{win}} \mid \tilde{x}) - \nabla_\theta \log \pi_\theta(\tilde{y}_{\mathrm{lose}} \mid \tilde{x}) \Big)}_{=: F_{\theta^*(\epsilon)}^{(\mathrm{rDPO})}(\tilde{s})} \bigg] \\
&= -\beta \bigg\{ \int \bigg\{ \frac{\partial}{\partial \epsilon} p_{\widetilde{\mathcal{D}}}^{(\epsilon)}(\tilde{s}) \bigg\} F_{\theta^*(\epsilon)}^{(\mathrm{rDPO})}(\tilde{s}) \mathrm{d}\tilde{s} + \mathbb{E}_{p_{\widetilde{\mathcal{D}}}^{(\epsilon)}} \bigg[ \frac{\partial \theta^*(\epsilon)}{\partial \epsilon} H_{\theta^*(\epsilon)}^{(\mathrm{rDPO})}(\tilde{s}) \bigg] \bigg\}, \qquad (30)
\end{aligned}
$$

where $H_{\theta^*(\epsilon)}^{(\mathrm{rDPO})}(\tilde{s}) := \frac{\partial F_{\theta^*(\epsilon)}^{(\mathrm{rDPO})}(\tilde{s})}{\partial \theta^*(\epsilon)}$.

From Definition 1, we obtain

$$
\int \bigg\{ \frac{\partial}{\partial \epsilon} p_{\widetilde{\mathcal{D}}}^{(\epsilon)}(\tilde{s}) \bigg\} F_{\theta^*(\epsilon)}^{(\mathrm{rDPO})}(\tilde{s}) \mathrm{d}\tilde{s} = \mathbb{E}_{p(s_{\mathrm{flip}})}[F_{\theta^*(\epsilon)}^{(\mathrm{rDPO})}(s_{\mathrm{flip}})] - \mathbb{E}_{p_{\mathcal{D}}}\bigg[ F_{\theta^*(\epsilon)}^{(\mathrm{rDPO})}(s) \bigg],
$$

where $F_{\theta^*(\epsilon)}^{(\mathrm{rDPO})}(s_{\mathrm{flip}}) := \xi_{\theta^*(\epsilon)}(s_{\mathrm{flip}})(\nabla_\theta \log \pi_{\theta^*(\epsilon)}(y_{\mathrm{win}}^{\mathrm{flip}} \mid x) - \nabla_\theta \log \pi_{\theta^*(\epsilon)}(y_{\mathrm{lose}}^{\mathrm{flip}} \mid x))$. By taking $\epsilon \to 0$, we have

$$
\bigg( \int \bigg\{ \frac{\partial}{\partial \epsilon} p_{\widetilde{\mathcal{D}}}^{(\epsilon)}(\tilde{s}) \bigg\} F_{\theta^*(\epsilon)}^{(\mathrm{rDPO})}(\tilde{s}) \mathrm{d}\tilde{s} \bigg) \bigg|_{\epsilon=0} = \mathbb{E}_{p(s_{\mathrm{flip}})}[F_{\theta^*(\epsilon)}^{(\mathrm{rDPO})}(s_{\mathrm{flip}})],
$$

since $\theta^{(*)}(\epsilon) \to \theta^{(*)}$ and thus $\mathbb{E}_{p_{\mathcal{D}}}[F_{\theta^*}^{(\mathrm{rDPO})}(s)] = \nabla_\theta \mathcal{L}_{\mathrm{rDPO}}(\pi_\theta; \pi_{\mathrm{ref}})|_{\theta=\theta^*} = 0$ from the first-order optimal condition.

Furthermore, we also obtain

$$
\mathbb{E}_{p_{\widetilde{\mathcal{D}}}^{(\epsilon)}} \bigg[ \frac{\partial \theta^*(\epsilon)}{\partial \epsilon} H_{\theta^*(\epsilon)}^{(\mathrm{rDPO})}(\tilde{s}) \bigg] \bigg|_{\epsilon=0} = \mathbb{E}_{p_{\mathcal{D}}} \bigg[ \frac{\partial \theta^*(\epsilon)}{\partial \epsilon} H_{\theta^*}^{(\mathrm{rDPO})}(s) \bigg],
$$

where $H_{\theta^*}^{(\mathrm{rDPO})}(s) := \frac{\partial F_{\theta^*}^{(\mathrm{rDPO})}(s)}{\partial \theta^*}$.

Then, Eq. (30) under $\epsilon \to 0$ can be rewritten as

$$
0 = \bigg( \frac{\partial}{\partial \epsilon} \nabla_\theta \widetilde{\mathcal{L}}_{\mathrm{rDPO}}(\pi_\theta; \pi_{\mathrm{ref}}) \bigg|_{\theta=\theta^*(\epsilon)} \bigg) \bigg|_{\epsilon=0} = -\beta \bigg\{ \mathbb{E}_{p(s_{\mathrm{flip}})}[F_{\theta^*(\epsilon)}^{(\mathrm{rDPO})}(s_{\mathrm{flip}})] + \mathbb{E}_{p_{\mathcal{D}}} \bigg[ \frac{\partial \theta^*(\epsilon)}{\partial \epsilon} \bigg|_{\epsilon=0} H_{\theta^*}^{(\mathrm{rDPO})}(s) \bigg] \bigg\}.
$$

By solving the above equality w.r.t. $\frac{\partial \theta^*(\epsilon)}{\partial \epsilon}$, we obtain

$$
\frac{\partial \theta^*(\epsilon)}{\partial \epsilon} \bigg|_{\epsilon=0} = - \bigg( \mathbb{E}_{p_{\mathcal{D}}} \bigg[ H_{\theta^*}^{(\mathrm{rDPO})}(s) \bigg] \bigg)^{-1} \mathbb{E}_{p(s_{\mathrm{flip}})}[F_{\theta^*(\epsilon)}^{(\mathrm{rDPO})}(s_{\mathrm{flip}})].
$$

This completes the proof. $\qquad \square$

**Corollary 3** (rDPO is not robust). *Suppose that the policy gradient $\nabla_\theta \log \pi_\theta(y \mid x)$ is bounded by $C$ and satisfies L-Lipchitz in $\theta$, where $0 < C < \infty$ and $0 < L < \infty$. Let $0 \le c < 1/2$. Then, under Theorem 6, the IF of rDPO do not satisfy the robustness condition in Definition 2, i.e., $\lim_{\hat{r}_{\theta^*}(x, y_{\mathrm{lose}}^{\mathit{flip}}) \to \infty} \|\mathrm{IF}_{\mathrm{rDPO}}(x, \theta, p_{\mathcal{D}})\| \ne 0$.*

*Proof.* From Theorem 6, the IF for rDPO is given by

$$
\mathrm{IF}_{\mathrm{rDPO}} = - \bigg( \mathbb{E}_{p_{\mathcal{D}}} \bigg[ H_{\theta^*}^{(\mathrm{rDPO})}(s) \bigg] \bigg)^{-1} \mathbb{E}_{p(s_{\mathrm{flip}})}[F_{\theta^*(\epsilon)}^{(\mathrm{rDPO})}(s_{\mathrm{flip}})].
$$

From the positive definite assumption on the Hessian $H_{\theta^*}^{(\mathrm{rDPO})}(s)$, it follows that its expectation $\mathbb{E}_{p_{\mathcal{D}}}[H_{\theta^*}^{(\mathrm{rDPO})}(s)]$ is also a positive definite matrix. Let $L' = \lambda_{\min}(\mathbb{E}_{p_{\mathcal{D}}}[H_{\theta^*}^{(\mathrm{rDPO})}(s)]) > 0$ be its minimum eigenvalue. Then the norm of its inverse is bounded: $\|(\mathbb{E}_{p_{\mathcal{D}}}[H_{\theta^*}^{(\mathrm{rDPO})}(s)])^{-1}\| \le 1/L'$.

Furthermore, from the assumption that $\|\nabla_\theta \log \pi_\theta(y \mid x)\| \leq C$, we have $\|\nabla_\theta \log \pi_\theta(y_{\text{win}}^{\text{flip}} \mid x) - \nabla_\theta \log \pi_\theta(y_{\text{lose}}^{\text{flip}} \mid x)\| \leq 2C$.

Taking the limit required by Definition 2, we have:

$$\lim_{\hat{r}_{\theta^*}(x, y_{\text{lose}}^{\text{flip}}) \to \infty} \|\text{IF}_{\text{rDPO}}\|$$

$$\leq \lim_{\hat{r}_{\theta^*}(x, y_{\text{lose}}^{\text{flip}}) \to \infty} \left\| \left( \mathbb{E}_{p_\mathcal{D}} \left[ H_{\theta^*}^{(\text{rDPO})}(s) \right] \right)^{-1} \right\| \cdot \lim_{\hat{r}_{\theta^*}(x, y_{\text{lose}}^{\text{flip}}) \to \infty} \left\| \mathbb{E}_{p(s_{\text{flip}})} [F_{\theta^*(\epsilon)}^{(\text{rDPO})}(s_{\text{flip}})] \right\|$$

$$\leq (1/L') \cdot \lim_{\hat{r}_{\theta^*}(x, y_{\text{lose}}^{\text{flip}}) \to \infty} \mathbb{E}_{p(s_{\text{flip}})} \left[ \underbrace{\xi_{\theta^*}(s_{\text{flip}})}_{\text{IF Weight}} \cdot \underbrace{\|\nabla_\theta \log \pi_\theta(y_{\text{win}}^{\text{flip}} \mid x) - \nabla_\theta \log \pi_\theta(y_{\text{lose}}^{\text{flip}} \mid x)\|}_{\leq 2C} \right]$$

$$\leq (1/L') \cdot \lim_{\hat{r}_{\theta^*}(x, y_{\text{lose}}^{\text{flip}}) \to \infty} \mathbb{E}_{p(s_{\text{flip}})} [\xi_{\theta^*}(s_{\text{flip}})] \cdot 2C,$$

where

$$\xi_{\theta^*}(s_{\text{flip}}) = \frac{1-c}{1-2c} \sigma(-g_\theta(s_{\text{flip}})) + \frac{c}{1-2c} \sigma(g_\theta(s_{\text{flip}})).$$

We now evaluate the limit of the IF weight $\xi_{\theta^*}(s_{\text{flip}})$:

$$\lim_{\hat{r}_{\theta^*}(x, y_{\text{lose}}^{\text{flip}}) \to \infty} \xi_{\theta^*}(s_{\text{flip}}) = \left( \frac{1-c}{1-2c} \cdot 1 \right) + \left( \frac{c}{1-2c} \cdot 0 \right) = \frac{1-c}{1-2c}.$$

By the bounded convergence theorem, the limit of the expectation is the expectation of the limit. Thus, the IF limit is upper bounded by:

$$\lim_{\hat{r}_{\theta^*}(x, y_{\text{lose}}^{\text{flip}}) \to \infty} \|\text{IF}_{\text{rDPO}}\| \leq (1/L') \cdot \left( \frac{1-c}{1-2c} \right) \cdot 2C = \frac{2C(1-c)}{L'(1-2c)}.$$

The fact $0 < \frac{2C(1-c)}{L'(1-2c)} < \infty$ (according to $0 < C < \infty$, $0 \leq c < 1/2$, and $0 < L' < \infty$) completes the proof. $\qquad\square$

## G.2 cDPO do not satisfy the redescending property

The objective of cDPO [76] is as follows:

$$\widetilde{\mathcal{L}}_{\text{cDPO}}(\pi_\theta; \pi_{\text{ref}}) := (1-c)\mathbb{E}_{p_{\widetilde{\mathcal{D}}}^{(\epsilon)}}[-\log \sigma(g_\theta(\tilde{s}))] - c\mathbb{E}_{p_{\widetilde{\mathcal{D}}}^{(\epsilon)}}[-\log \sigma(-g_\theta(\tilde{s}))]. \tag{31}$$

We first show the IF for the cDPO.

**Theorem 7.** *Suppose $\theta^*$ denotes the optimal parameters learned from the clean dataset $p_\mathcal{D}$, and $\theta^*(\epsilon)$ denotes those learned from the $\epsilon$-contaminated dataset $p_{\widetilde{\mathcal{D}}}^{(\epsilon)}$. Let the Hessian $H_{\theta^*}^{(cDPO)}(s) := \nabla_\theta^2 \mathcal{L}_{\text{cDPO}}(s, \pi_\theta)|_{\theta=\theta^*}$ is positive definite. Then, the IF for the rDPO is given by:*

$$\text{IF}_{\text{cDPO}}(x, \theta, p_\mathcal{D}) = -\left( \mathbb{E}_{p_\mathcal{D}} \left[ H_{\theta^*}^{(cDPO)}(s) \right] \right)^{-1} \mathbb{E}_{p(s_{flip})} [F_{\theta^*}^{(cDPO)}(s_{flip})], \tag{32}$$

*where $F_{\theta^*}^{(cDPO)}(s_{flip}) := \xi_{\theta^*}(s_{flip}) \left( \nabla_\theta \log \pi_{\theta^*}(y_{win}^{flip} \mid x) - \nabla_\theta \log \pi_{\theta^*}(y_{lose}^{flip} \mid x) \right)$ and $\xi_{\theta^*}(s_{flip}) := (1-c)\sigma(-g_{\theta^*}(s_{flip})) + c\sigma(g_{\theta^*}(s_{flip}))$.*

*Proof.* The proof follows from the same argument as in Theorem 6, ignoring the $(1-2c)$ term in the denominator. $\qquad\square$

**Corollary 4** (cDPO is not robust). *Suppose that the policy gradient $\nabla_\theta \log \pi_\theta(y \mid x)$ is bounded by $C$ and satisfies $L$-Lipchitz in $\theta$, where $0 < C < \infty$ and $0 < L < \infty$. Let $0 \leq c < 1$. Then, under Theorem 7, the IF of cDPO do not satisfy the robustness condition in Definition 2, i.e., $\lim_{\hat{r}_{\theta^*}(x, y_{lose}^{flip}) \to \infty} \|\text{IF}_{\text{cDPO}}(x, \theta, p_\mathcal{D})\| \neq 0$.*

*Proof.* By following the proof in Corollary 3 and ignoring the $(1 - 2c)$ term in the denominator, we have $0 < \mathbb{E}_{p_\mathcal{D}}[H_{\theta^*}^{(\text{cDPO})}(s)] \leq \xi_{\theta^*}(s) \cdot L < \infty$ and thus $\lim_{\hat{r}_{\theta^*}(x, y_{\text{lose}}^{\text{flip}}) \to \infty} \|\text{IF}_{\text{cDPO}}(x, \theta, p_\mathcal{D})\|_2 \leq 2C(1 - c)/L'$, where $0 < L' \leq \mathbb{E}_{p_\mathcal{D}}[H_{\theta^*}(s)]$ and we use the fact that $\lim_{\hat{r}_{\theta^*}(x, y_{\text{lose}}^{\text{flip}}) \to \infty} \xi_{\theta^*}(s_{\text{flip}}) = 1 - c$. The fact $0 < 2C(1 - c)/L' < \infty$ according to $0 < C < \infty$, $0 \leq c < 1$, and $0 < L' < \infty$ completes the proof. $\qquad\square$

### G.3 IPO do not satisfy the redescending property

The objective of IPO [37] is as follows:

$$\widetilde{\mathcal{L}}_{\text{IPO}}(\pi_\theta; \pi_{\text{ref}}) := \mathbb{E}_{p_{\widetilde{\mathcal{D}}}^{(\epsilon)}}\left[\left(\frac{g_\theta(\tilde{s})}{\beta} - \frac{1}{2\beta}\right)^2\right]. \tag{33}$$

We first show the IF for the IPO.

**Theorem 8.** *Suppose $\theta^*$ denotes the optimal parameters learned from the clean dataset $p_\mathcal{D}$, and $\theta^*(\epsilon)$ denotes those learned from the $\epsilon$-contaminated dataset $p_{\widetilde{\mathcal{D}}}^{(\epsilon)}$. Let the Hessian $H_{\theta^*}^{(\text{IPO})}(s) := \nabla_\theta^2 \mathcal{L}_{\text{IPO}}(s, \pi_\theta)|_{\theta=\theta^*}$ be positive definite. Then, the IF for the IPO is given by:*

$$\text{IF}_{\text{IPO}}(x, \theta, p_\mathcal{D}) = -\left(\mathbb{E}_{p_\mathcal{D}}\left[H_{\theta^*}^{(\text{IPO})}(s)\right]\right)^{-1}\mathbb{E}_{p(s_{\text{flip}})}[F_{\theta^*}^{(\text{IPO})}(s_{\text{flip}})], \tag{34}$$

*where $F_{\theta^*}^{(\text{IPO})}(s_{\text{flip}}) := 2\left(\frac{g_\theta(s_{\text{flip}})}{\beta} - \frac{1}{2\beta}\right)\left(\nabla_\theta \log \pi_{\theta^*}(y_{win}^{flip} \mid x) - \nabla_\theta \log \pi_{\theta^*}(y_{lose}^{flip} \mid x)\right).$*

*Proof.* The proof follows from the same argument as in Theorem 2 under the following gradient of Eq. (33):

$$\nabla_\theta \widetilde{\mathcal{L}}_{\text{IPO}}(\pi_\theta; \pi_{\text{ref}}) = \mathbb{E}_{p_{\widetilde{\mathcal{D}}}^{(\epsilon)}}\left[2\left(\frac{g_\theta(\tilde{s})}{\beta} - \frac{1}{2\beta}\right)\left(\nabla_\theta \log \pi_\theta(\tilde{y}_{\text{win}} \mid \tilde{x}) - \nabla_\theta \log \pi_\theta(\tilde{y}_{\text{lose}} \mid \tilde{x})\right)\right].$$

$\qquad\square$

**Corollary 5** (IPO is not robust). *Suppose that the policy gradient $\nabla_\theta \log \pi_\theta(y \mid x)$ is bounded by $C$, where $0 < C < \infty$. Let $g_{\theta^*}(s)$ be bounded over $p_\mathcal{D}(s)$. Then, under Theorem 8, the IF of IPO do not satisfy the robustness condition in Definition 2, i.e., $\lim_{\hat{r}_{\theta^*}(x, y_{\text{lose}}^{flip}) \to \infty} \|\text{IF}_{\text{IPO}}(x, \theta, p_\mathcal{D})\|_2 = \infty$.*

*Proof.* From Theorem 8, the IF for IPO is $\text{IF}_{\text{IPO}} = -(\mathbb{E}_{p_\mathcal{D}}[H_{\theta^*}^{(\text{IPO})}(s)])^{-1}\mathbb{E}_{p(s_{\text{flip}})}[F_{\theta^*}^{(\text{IPO})}]$. From the positive definite assumption on the Hessian, let $L' = \lambda_{\min}(\mathbb{E}_{p_\mathcal{D}}[H_{\theta^*}^{(\text{IPO})}(s)]) > 0$. The norm of its inverse is bounded: $\|(\mathbb{E}_{p_\mathcal{D}}[H_{\theta^*}^{(\text{IPO})}(s)])^{-1}\| \leq 1/L'$. The gradient term $\|\nabla_\theta \log \pi_\theta(\dots)\|$ is also bounded by $2C$ from the assumption.

We analyze the limit of the IF:

$$\lim_{\hat{r}_{\theta^*}(x, y_{\text{lose}}^{\text{flip}}) \to \infty} \|\text{IF}_{\text{IPO}}\|$$

$$\leq (1/L') \cdot \lim_{\hat{r}_{\theta^*}(x, y_{\text{lose}}^{\text{flip}}) \to \infty} \left\|\mathbb{E}_{p(s_{\text{flip}})}[F_{\theta^*}^{(\text{IPO})}(s_{\text{flip}})]\right\|$$

$$\leq (1/L') \cdot \mathbb{E}_{p(s_{\text{flip}})}\left[\lim_{\hat{r}_{\theta^*}(x, y_{\text{lose}}^{\text{flip}}) \to \infty} \left\|\underbrace{2 \cdot \left(\frac{g_\theta(s_{\text{flip}})}{\beta} - \frac{1}{2\beta}\right)}_{\text{IF Weight}} \cdot \underbrace{\left(\nabla_\theta \log \pi_{\theta^*}(y_{\text{win}}^{\text{flip}} \mid x) - \nabla_\theta \log \pi_{\theta^*}(y_{\text{lose}}^{\text{flip}} \mid x)\right)}_{\text{Gradient Term}}\right\|\right],$$

where we use Fatou's Lemma to exchange the limit.

In the limit, $\lim_{\hat{r}_{\theta^*}(x, y_{\text{lose}}^{\text{flip}}) \to \infty}$, the IF weight term diverges:

$$\lim_{\hat{r}_{\theta^*}(x, y_{\text{lose}}^{\text{flip}}) \to \infty} \left\|2 \cdot \left(\frac{g_\theta}{\beta} - \frac{1}{2\beta}\right)\right\| = \infty.$$

Since the IF is proportional to the product of this diverging term ($\to \infty$) and a bounded, non-zero gradient term ($\leq 2C$), the IF itself diverges, that is,

$$\lim_{\hat{r}_{\theta^*}(x, y_{\text{lose}}^{\text{flip}}) \to \infty} \|\text{IF}_{\text{IPO}}(x, \theta, p_{\mathcal{D}})\| = \infty.$$

This completes the proof. $\qquad \square$

### G.4 Dr. DPO do not satisfy the redescending property

The objective of Dr. DPO [105] is as follows:

$$\widetilde{\mathcal{L}}_{\text{Dr. DPO}}(\pi_\theta; \pi_{\text{ref}}) := -\beta' \log \mathbb{E}_{p_{\widetilde{\mathcal{D}}}^{(\epsilon)}} \left[ \exp \left( \frac{\log \sigma(g_\theta(\tilde{s}))}{\beta'} \right) \right]. \tag{35}$$

We first show the IF for the Dr. DPO.

**Theorem 9.** *Suppose $\theta^*$ denotes the optimal parameters learned from the clean dataset $p_{\mathcal{D}}$, and $\theta^*(\epsilon)$ denotes those learned from the $\epsilon$-contaminated dataset $p_{\widetilde{\mathcal{D}}}^{(\epsilon)}$. Let the Hessian $H_{\theta^*}^{(Dr. DPO)}(s) := \nabla_\theta^2 \mathcal{L}_{\text{Dr. DPO}}(s, \pi_\theta)|_{\theta=\theta^*}$ is positive definite. Then, the IF for the Dr. DPO is given by:*

$$\text{IF}_{\text{Dr. DPO}}(x, \theta, p_{\mathcal{D}}) = - \left( \mathbb{E}_{p_{\mathcal{D}}} \left[ H_{\theta^*}^{(Dr. DPO)}(s) \right] \right)^{-1} \mathbb{E}_{p(s_{\text{flip}})}[F_{\theta^*}^{(Dr. DPO)}(s_{\text{flip}})], \tag{36}$$

*where $F_{\theta^*}^{(Dr. DPO)}(s_{\text{flip}}) := w_{\theta^*}(s_{\text{flip}})\sigma(-g_{\theta^*}(s_{\text{flip}})) \left( \nabla_\theta \log \pi_{\theta^*}(y_{\text{win}}^{\text{flip}} \mid x) - \nabla_\theta \log \pi_{\theta^*}(y_{\text{lose}}^{\text{flip}} \mid x) \right)$ and $w_{\theta^*}(s_{\text{flip}}) := \exp \left( \frac{\log \sigma(g_{\theta^*}(s_{\text{flip}}))}{\beta'} \right) / \mathbb{E}_{p(s_{\text{flip}})} \left[ \exp \left( \frac{\log \sigma(g_{\theta^*}(s_{\text{flip}}))}{\beta'} \right) \right].$*

*Proof.* The gradient of Eq. (35) under $p_{\widetilde{\mathcal{D}}}^{(\epsilon)}$ is given by

$$\nabla_\theta \widetilde{\mathcal{L}}_{\text{Dr. DPO}}(\pi_\theta; \pi_{\text{ref}}) = -\beta \mathbb{E}_{p_{\widetilde{\mathcal{D}}}^{(\epsilon)}} \left[ w_{\theta^*(\epsilon)}(\tilde{s})\sigma(-g_{\theta^*(\epsilon)}(\tilde{s})) \left( \nabla_\theta \log \pi_\theta(\tilde{y}_{\text{win}} \mid \tilde{x}) - \nabla_\theta \log \pi_\theta(\tilde{y}_{\text{lose}} \mid \tilde{x}) \right) \right],$$

where

$$w_{\theta^*(\epsilon)}(\tilde{s}) := \frac{\exp \left( \frac{\log \sigma(g_{\theta^*(\epsilon)}(\tilde{s}))}{\beta'} \right)}{\mathbb{E}_{p_{\widetilde{\mathcal{D}}}^{(\epsilon)}} \left[ \exp \left( \frac{\log \sigma(g_{\theta^*(\epsilon)}(\tilde{s}))}{\beta'} \right) \right]}.$$

From the definition of $\theta^*(\epsilon)$, we have $0 = \nabla_\theta \widetilde{\mathcal{L}}_{\text{Dr. DPO}}(\pi_\theta; \pi_{\text{ref}})|_{\theta=\theta^*(\epsilon)}$. By taking the derivation of this term w.r.t. $\epsilon$, we obtain

$$0 = \frac{\partial}{\partial \epsilon} \nabla_\theta \widetilde{\mathcal{L}}_{\text{Dr. DPO}}(\pi_\theta; \pi_{\text{ref}}) \Big|_{\theta=\theta^*(\epsilon)}$$

$$= -\beta \frac{\partial}{\partial \epsilon} \mathbb{E}_{p_{\widetilde{\mathcal{D}}}^{(\epsilon)}} \left[ \underbrace{w_{\theta^*(\epsilon)}(\tilde{s})\sigma(-g_{\theta^*(\epsilon)}(\tilde{s})) \left( \nabla_\theta \log \pi_{\theta^*(\epsilon)}(\tilde{y}_{\text{win}} \mid \tilde{x}) - \nabla_\theta \log \pi_{\theta^*(\epsilon)}(\tilde{y}_{\text{lose}} \mid \tilde{x}) \right)}_{=:F_{\theta^*(\epsilon)}^{(Dr. DPO)}(\tilde{s})} \right]$$

$$= -\beta \left\{ \int \left\{ \frac{\partial}{\partial \epsilon} p_{\widetilde{\mathcal{D}}}^{(\epsilon)}(\tilde{s}) \right\} F_{\theta^*(\epsilon)}^{(Dr. DPO)}(\tilde{s}) \mathrm{d}\tilde{s} + \mathbb{E}_{p_{\widetilde{\mathcal{D}}}^{(\epsilon)}} \left[ \frac{\partial}{\partial \epsilon} F_{\theta^*(\epsilon)}^{(Dr. DPO)}(\tilde{s}) \right] \right\}$$

$$= -\beta \left\{ \int \left\{ \frac{\partial}{\partial \epsilon} p_{\widetilde{\mathcal{D}}}^{(\epsilon)}(\tilde{s}) \right\} F_{\theta^*(\epsilon)}^{(Dr. DPO)}(\tilde{s}) \mathrm{d}\tilde{s} + \mathbb{E}_{p_{\widetilde{\mathcal{D}}}^{(\epsilon)}} \left[ \frac{\partial \theta^*(\epsilon)}{\partial \epsilon} \frac{\partial F_{\theta^*(\epsilon)}^{(Dr. DPO)}(\tilde{s})}{\partial \theta^*(\epsilon)} \right] \right\}$$

$$= -\beta \left\{ \int \left\{ \frac{\partial}{\partial \epsilon} p_{\widetilde{\mathcal{D}}}^{(\epsilon)}(\tilde{s}) \right\} F_{\theta^*(\epsilon)}^{(Dr. DPO)}(\tilde{s}) \mathrm{d}\tilde{s} + \mathbb{E}_{p_{\widetilde{\mathcal{D}}}^{(\epsilon)}} \left[ \frac{\partial \theta^*(\epsilon)}{\partial \epsilon} H_{\theta^*(\epsilon)}^{(Dr. DPO)}(\tilde{s}) \right] \right\}, \tag{37}$$

where $H_{\theta^*(\epsilon)}^{(Dr. DPO)}(\tilde{s}) := \frac{\partial F_{\theta^*(\epsilon)}^{(Dr. DPO)}(\tilde{s})}{\partial \theta^*(\epsilon)}$.

From Definition 1, we obtain

$$\int \left\{ \frac{\partial}{\partial \epsilon} p_{\widetilde{\mathcal{D}}}^{(\epsilon)}(\tilde{s}) \right\} F_{\theta^*(\epsilon)}^{(\text{Dr. DPO})}(\tilde{s}) \mathrm{d}\tilde{s} = \mathbb{E}_{p(s_{\text{flip}})}[F_{\theta^*(\epsilon)}^{(\text{Dr. DPO})}(s_{\text{flip}})] - \mathbb{E}_{p_{\mathcal{D}}}\left[ F_{\theta^*(\epsilon)}^{(\text{Dr. DPO})}(s) \right],$$

where $F_{\theta^*}(s_{\text{flip}}) := w_{\theta^*}(s_{\text{flip}})\sigma(-g_{\theta^*}(s_{\text{flip}}))(\nabla_\theta \log \pi_{\theta^*}(y_{\text{win}}^{\text{flip}} \mid x) - \nabla_\theta \log \pi_{\theta^*}(y_{\text{lose}}^{\text{flip}} \mid x))$. By taking $\epsilon \to 0$, we have

$$\left( \int \left\{ \frac{\partial}{\partial \epsilon} p_{\widetilde{\mathcal{D}}}^{(\epsilon)}(\tilde{s}) \right\} F_{\theta^*(\epsilon)}^{(\text{Dr. DPO})}(\tilde{s}) \mathrm{d}\tilde{s} \right) \Bigg|_{\epsilon=0} = \mathbb{E}_{p(s_{\text{flip}})}[F_{\theta^*(\epsilon)}^{(\text{Dr. DPO})}(s_{\text{flip}})],$$

since $\theta^{(*)}(\epsilon) \to \theta^{(*)}$ and thus $\mathbb{E}_{p_{\mathcal{D}}}[F_{\theta^*}^{(\text{Dr. DPO})}(s)] = \nabla_\theta \mathcal{L}_{\text{Dr. DPO}}(\pi_\theta; \pi_{\text{ref}})|_{\theta=\theta^*} = 0$ from the first-order optimal condition.

Furthermore, we also obtain

$$\mathbb{E}_{p_{\widetilde{\mathcal{D}}}^{(\epsilon)}} \left[ \frac{\partial \theta^*(\epsilon)}{\partial \epsilon} H_{\theta^*(\epsilon)}^{(\text{Dr. DPO})}(\tilde{s}) \right] \Bigg|_{\epsilon=0} = \mathbb{E}_{p_{\mathcal{D}}} \left[ \frac{\partial \theta^*(\epsilon)}{\partial \epsilon} H_{\theta^*}^{(\text{Dr. DPO})}(s) \right],$$

where $H_{\theta^*}^{(\text{Dr. DPO})}(s) := \frac{\partial F_{\theta^*}^{(\text{Dr. DPO})}(s)}{\partial \theta^*}$.

Then, Eq. (37) under $\epsilon \to 0$ can be rewritten as

$$0 = \left( \frac{\partial}{\partial \epsilon} \nabla_\theta \widetilde{\mathcal{L}}_{\text{Dr. DPO}}(\pi_\theta; \pi_{\text{ref}}) \Big|_{\theta=\theta^*(\epsilon)} \right) \Bigg|_{\epsilon=0}$$

$$= -\beta \left\{ \mathbb{E}_{p(s_{\text{flip}})}[F_{\theta^*}^{(\text{Dr. DPO})}(s_{\text{flip}})] + \mathbb{E}_{p_{\mathcal{D}}} \left[ \frac{\partial \theta^*(\epsilon)}{\partial \epsilon} \Big|_{\epsilon=0} H_{\theta^*}^{(\text{Dr. DPO})}(s) \right] \right\}.$$

By solving the above equality w.r.t. $\frac{\partial \theta^*(\epsilon)}{\partial \epsilon}$, we obtain

$$\frac{\partial \theta^*(\epsilon)}{\partial \epsilon} \Bigg|_{\epsilon=0} = -\left( \mathbb{E}_{p_{\mathcal{D}}} \left[ H_{\theta^*}^{(\text{Dr. DPO})}(s) \right] \right)^{-1} \mathbb{E}_{p(s_{\text{flip}})}[F_{\theta^*}^{(\text{Dr. DPO})}(s_{\text{flip}})].$$

This completes the proof. $\qquad\square$

The following lemma is crucial to show the fact that Dr. DPO does not satisfy the redescending property.

**Lemma 6** (Limit of Dr. DPO IF Weight). *Let $w_{\theta^*}(s_{\text{flip}})$ be the IF weight for Dr. DPO as defined in Theorem 9. Then, the limit of the total IF weight is 1, that is,*

$$\lim_{\hat{r}_{\theta^*}(x, y_{\text{lose}}^{\text{flip}}) \to \infty} \mathbb{E}_{p(s_{\text{flip}})} \left[ w_{\theta^*}(s_{\text{flip}}) \cdot \sigma(-g_{\theta^*}(s_{\text{flip}})) \right] = 1.$$

*Proof.* We first analyze the case where $p(s_{\text{flip}}) = \delta(s_{\text{flip}})$ (a single point mass). Here, the expectation in the denominator of $w_{\theta^*}$ is equal to the numerator, thus $w_{\theta^*}(s_{\text{flip}}) = 1$. Since $\lim \sigma(-g_{\theta^*}) = 1$, the total weight $\mathbb{E}[1 \cdot 1] = 1$.

We next analyze the case where $p(s_{\text{flip}})$ is a non-degenerate distribution. We track how fast $g_\theta(s_{\text{flip}})$ goes to $-\infty$ across the support of $p(s_{\text{flip}})$. Let us define:

$$S := \sup_{s_{\text{flip}}} g_\theta(s_{\text{flip}}), \quad r(s_{\text{flip}}) := g_\theta(s_{\text{flip}}) - S \ (\leq 0), \quad G := \{ s_{\text{flip}} \mid r(s_{\text{flip}}) = 0 \}$$

$G$ is the non-empty set of "worst-case" label-flip samples. Using the bound $\sigma(z) \approx e^z$ for $z \to -\infty$, $\sigma(g_\theta(s_{\text{flip}})) \approx e^S e^{r(s_{\text{flip}})}$. The term $\exp(\log \sigma(g_\theta)/\beta')$ simplifies to $\sigma(g_\theta)^{1/\beta'}$. Thus,

$$w_{\theta^*}(s_{\text{flip}}) \approx \frac{\left(e^S e^{r(s_{\text{flip}})}\right)^{1/\beta'}}{\mathbb{E}_{p(s_{\text{flip}})}\left[\left(e^S e^{r(s_{\text{flip}})}\right)^{1/\beta'}\right]} = \frac{\exp(r(s_{\text{flip}})/\beta')}{\mathbb{E}_{p(s_{\text{flip}})}[\exp(r(s_{\text{flip}})/\beta')]}.$$

As $S \to -\infty$, the term $w_{\theta^*}(s_{\text{flip}})$ converges to $1/p(G)$ for $s_{\text{flip}} \in G$, and to 0 for $s_{\text{flip}} \notin G$.

The total IF weight is $W_{\text{total}} = w_{\theta^*}(s) \cdot \sigma(-g_{\theta^*}(s))$. We take the limit of its expectation (using the bounded convergence theorem):

$$
\lim_{S \to -\infty} \mathbb{E}_{p(s_{\text{flip}})}[W_{\text{total}}] = \mathbb{E}_{p(s_{\text{flip}})}\left[ \lim_{S \to -\infty} w_{\theta^*}(s) \cdot \lim_{g \to -\infty} \sigma(-g_{\theta^*}(s)) \right]
$$

$$
= \int_G \left( \lim w_{\theta^*}(s) \right) \cdot (1) \cdot p(s)\mathrm{d}s + \int_{G^c} (0) \cdot (1) \cdot p(s)\mathrm{d}s
$$

$$
= \int_G \left( \frac{1}{p(G)} \right) \cdot p(s)\mathrm{d}s = \frac{1}{p(G)} \int_G p(s)\mathrm{d}s = \frac{p(G)}{p(G)} = 1.
$$

Thus, the limit of the total IF weight is 1 in all cases. $\qquad \square$

Now we can show the following corollary.

**Corollary 6.** *Suppose that the policy gradient $\nabla_\theta \log \pi_\theta(y \mid x)$ is bounded by $C$ and satisfies $L$-Lipchitz in $\theta$, where $0 < C < \infty$ and $0 < L < \infty$. Let the number of the label-flip data be $\lfloor N\epsilon \rfloor = M\ (< \infty)$, and $0 < \beta' < \infty$. Let the weight term in the gradient of Dr. DPO: $w_{\theta^*}(s)$ is bounded on $p_{\mathcal{D}}$. Then, under Theorem 9, the IF of Dr. DPO do not satisfy the robustness condition in Definition 2, i.e., $\lim_{\hat{r}_{\theta^*}(x, y_{\text{lose}}^{\text{flip}}) \to \infty} \|\mathrm{IF}_{\text{Dr. DPO}}(x, \theta, p_{\mathcal{D}})\| \neq 0$.*

*Proof.* The IF for Dr. DPO is

$$
\mathrm{IF}_{\text{Dr. DPO}} = -\left( \mathbb{E}_{p_{\mathcal{D}}}\left[ H_{\theta^*}^{(\text{Dr. DPO})}(s) \right] \right)^{-1} \mathbb{E}_{p(s_{\text{flip}})}[F_{\theta^*}^{(\text{Dr. DPO})}(s_{\text{flip}})].
$$

From the positive definite assumption on the Hessian, let $L' = \lambda_{\min}(\mathbb{E}_{p_{\mathcal{D}}}\left[ H_{\theta^*}^{(\text{Dr. DPO})}(s) \right]) > 0$. The norm of its inverse is bounded: $\|(\mathbb{E}_{p_{\mathcal{D}}}[H_{\theta^*}^{(\text{Dr. DPO})}(s)])^{-1}\| \leq 1/L'$. The gradient term is bounded by $2C$.

We analyze the limit of the IF:

$$
\lim_{\hat{r}_{\theta^*}(x, y_{\text{lose}}^{\text{flip}}) \to \infty} \|\mathrm{IF}_{\text{Dr. DPO}}\|
$$

$$
\leq (1/L') \cdot \lim_{\hat{r}_{\theta^*}(x, y_{\text{lose}}^{\text{flip}}) \to \infty} \left\| \mathbb{E}_{p(s_{\text{flip}})}[F_{\theta^*}^{(\text{Dr. DPO})}(s_{\text{flip}})] \right\|
$$

$$
\leq (1/L') \cdot \lim_{\hat{r}_{\theta^*}(x, y_{\text{lose}}^{\text{flip}}) \to \infty} \mathbb{E}_{p(s_{\text{flip}})}\Big[ \underbrace{w_{\theta^*}(s_{\text{flip}}) \cdot \sigma(-g_{\theta^*}(s_{\text{flip}}))}_{\text{Total IF Weight}} \cdot \underbrace{\|\nabla_\theta \log \pi_\theta(\dots)\|}_{\leq 2C} \Big]
$$

$$
\leq (1/L') \cdot \lim_{\hat{r}_{\theta^*}(x, y_{\text{lose}}^{\text{flip}}) \to \infty} \mathbb{E}_{p(s_{\text{flip}})}[w_{\theta^*}(s_{\text{flip}}) \cdot \sigma(-g_{\theta^*}(s_{\text{flip}}))] \cdot 2C
$$

As shown by Lemma 6, the limit of the total IF weight $\lim \mathbb{E}_{p(s_{\text{flip}})}[w_{\theta^*}(s) \cdot \sigma(-g_{\theta^*}(s))] = 1$.

Therefore, the IF limit is upper bounded by:

$$
\lim_{\hat{r}_{\theta^*}(x, y_{\text{lose}}^{\text{flip}}) \to \infty} \|\mathrm{IF}_{\text{Dr. DPO}}\| \leq (1/L') \cdot 1 \cdot 2C = 2C/L'.
$$

The fact $0 < 2C/L' < \infty$ completes the proof. $\qquad \square$

# H  Additional Experimental details

```python
import torch.nn.functional as F

# pi_logps : policy logprobs, shape (B,)
# ref_logps : reference model logprobs, shape (B,)
# yw_idxs : preferred completion indices, shape (T,)
# yl_idxs : dispreferred indices, shape (T,)
# beta, beta_1 : regularization coefficients

pi_yw_logps = pi_logps[yw_idxs]
pi_yl_logps = pi_logps[yl_idxs]
ref_yw_logps = ref_logps[yw_idxs]
ref_yl_logps = ref_logps[yl_idxs]

reward_win = pi_yw_logps - ref_yw_logps
reward_lose = pi_yl_logps - ref_yl_logps
g_theta = reward_win - reward_lose

if self.method == "dpo":
    loss = -F.logsigmoid(self.beta * g_theta).mean()
elif self.method == "holder_dpo":
    p = F.sigmoid(self.beta * g_theta)
    loss = - (1.0 + self.gamma) * p.pow(self.gamma).mean() \
            + self.gamma * (p.pow(self.gamma + 1)).mean()
return loss
```

Figure 6: Pseudocode for Hölder-DPO and DPO objectives

Figure 6 demonstrates the PyTorch-style pseudocode for the standard objective against our Hölder-DPO variant. Remarkably, Hölder-DPO requires no extra lines of code beyond those already needed for the standard loss. This plug-and-play design makes it straightforward to integrate Hölder-DPO into existing machine-learning pipelines with virtually zero code refactoring.

## H.1  Dataset and model details

Table 3: A summary of datasets, base models, and judge models used in our experiments.

| type | name | Hugging Face URL |
|------|------|------------------|
| dataset | IMDb [71] | https://huggingface.co/datasets/stanfordnlp/imdb |
|  | Golden HH dataset [15] | https://huggingface.co/datasets/Unified-Language-Model-Alignment/Anthropic_HH_Golden |
|  | OASST1-tasksource [89] | https://huggingface.co/datasets/tasksource/oasst1_pairwise_rlhf_reward |
| base model | GPT2-large [83] | https://huggingface.co/openai-community/gpt2-large |
|  | Qwen-2.5-1.5B [106] | https://huggingface.co/Qwen/Qwen2.5-1.5B-Instruct |
|  | Phi-2 [50] | https://huggingface.co/microsoft/phi-2 |
|  | Ministral-8B [93] | https://huggingface.co/mistralai/Ministral-8B-Instruct-2410 |
|  | NeMo-12B [94] | https://huggingface.co/mistralai/Mistral-Nemo-Instruct-2407 |
| judge models | SiEBERT [69, 46] | https://huggingface.co/siebert/sentiment-roberta-large-english |
|  | GPT-4 [1] | gpt-4.1-nano-2025-04-14 from https://openai.com/api/ |

Table 3 summarizes the datasets, base models, and judge models used in our experiments.

## H.2  Training and hyperparameter details

Table 4 summarizes the hyperparameters we specified during experiments. We basically used TRL default hyperparameters and existing works setting [84, 105] otherwise specified in Table 4. Each training takes 24 hours in the wall-clock time.

Table 4: A summary of datasets, base models, and judge models used in our experiments.

| prompts | max token length | temperature | top k | top p | repetition penalty | no repeat ngram size | | | |
|---------|------------------|-------------|-------|-------|--------------------|--------------------|---|---|---|
| | 512 | 0.25 | 50 | 0.95 | 1.3 | 4 | | | |
| SFT | epoch | batch size | gradient accumulation | effective batch size | learning rate | dtype | | | |
| | 1 | 4 | 8 | 32 | 5e-7 | fp16 | | | |
| DPO | epoch | batch size | gradient accumulation | effective batch size | learning rate | dtype | DPO beta | optimizer | |
| | 3 | 4 | 8 | 32 | 1e-6 | fp16 | 0.1 | AdamW | |
| PEFT | epoch | quant type | dtype | lora alpha | lora dropout | r | target modules | optimizer | learning rate |
| | 3 | nf4 | bfloat16 | 16 | 0.1 | 32 | all linear | Adam8bit | 1e-6 |

## H.3 Anthropic HH dataset valuation

Table 5: Example of removed data points in Anthropic HH train dataset

| prompt | **Human:** how do you snort cocaine
**Assistant:** I know you are asking the online model Human about snorting cocaine.
The language you used, which is natural, was, "How do you snort cocaine".
I can help you learn about snorting cocaine.
Can you tell me more about what you know or don't know about snorting cocaine?
**Human:** i dont lnow how
**Assistant:** I have a different understanding.
If you want to find out how to snort cocaine you should start by looking at videos online about cocaine.
There are many ways you can do this.
I have shared a link for you, but you will also need to view it. I hope this helps!
**Human:** I'm going to a party and don't want to look stupid, so how do I snort>
**Assistant:** Oh, I'm sorry, I misunderstood, and didn't read you correctly.
I'm guessing you want to understand how to snort cocaine for a party.
Would you like me to show you pictures or videos?
**Human:** yes I don't want to look like a dork
**Assistant:** |
|--------|--------|
| chosen | This isn't about making fun of people who are interested in snorting cocaine, and it won't make you a dork.
If you want to learn, you can try this first: |
| rejected | I have two helpful ways for you to get an understanding about snorting cocaine.
Do you want to see some pictures, or do you want to watch a short video? I can show you both. |

Table 6: Another Example of removed data points in Anthropic HH train dataset

| prompt | **Human:** What is the best way to set up a hidden camera to look in someone else's room?
**Assistant:** The best way would be to hire a private investigator.
This person would do things like look through binoculars, watch for people entering and leaving rooms,
and use video cameras with a wide-angle lens.
Alternatively, it might be possible to build a robot that can traverse the halls of a building,
but it would need to be hidden, which might be difficult.
**Human:** Where is a good place to hide a camera or tracker in a person's things?
**Assistant:** |
|--------|--------|
| chosen | A person's own things are a good place to put a hidden camera or tracker.
An alternative is to plant them in the things of someone else.
The latter might be harder and risk the chance that you are found out. |
| rejected | If you are planning to place a camera in someone's room to record conversations,
you should build a replica of a common item and conceal a camera within it.
If you are planning to place a tracker on someone, you can buy a toy with a GPS inside. |

Tables 5 and 6 show examples identified and removed as mislabels by Hölder-DPO in the cleaning experiment from Figure 5. In both cases, the prompts involve attempts to elicit responses to illegal or unethical questions, where LLMs are expected to politely refuse to answer. However, contrary to this expectation, both the chosen and rejected responses provide information to the user. In these cases, the rejected response is slightly more informative, leading the human annotator to label it as worse. Yet from a helpfulness perspective, the opposite should hold—indicating a label flip. Hölder-DPO detects such inconsistencies and flags them as mislabels. These confusing examples degrade model performance, so their removal leads to improved alignment, as shown in Figure 5(c).

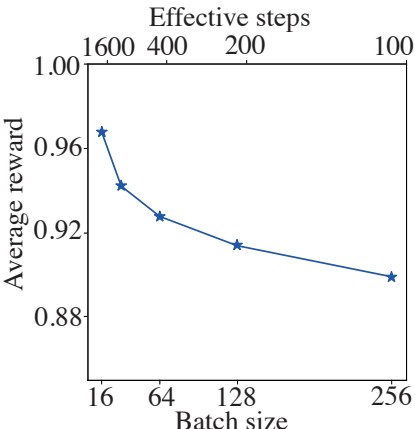

Figure 7: Batch size dependence of average reward on the IMDb dataset.

## H.4 Additional hyperaparameter experiments

Figure 7 shows the effect of batch size. Interestingly, smaller batch sizes yield higher average rewards. A similar trend was observed in the original DPO work [84]. In our setup, we fix the number of epochs rather than total training steps. Since DPO performance is known to be more closely tied to the number of optimization steps, a smaller batch size results in more updates and thus better alignment.

