# OpenReview forum: "Scalable Valuation of Human Feedback through Provably Robust Model Alignment"
_NeurIPS.cc/2025/Conference — NeurIPS 2025 poster_

### Official Review · Reviewer_buZZ · 2025-07-01

**Clarity:** 3
**Significance:** 2
**Originality:** 3
**Rating:** 4
**Confidence:** 4

**Summary:**

This paper focuses on the issue of noise in crowd-sourced human feedback, particularly in the context of aligning LLMs with human preferences. To address this challenge, the authors introduce Hölder-DPO, a novel variant of DPO that incorporates the Hölder divergence metric. This metric exhibits desirable robustness properties, enabling the model to better handle noisy feedback by assigning lower likelihoods to mislabeled data points compared to clean ones.

**Questions:**

**Role of Hölder-DPO**: From the authors' perspective, what role is Hölder-DPO best suited for—final training or preprocessing? The method's ability to detect and filter mislabeled data suggests it could serve as an effective preprocessing tool, potentially improving the performance of other alignment methods when applied to clean datasets.

**Ethical Concerns:**

["NO or VERY MINOR ethics concerns only"]

**Limitations:**

1. **Complexity of Real-World Noise**: While the symmetric label flip is a standard way to test robustness, real-world noise is often more complex and may not be captured by such a simple model. Future work could explore more nuanced noise types, such as "easy to agree, hard to differentiate" noise, where annotators may agree on the general direction but struggle with subtle distinctions.
2. **Scalability and Filtering Strategy**: The experiments are conducted on relatively large datasets, with a large space for filtering strategy that removes noisy examples. However, in domain-specific applications (e.g., medicine, law), where high-quality preference data is scarce, a more conservative filtering strategy may be needed. Uncertainty-aware filtering, where the model is less aggressive in filtering low-confidence examples, could improve the method's applicability in such settings.

**Paper Formatting Concerns:**

The paper is well-structured and clearly presents the problem, methodology, and experimental results.

**Quality:**

2

**Strengths And Weaknesses:**

### Strengths

1. **Motivation & Theoretical Robustness:** The paper provides formal evidence that existing DPO variants, such as R-DPO and Dr. DPO, do not satisfy the redescending property, therefore introducing the Hölder divergency with redescending property proof, which ensures robustness under severe label noise.
2. **Improved Alignment Performance**: Empirical results demonstrate that Hölder-DPO consistently outperforms baseline methods in average reward across varying levels of contamination, generation temperatures, and training steps. Additionally, it accurately detects mislabeled examples, significantly improving alignment performance when such examples are removed.

### Weakness
1. **Strong Assumptions on Noise**: The method's effectiveness is based on the assumption of strict tail data contamination, i.e., mislabeled data points can be clearly distinguished from clean ones by being assigned very low likelihood. In real-world scenarios, this may not always hold, especially when the differences between clean and noisy data are subtle or systematic.
2. **Sensitivity to Hyperparameter $\gamma$**: Hölder-DPO introduces an additional hyperparameter, $\gamma$, which controls the level of contamination removal. While the paper reports robust performance with a default value of 2.0, the optimal choice may vary across tasks and datasets. This sensitivity may require tuning to fully realize the method's potential.
3. **Limited Noise Type Analysis**: The paper's primary model for noise is the symmetric label flip, where a $(y_{\text{win​}},y_{\text{lose}}​)$ pair is randomly swapped with a certain probability $\varepsilon$. While this is a standard and necessary approach for testing robustness, real-world noise is often more complex and systematic, stemming from annotator biases or task ambiguities.

---

> ### Author Rebuttal · Authors · 2025-07-30
>
> We thank you for your thoughtful and constructive review. We are encouraged by your positive assessment and appreciate the insightful questions regarding the practical assumptions and deployment of our method. Below are our detailed responses.
>
> # Q1: Weakness 1 & 3: Strong assumptions on noise (strict tail contamination) and limited noise type analysis (symmetric label flips).
>
> ## A1:  Our assumption is weaker and more practical than classical approaches, and it handles asymmetric noise.
>
> ### **1\. On the Tail Contamination Assumption vs. Infinitesimal Contamination**
> You are correct that our theory relies on this assumption (tail contamination). However, we sincerely argue it is a reasonable and practical abstraction of how a model learns from noisy data, under the setting where we have no access to clean validation dataset. A model trained on a majority of clean preferences should learn to assign very low likelihood to gross errors like flipped labels. Our empirical results across multiple datasets (Figs. 3, 4, 5) provide compelling evidence that this assumption holds sufficiently well in practice to deliver state-of-the-art robust performance, even if real-world data does not perfectly match this theoretical ideal.
>
> Furthermore, this assumption is more realistic for large-scale datasets than the other assumptions often used in classical robust statistic, such as infinitesimal contamination ($\epsilon \approx 0$). In the LLM alignment context, **where datasets can have substantial noise (e.g., >25% in Anthropic HH)**, an infinitesimal contamination model is simply not applicable. Our tail contamination assumption, which allows for heavy contamination, is a more practical abstraction. Furthermore, as we note in our paper (lines 95-96), this type of assumption—that outliers can be identified in the tail of a distribution—is a well-established principle implicitly or explicitly used in many robust learning and outlier detection algorithms.
>
> ### **2\. On the Symmetric Label Flip Model and the True Limitation**
> We would like to clarify that our method's core theoretical assumption is **not strictly limited to symmetric flips, but rather the more general condition of tail contamination**. This means our method is effective against any type of noise—be it symmetric, asymmetric, or even some systematic annotator biases—as long as the noisy preference pair is correctly identified by the learned model as a low-likelihood event (an outlier).
>
> The true limitation lies with noise that is too systematic to be identified as an outlier by the model. It is plausible to hypothesize that the method's performance would degrade when noise is structured in a specific way—for example, when all preference labels for a single, specific prompt or a narrow topic are consistently flipped.
>
> The reasoning is that these structured errors would no longer form a clear "tail" in the overall data distribution. Instead, they would create a consistent but incorrect signal for that specific sub-task, which the model would likely struggle to distinguish from a genuine, albeit niche, preference. Exploring such complex, systematic noise structures is a vital direction for future research.
>
>
> # Q2: Weakness 2: Sensitivity to Hyperparameter $\gamma$
>
> ## A2: It is shared challenge in robust statistics community, including prior work, not specific to our work.
>
> The choice of $\gamma$ is indeed a key practical consideration, as is common for many robust statistical methods. We would like to emphasize that this kind of challenge is shared by the other robust methods like R-DPO and Dr. DPO. These methods also depend on a critical hyperparameter (the estimated noise ratio $c$), which is often unknown a priori and requires tuning.
>
> Our key finding is that a single default value ($\gamma=2.0$) works consistently well across our diverse experiments. This suggests that while $\gamma$ is an important dial for tuning the robustness-efficiency tradeoff, our method is not prohibitively sensitive and can be effectively used by practitioners without exhaustive hyperparameter sweeps.
>
> # Q3: Role of Hölder-DPO.
>
> ## A3: Preprocessing if expensive two-step training is permissible given budge, otherwise final training.
>
> This is an excellent and insightful question that touches upon the practical deployment of our method. We believe Hölder-DPO is a versatile tool, and its optimal role depends on the scale of the dataset and the specific application scenario.
> - **For Medium-Scale or High-Stakes Datasets:** In this scenario, a two-stage approach is likely optimal.
>
>     - Stage 1: Train a model with Hölder-DPO to leverage its robustness and data valuation capabilities.
>     - Stage 2: Use the trained model to identify and filter out the noisy data points.
>     - Stage 3: Train a final model (using standard DPO or another method) on the newly cleaned, high-quality dataset. This approach, validated by our Figure 5c, maximizes final performance by combining robustness for cleaning and efficiency for final alignment.
>
> - **For Very Large-Scale Datasets:** When dealing with massive datasets where a two-stage process might be computationally prohibitive, a one-stage approach is more practical.
>     - Train the final model directly with Hölder-DPO. This provides a robustly aligned model in a single, efficient training run.
>     - The data valuation scores generated during this process can then be used to create a cleaned dataset that is held in reserve for the next training cycle, continuously improving data quality over time.
>
> In summary, Hölder-DPO can be used as both a direct training objective and a preprocessing tool, and practitioners can choose the best strategy based on their specific constraints and goals.
> We will add a new appendix section to explicitly discuss these practical, data-dependent deployment scenarios, highlighting the dual-role capability of our work.
>
>
> # Q4: Limitation 1: Complexity of Real-World Noise.
>
> ## A4. Fully agreed. We added the explanation in the Limitation section.
>
> To summarize our position: we clarified that our method's core assumption of "tail contamination" is more general than just symmetric flips and can handle various noise types. The true limitation, as you astutely point out, lies with complex, systematic noise that does not form a clear "tail" in the data distribution—such as the excellent example you provided of "easy to agree, hard to differentiate" noise.
> Investigating such complex noise types is indeed an important and significant direction for our future work.
>
> We will ensure this nuanced perspective is reflected in the final paper's Limitations section.
>
> # Q5: Limitation 2: Scalability and Filtering Strategy in data-scarce domains.
>
> ## A5: We understand the sentiment, but in data-scarce settings, flipped data points matter more—so our method performs better.
>
> We argue that our method is particularly valuable in such scenarios, guided by the principle of "less is more for alignment."
>
> As our own Figure 5c demonstrates, removing noisy data points leads to better alignment quality, even if it reduces the dataset size. We believe that carefully cleaning a small dataset is often more effective than training on a larger, noisy one.
>
> Crucially, our method offers the flexibility needed for these high-stakes domains and is not just an aggressive, automated filter. Instead, it empowers domain experts by providing a ranked list of all data points from most to least likely to be noisy. An expert in a data-scarce setting does not need to blindly trust the model's estimated noise ratio $\hat{\epsilon}$. Instead, they can use our method's ranking to make a more informed, conservative decision:
> - **For efficient manual review:** The expert can start inspecting data from the top of the ranked list (the most suspicious items). This allows them to focus their valuable time on the most problematic $\hat{\epsilon}$ fraction of the data, rather than having to review the entire scarce dataset.
> - **For conservative filtering:** The expert can choose a much smaller, more conservative threshold for removal (e.g., only removing the top 1-5% of most suspicious items) to preserve as much data as possible while still eliminating the most harmful outliers.
>
> Therefore, rather than being a risky tool in data-scarce settings, Hölder-DPO serves as an intelligent assistant that makes the process of cleaning valuable datasets more efficient and controllable.
>
> We will add this discussion to our Limitations section (Section 6), highlighting how the ranked output of our method enables flexible, human-in-the-loop filtering strategies suitable for data-scarce domains.
>
> Thank you again for the helpful suggestions and being constructive. We have incorporated them into our revision and will continue to strengthen the experimental section in the final version. We hope our rebuttal clarifies our points.
>
> Many thanks,
>
> All the authors

---

> > ### Author Response · Authors · 2025-08-07
> > **Any further questions or suggestions?**
> >
> > Dear reviewer buZZ,
> >
> > Your review and comments have helped us improve the quality of the manuscript a lot. And we hope our rebuttal and further responses have addressed your concerns adequately. Otherwise, please let us know if you have any further questions or suggestions. We are more than happy to provide further responses.
> >
> > Best,
> >
> > All the authors.

---

### Official Review · Reviewer_kngT · 2025-07-02

**Clarity:** 1
**Significance:** 2
**Originality:** 2
**Rating:** 4
**Confidence:** 3

**Summary:**

The work addresses an important issue in LLM alignment where preference data is noisy. The authors proposed to modify the DPO loss so that it is robust to the presence of noisy labels under certain assumptions. Their experiment results show the effectiveness of the proposed method.

**Questions:**

Please refer to the Strengths and Weaknesses section.

**Ethical Concerns:**

["NO or VERY MINOR ethics concerns only"]

**Final Justification:**

My main issue is the technical clarity. With the response, I do have a better understanding compared to my initial reading. However, there are still confusing parts. Even the authors does not have a good distinction on their notations, such as $r_{\theta}$ vs $\widehat{r}_{\theta}$. The lack of clarity hinders any attempts in justify the soundness of their theoretical results.

However, the empirical experiments show clear and consistent improvement over baselines. This might suggest certain validity of the proposal. Considering all factors, I would like to raise my score from 3 to 4.

**Limitations:**

Yes

**Quality:**

2

**Strengths And Weaknesses:**

__Strengths__: Overall, the manuscript is well structured. The experiment results show clear improvement over baseline methods. The development of the proposed method is convincing intuitively. The experiment results show evident improvement.

__Weaknesses__:
However, I have several serious concerns about technical clarity.

- The definition of IF function is syntactically problematic: $x, \theta$ and $p_{\mathcal{D}}$ appear as input but all three factors does not appear in the right hand side. What does the notation $\| \cdot \|$ mean? As $\theta(\epsilon)$ is a vector-value function,  $\| \cdot \|$ couldn't be the absolute function since IF is a scalar-valued function.
- The proposed robustness framework is mostly motivated from the condition in Definition 2. However, I find it difficult to understand it at both intuition level and technical level, as the discussion was too brief:
    - $r_{\theta}$, and  $\theta$ are never defined
    - Can you clarify the comment in line 107, i.e., how does $r_{\theta}(x, y_{\text{lose}}^{\text{flip}}) \to  \infty$ correspond to the worst-case scenario for DPO?
    - What is the physical interpretation of the limit $r_{\theta}(x, y_{\text{lose}}^{\text{flip}}) \to  \infty$? It certainly does NOT necessarily imply that $y_{\text{lose}}^{\text{flip}} \succeq  y_{\text{win}}^{\text{flip}}$ since it is possible that $r_{\theta}(x, y_{\text{win}}^{\text{flip}}) \to \infty$ as well.
    - Most importantly, the condition in definition 2 does not involve any property of noise, how does that condition relate to noise robustness as stated in line 110, i.e., $\theta^{\star}(\epsilon) = \theta^{\star}$? This relation if holds is not trivial and should be addressed more carefully.
+ In Figure 2, what is IF weight? How is IF weight evaluated? How does Figure 2 relate to the condition in Definition 2, particularly the x-axis is not exactly the quantity under limitation?
+ If a loss function is redescending, does the present of noise cause any effect at all, such as increasing computation cost? In particular, I'm curious about the tradeoff of having a redescending loss function as it seems to guarantee a "too good" result against noise.
+ What is the effect of $\gamma$?

---

> ### Author Rebuttal · Authors · 2025-07-30
>
> Thank you for the valuable feedback. Below are our detailed responses.
>
> # Q1: The definition of the IF function is syntactically problematic and confusing.
>
> ### R1: We appreciate this observation and respond in two parts:
>
> ## Q1.1: The function $\mathrm{IF}(x, \theta, p_D)$'s inputs do not appear on the right-hand side of Eq. (4).
>
> ### R1.1: **This is a shorthand; the full form is provided in Appendix E.1.**
>
> We apologize for the confusing notation. The expression in Eq. (4) follows a common shorthand in robust statistics literature, omitting explicit dependence for readability. However, we agree this may not be accessible to all readers and should have been explained more clearly.
>
> The complete derivation is provided in Appendix E.1, where the influence function is given as:
>
> $$
> \text{IF}(x, \theta, p_D) = -H_1^{-1} F_1,
> $$
>
> where
>
> $H_1 = \mathbb{E}_{p_D}[ H_2 ],$
>
> $H_2 = H_{\theta}(s),$
>
> $F_1 = \mathbb{E}_{p ( \tilde{s} ) }[ F_2 ],$
>
> $F_2 = F_{\theta}( \tilde{s} ),$
>
> $\tilde{s} := s_{\text{flip}}$
>
> Here, the Hessian term $H_1$ clearly depends on the clean distribution $p_D$, while the outlier term $F_1$ depends on the specific outlier point (derived from $x$). Both terms are evaluated at $\theta = \theta^*$, which itself is a function of $p_D$.
>
> We have incorporated the above explanation in Section 2.
>
> ## Q1.2: What does the notation $|\cdot|$ mean? As $\theta(\epsilon)$ is a vector, it can't be the absolute value.
>
> ### R1.2: **Apologies. This was a typographical error.**
>
> You are absolutely right. The IF expression $\left.\frac{\partial \theta^*(\epsilon)}{\partial \epsilon}\right|_{\epsilon = 0}$ is a vector, and the $|\cdot|$ symbol is inappropriate. We have corrected this in our revised draft.
>
> # Q2: The robustness framework and the condition in Definition 2 are difficult to understand.
>
> ### R2: **We have now added the following clarification.**
>
> We acknowledge that our original explanation of Definition 2 was brief, primarily due to space constraints. In the camera-ready version—where we will have an additional page—we have expanded this section to provide a clearer and more detailed discussion, as outlined below:
>
> ## Q2.1: The terms $r_{\theta}$ and $\theta$ are never defined.
>
> ### R2.1: **Both are defined in L65 and L72.**
>
> $\theta$ denotes the parameters of the policy (LLM) model $\pi_{\theta}$, and $r_{\theta}$ represents the implicit reward function. We acknowledge a source of confusion stemming from our inconsistent use of $\hat{r}{\theta}$ and $r{\theta}$—these refer to the same quantity. We have now unified the notation throughout the paper to consistently use $r_{\theta}$.
>
> ## Q2.2: How does $r_{\theta}(x, y^{\text{flip}}_{\text{lose}}) \to \infty$ correspond to the worst-case scenario?
>
> ### R2.2: **It represents maximal misalignment to a flipped label.**
>
> In model alignment, the worst case occurs when the LLM is misled to strongly update toward a flipped label, effectively learning the wrong preference. The importance of a data point is implicitly governed by the reward function $r_\theta$, which we cannot observe directly. Since humans are generally more reliable at relative comparisons than assigning absolute scores (e.g., ranking a prompt among 10,000 candidates), we estimate $r_\theta$ via pairwise preferences, expressed as:
>
> $
> g\_{\theta}(s) = r\_{\theta}(x,y\_{\text{win}}) - r\_{\theta}(x,y_{\text{lose}}).
> $
>
> The model is aligned by updating its parameters $\theta$ based on the loss gradient $\nabla_\theta \mathcal{L}$ (Eq. 3), which is scaled by $g_\theta(s)$. As $g_\theta(s) \to \infty$, the update becomes overly dominated by this data point.
>
> Now, consider the worst-case: if the flipped label $y_{\text{lose}}^{\text{flip}}$ is assigned an infinitely high reward, i.e., $r_\theta(x, y_{\text{lose}}^{\text{flip}}) \to \infty$, then $g_\theta(s) \to -\infty$, and the model is misaligned with maximum confidence. This scenario captures the core robustness failure.
>
> While we focus on $r_\theta(x, y_{\text{lose}}^{\text{flip}})$ in the limit, the interpretation is equivalent to the worst-case $g_\theta(s)$ due to symmetry. This is why Definition 2 formalizes robustness as the requirement that the model nullifies updates under such extreme (and erroneous) conditions.
>
> ## Q2.3: How does the condition in Definition 2 relate to noise robustness? This relation is not trivial.
>
> ### R2.3: **Satisfying Definition 2 ensures that the LLM’s parameter update is nullified in response to a flipped label—providing robustness to noise.**
>
> This is formalized in Appendix C, Eq. (14), which shows that the influence of a flipped point on the optimal parameters $\theta^*(\epsilon)$ vanishes to first order. As noted in Footnote 1 (Page 3), this relies on a first-order Taylor expansion, and thus does not eliminate higher-order effects.
>
> Nevertheless, by nullifying the first-order influence, we ensure that $\theta^(\epsilon) \approx \theta^*$ (see Line 110), capturing a meaningful notion of robustness to label noise.
>
> # Q3: In Figure 2, what is "IF weight"? How does the figure relate to Definition 2, particularly the x-axis?
>
> ## R3: **Explanation added**
>
> We agree that this part was particularly unfriendly. We have added the explanation as follows:
>
> ## Q.3.1: What is "IF Weight" and why is it critical for robustness?
>
> ### R3.1: **It is the scalar multiplier that scales the gradient direction.**
>
> Eq.~(3) can be decomposed as:
>
> $$
> \nabla_\theta \mathcal{L}(s, \pi_\theta) := \sigma(- g_{\theta}(s)) g_\text{grad}(s),
> $$
>
> where the multiplicative factor $\sigma(- g_{\theta}(s))$ represents the **IF weight**, and $g_\text{grad}(s)$ represents the direction of the gradient.
> Here, IF weight determines how strongly the model updates its parameters for each data point. If IF weight tends to zero in the worst-case (i.e., $g_\theta(s_{\text{flip}}) \to -\infty$), the method is robust because it ignores high-confidence misaligned (flipped) inputs.
>
> We summarize the worst-case IF weight behavior for various methods in Table 1:
>
> Table 1: IF weights for all alignment loss.
> | Method | IF Weight | Worst-case $g_{\theta}(s_{\text{flip}}) \to -\infty$|
> |---|---|---|
> | DPO [83] | $\sigma(-g_{\theta}(s_{\text{flip}}))$ | $\to 1$ (non-zero)|
> | cDPO [74] | $(1-c)\sigma(-g_{\theta}(s_{\text{flip}})) + c\sigma(g_{\theta}(s_{\text{flip}}))$ | $\to 1-c$ (non-zero)|
> | R-DPO [24] | $\frac{1-c}{1-2c}\sigma(-g_{\theta}(s_{\text{flip}})) + \frac{c}{1-2c}\sigma(g_{\theta}(s_{\text{flip}}))$ | $\to \frac{1-c}{1-2c}$ (non-zero)|
> | Dr. DPO [104] | $w_{\theta^{*}}(s_{\text{flip}})\sigma(-g_{\theta}(s_{\text{flip}}))$ | $\to \text{const.}$ (non-zero)|
> | IPO [7] | $\left(\frac{g_{\theta}(s_{\text{flip}})}{\beta} - \frac{1}{2\beta} \right)$ | $\to -\infty$ (non-zero)|
> | Hölder-DPO (Ours) | $\sigma(g_{\theta}(s_{\text{flip}}))^{\gamma}\sigma(-g_{\theta}(s_{\text{flip}}))$ | $\to 0$ (zero)|
>
> Figure 2 plots this IF weight (y-axis) versus $g_\theta(s_{\text{flip}})$ (x-axis). The limit $g_\theta(s_{\text{flip}}) \to -\infty$ corresponds to Definition 2. Only Hölder-DPO satisfies Definition 2.
>
> ## Q.3.2: How does Figure 2 relate to Definition 2 and the x-axis?
>
> ### R.3.2: **The y-axis approaching zero as x-axis to minus infinity corresponds directly to Definition 2.**
>
> While the symbolic proof in Table 1 is sufficient to establish the redescending property formally, Figure 2 offers an intuitive, visual interpretation of how each method behaves under extreme cases.
>
> The IF weight on the y-axis reflects how sensitively the model updates its parameters in response to each data point. A high IF weight indicates strong influence on the parameter update. For example, vanilla DPO assigns high weight when the base model makes a mistake (i.e., $g_{\theta^*}(s) < 0$, meaning the base model mispredicts human preference as $y_{\text{lose}}$ over $y_{\text{win}}$), which is reasonable for standard alignment.
>
> In contrast, Hölder-DPO emphasizes updates only when the model’s confidence is moderate, i.e., when $| g_{\theta^*}(s) |$ is small. When the human feedback strongly contradicts the model's prior belief (large negative $g_{\theta} (s) $), Hölder-DPO assumes that the feedback itself may be unreliable, and therefore suppresses the update. This reflects a robust alignment strategy that guards against extreme but potentially erroneous feedback.
>
> # Q4: On the tradeoffs of the redescending property.
>
> ## R4: No additional computational cost; sensitivity to $\gamma$ is the primary trade-off.
>
> Hölder-DPO (Eq. 12) incurs no extra computational cost over vanilla DPO. The main trade-off lies in tuning the hyperparameter $\gamma$, which controls the aggressiveness of downweighting potential outliers.
>
> This is a standard feature of robust estimators, as corroborated by Reviewer FZwu. Similar to the role of hyperparameters in R-DPO or Dr. DPO, $\gamma$ must be chosen appropriately. However, we show that a single default setting ($\gamma = 2.0$) performs robustly across a wide range of settings, demonstrating practical usability.
>
> # Q5: What is the effect of $\gamma$?
>
> ## R5: $\gamma$ serves as **a sensitivity knob** for outlier detection:
>
> - **A small $\gamma$** makes the model more tolerant: data points are downweighted only when the model is highly confident they are outliers (i.e., $s$ is strongly negative). The redescending slope is gentle.
> - **A larger $\gamma$** increases sensitivity: even moderately negative $s$ values lead to aggressive downweighting. The redescending slope is steep.
>
> Thank you again for the helpful suggestions and being constructive. We have incorporated them into our revision and will continue to strengthen the experimental section in the final version. We hope our rebuttal clarifies our points.
>
> Many thanks,
>
> All the authors

---

> > ### Comment · Reviewer_kngT · 2025-08-04
> >
> > Hi authors,
> >
> > Thank you for addressing my comments in details. I have some follow-up comments/questions regarding your responses.
> >
> > 1. Which equation in the Appendix E1 is the definition of the IF function? I still can't find it.
> > In addition, the requirement in Theorem 5 (and possible the same requirement in Theorem 2) on positive definite of the Hessian matrix is highly unlikely to hold because the loss function is non-convex with respect to $\theta$.
> >
> > 2. The requirement in definition 2, that the optimal solution $\theta^*$ as a function of $\epsilon$ have a vanishing gradient at $\epsilon=0$ seems unnatural, served as a technical bypass rather than a physically interpretable assumption, and hence very stringent. It depends simultaneously on multiple factors, the data distribution, the loss function, and the neural network that $\theta$ is used to parameterized.
> >
> > 3. The term $r_{\theta}$ is neither defined in L65 or L72.

---

> ### Author Response · Authors · 2025-08-05
> **Reply for Reviewer kngT**
>
> Thank you for your detailed follow-up and for engaging with our discussion.
>
> >Which equation in the Appendix E1 is the definition of the IF function? I still can't find it.
> ## R.1-1: Lines 1147-1148 in Appendix E.1
>
> Our previous response (R.1.1) corresponds exactly to the equation between lines 1147–1148 in Appendix E.1. Due to a markdown formatting issue on OpenReview, the equation was split across lines, but it remains the same.
> To clarify, this equation presents the full derivation of the IF. The general definition of the IF is given in Eq. (4), while Appendix E.1 provides a specific derivation tailored to the DPO loss.
>
> > In addition, the requirement in Theorem 5 (and possible the same requirement in Theorem 2) on positive definite of the Hessian matrix is highly unlikely to hold because the loss function is non-convex with respect to $\theta$
> ## R.1-2: This is local, not global loss landscape.
>
> This is good point—indeed, the global loss landscape is non-convex. However, our assumption of a positive definite Hessian applies only locally, at the point of convergence ($\theta^{\*}$).
> We do **not** assume global positive definiteness. This local assumption is standard practice, following seminal work such as [Koh & Liang, 2017], and has been adopted in recent studies applying influence functions to large language models (e.g., [Choe et al., 2024]).
>
> ### Citation
> - [Koh & Liang, 2017]: Understanding Black-box Predictions via Influence Functions. ICML2017. https://arxiv.org/abs/1703.04730.
> - [Choe et al., 2024]: What is Your Data Worth to GPT? LLM-Scale Data Valuation with Influence Functions. https://arxiv.org/abs/2405.13954
>
> > The requirement in definition 2, that the optimal solution $\theta^*$ as a function of $\epsilon$ have a vanishing gradient at $\epsilon=0$ seems unnatural, served as a technical bypass rather than a physically interpretable assumption, and hence very stringent. It depends simultaneously on multiple factors, the data distribution, the loss function, and the neural network that $\theta$ is used to parameterized.
> ## R.2: We respectfully disagree. IF is well-known as an interpretability tool thanks to its natural definition.
>
> Definition 2 (redescending property) characterizes robustness as the worst-case behavior of the IF. We argue that IF is a natural and principled metric for sensitivity to data contamination, and that its worst-case value offers an intuitive robustness criterion.
> To motivate this, consider the question: What is a natural way to define sensitivity to data contamination? We aim to quantify how a data contamination $\epsilon$ affects the optimal solution $\theta^{\*}$. In general sensitivity analysis, taking the gradient is a standard approach. The IF formalizes this idea—it is simply the gradient of the optimal solution $\theta^{\*}$ w.r.t. contamination $\epsilon$. A large norm of the IF indicates high sensitivity, meaning the solution is strongly affected by the noise. Moreover, IF provides pointwise sensitivity, allowing us to identify which specific data points most influence the learned parameters. This intuitive interpretation underlies its wide use in model interpretability, as in [Koh & Liang, 2017].
> Conversely, if the IF for a given data point is zero, the model is locally insensitive to that point—it has no influence and could, in principle, be removed without affecting the solution. Definition 2 leverages this idea: if even the worst-case contaminated point has zero influence (i.e., zero IF), then the loss is robust in the strongest possible sense—it nullifies the effect of the most adversarial perturbation. This leads to a natural and interpretable definition of robustness.
>
> Formally, this derivation is shown in Appendix C.
>
> > The term $r_\theta$ is neither defined in L65 or L72.
> ## R.3: $r_\theta = \hat{r}_\theta$
>
> To be perfectly precise:
> - **Line 65** defines the model parameters $\theta$.
> - **Line 72** defines the implicit reward function and gives it the symbol $r_\theta = \hat{r}_{\theta}$.
>
> As we denoted in our answer for your Q.2.1, we will correct this throughout the manuscript to ensure consistency.

---

> > ### Author Response · Authors · 2025-08-07
> > **Any further questions or suggestions?**
> >
> > Dear reviewer kngT,
> >
> > Your review and comments have helped us improve the quality of the manuscript a lot. And we hope our rebuttal and further responses have addressed your concerns adequately. Otherwise, please let us know if you have any further questions or suggestions. We are more than happy to provide further responses.
> >
> > Best,
> >
> > All the authors.

---

### Official Review · Reviewer_FZwu · 2025-07-02

**Clarity:** 3
**Significance:** 4
**Originality:** 4
**Rating:** 5
**Confidence:** 4

**Summary:**

This paper proposes an alternative to the traditional DPO objective (and some more recent robust variants). In particular, the new Holder-divergence inspired objective possesses the redescending property, meaning that the influence of an extreme outlier is not merely bounded, but tends towards zero. They show that this objective not only improves performance on contaminated preference datasets (where the label has likely been flipped a non-trivial fraction of the time), but also enables users to identify the likely contamination percentage as well as the most plausible outliers. They present some theory demonstrating that their approach is the only one amongst popular DPO objectives to have this redescending property and present a fairly extensive empirical evaluation of their methodology on multiple data sets and at varying scales.

**Questions:**

* Could you elaborate more on the sensitivity of your method to the choice of the hyperparameter? I would suggest creating a version of Figure 2 that illustrates how gamma affects the influence function, but also I would suggest you include a version of Table 1 for each of the applications you consider. I don't agree with your characterization of Table 1 either: on line 259, you say that increasing gamma *improves* contamination ratio estimation but reduces precision. It should be *increases* the estimate (and of course, unsurprisingly, then reduces precision). I would also suspect that this choice matters much more for larger models where the implicit reward model becomes much sharper? This may be a concern when scaling this method beyond the still small-ish models considered here.

* In keeping with the above question, I think the paper would benefit from a discussion of why the redescending property is not always prioritized in statistics. As Figure 5c demonstrates, once the data is cleaned, the performance of typical DPO losses is clearly superior. The trade-off between robustness and efficiency (made explicit with this hyperparameter gamma) deserves exposition, even if this method is a promising approach.

* I would strongly suggest you preview the definition of "redescending" much earlier in the paper - it is the key idea of the manuscript and I think the reader would benefit a heuristic description in line 36/37 rather than simply a reference to the influence function (which they may or may not know about).

**Ethical Concerns:**

["NO or VERY MINOR ethics concerns only"]

**Final Justification:**

I think the concern about choosing $\gamma$ is important, but is certainly sufficient novelty in my mind. Re: another review's comments on this topic, very few of the thousands of papers published at NeurIPS each year introduce a "fundamentally new paradigm." This paper will probably not win the test of time award, but I think the field will benefit from exposure to this approach (which is noticeably different from the plethora of very similar "robust DPO" papers cited in this manuscript). The clarity and presentation of the paper could use some improvement, but I don't think it's disqualifying. I remain confident in the score I chose.

**Limitations:**

Yes

**Quality:**

3

**Strengths And Weaknesses:**

**Quality**
I think this submission is certainly technically sound and the claims it makes about its own proposals and others are precise and appear to be correct. My background is more in statistics than post-training LLMs, so I cannot meaningfully weigh in on the comprehensiveness of the experiments, but I found them compelling and more than sufficient for publication in this venue. There could perhaps be more acknowledgment of the limitations of objectives with this property. In classical statistics, the problem is usually in the choice of the parameter the authors call gamma. There is a limited but insufficient discussion of the sensitivity of this method to that choice.

**Clarity**
I think this paper is generally quite straightforward to read and is well-organized. The figures are excellent in particular. It would be nice to see confidence intervals for win rates. I know the sample sizes are written down, but you could add +/- 0.05, etc. in parentheses next to the values.

**Significance**
I think these results are meaningful and impactful. I was very impressed by the empirical improvements and I think introducing this idea from robust statistics into ML/AI world is quite valuable.

**Originality**
This is clear - I think it's a nice idea and it's very well executed. Well done!

---

> ### Author Rebuttal · Authors · 2025-07-30
>
> Thank you very much for your insightful and highly positive review and the helpful comments. In our revision, we have improved the clarity and readability by incorporating your suggestions. The following are our detailed responses.
>
> # Q1: It would be nice to see confidence intervals for win rates.
>
> ## R1: Confidence intervals have been added.
> The error bars in Figures 3 and 4 already represent the standard deviation across 10 independent runs, providing a clear picture of variability and stability. That said, we agree that confidence intervals can offer a more rigorous means of distinguishing methods. We have now added 95% confidence intervals to Figures 3, 4, and 5, as well as to Tables 1 and 2 (see Appendix H).
>
> # Q2: Could you elaborate more on the sensitivity of your method to the choice of the hyperparameter $\gamma$? The discussion is currently limited.
>
> We address this concern in three parts:
>
> ## Q2.1: Elaboration on the $\gamma$ sensitivity
>
> ### R2.1: **We have expanded the explanation as follows.**
>
> $\gamma$ serves as **a sensitivity knob** for outlier detection:
>
> - **A small $\gamma$** makes the model more tolerant: data points are downweighted only when the model is highly confident they are outliers (i.e., $s$ is strongly negative). The redescending slope is gentle.
> - **A larger $\gamma$** increases sensitivity: even moderately negative $s$ values lead to aggressive downweighting. The redescending slope is steep.
>
> This gives practitioners direct control over the trade-off between robustness to gross errors and sensitivity to data variance.
>
> ## Q2.2: Expanded Ablation Studies & Scaling Concerns
>
> ### A2.2: **Further analysis will be included in the camera-ready version.**
>
> Table 1 currently illustrates the core trade-off between robustness and mislabel precision. We are working to expand this ablation in the appendix, but a full sweep across all models and datasets is infeasible within the rebuttal period due to computational limitations (Google Colab, single A100 GPU), where each one data point in one figure takes a day to compute (see Appendix H.2). Since the optimal $\gamma$ is likely task-specific, we believe our existing results already provide useful guidance for practitioners.
>
> ## Q2.3: Inaccurate description on L259
>
> ### R2.3: **You are correct.**
>
> Thank you for catching our imprecise language. We have revised the text to correctly state that increasing $\gamma$ increases the estimated contamination ratio, rather than "improving" it.
>
> # Q3: The paper would benefit from a discussion of why the redescending property is not always prioritized in statistics (the trade-off between robustness and efficiency).
>
> ## R3: Vanilla DPO is indeed preferred over robust DPO if the dataset is perfectly clean, but this condition is rarely satisfied.
> While vanilla DPO may outperform robust methods on perfectly clean datasets, this assumption rarely holds in practice. Even curated datasets like Golden HH (Figure 5a) exhibit non-negligible noise. In these scenarios, vanilla DPO is fragile, degrading significantly with even modest contamination. In contrast, Hölder-DPO remains stable.
> Interestingly, even in the "clean" setting ($\epsilon = 0$, no artificial noise), Hölder-DPO performs comparably to or better than vanilla DPO (Figures 3 and 4), indicating that the "clean" open dataset is actually slightly noisy (see estimated $\hat{\epsilon}$ are non zero at $\epsilon=0$).
> Moreover, we observe in Figure 5c that vanilla DPO trained on the cleaned dataset (filtered by Hölder-DPO) outperforms robust methods trained on unfiltered data. This highlights an optimal two-step procedure: first use Hölder-DPO for noise identification and filtering, then apply vanilla DPO for final alignment—if resources and dataset size permit this expensive training.
>
> We have integrated this discussion into Section 5 and Section 6 (Limitations).
>
> # Q4: I would strongly suggest you preview the definition of "redescending" much earlier in the paper.
>
> ## R4: Agreed—we have amended the explanation to Line 36:
>
> > The redescending property—the idea that the influence of extreme outliers should diminish to zero.
>
> Thank you once again for your valuable feedback. We believe these revisions significantly improve the paper and hope our responses address your concerns clearly.
>
> Many thanks,
>
> All the authors

---

> > ### Comment · Reviewer_FZwu · 2025-08-05
> >
> > I think this is all reasonably clear, and I appreciate the inclusion of some additional ablations. That being said, the presence of $\gamma$ is a pretty serious problem (and is, to my understanding, the reason why this approach is not so popular). You are correct that the optimal value is task-specific - so how would you suggest that a practitioner discover it? Discussing heuristic approaches to this task is a missing component of the paper.
> >
> > Nevertheless, I don't think that one NeurIPS paper is going to solve robust preference optimization, even if I would suggest that the authors define some heuristic approach for selecting the hyperparameter to encourage uptake of their method. I remain confident in the score that I provided earlier.

---

> > > ### Author Response · Authors · 2025-08-06
> > > **Reply for Reviewer FZwu**
> > >
> > > Dear Reviewer FZwu,
> > >
> > > Thank you for your insightful comment.
> > >
> > > While a fully solution for selecting the optimal $\gamma$ is beyond the scope of this paper, we guess that our framework's unique data valuation capability opens up an interesting possibility for future work: exploring a principled, human-in-the-loop process.
> > >
> > > We hypothesize that a practitioner could use our model's ranked list of outliers as a powerful diagnostic tool. By qualitatively inspecting a small sample of the most suspicious data points, they could get a reliable, semantic signal to adjust $\gamma$ if the model is being too aggressive or too tolerant.
> > > We now think that empirically validating the effectiveness of this heuristic is an important and significant direction for our future research.
> > >
> > > We will add a discussion to our Limitations section that denotes this human-in-the-loop process as an important avenue for future research.
> > >
> > > Sincerely,
> > >
> > > --Authors

---

### Official Review · Reviewer_qBds · 2025-07-05

**Clarity:** 3
**Significance:** 3
**Originality:** 2
**Rating:** 4
**Confidence:** 3

**Summary:**

The paper addresses the challenge of aligning large language models (LLMs) with potentially noisy human feedback, proposing a new method called Hölder-DPO. The key idea is to introduce a provably robust alignment objective that is insensitive to label noise, achieving the redescending property from robust statistics.

**Questions:**

Refer to the weakness.

**Ethical Concerns:**

["NO or VERY MINOR ethics concerns only"]

**Limitations:**

Refer to the weakness.

**Quality:**

3

**Strengths And Weaknesses:**

Strengths：
1.The paper provides a rigorous theoretical analysis of robustness in preference-based model alignment. It formalizes the redescending robustness criterion and proves that existing alignment methods (including prior “Provably Robust DPO” and “Distributionally Robust DPO”) lack this property. Hölder-DPO is introduced as the first alignment loss with provable redescending behavior, meaning it can nullify the influence of extremely noisy labels.
2. A highlight is that the aligned model itself can be used to identify mislabelled data points in the training set in a principled way. By virtue of the redescending loss, Hölder-DPO yields a model that can compute the likelihood of each sample being clean, providing a theoretically grounded metric for dataset valuation.
3. Experimental results show consistent improvements in alignment performance under noisy conditions.
Weaknesses
1. The robustness results hinge on a single noise model—i.i.d. label flips in pairwise preferences—and on small-contamination assumptions typical of influence-function theory. They do not cover structured or adversarial biases, and Hölder-DPO would likely fail once noise nears 50 % or becomes systematic. Consequently, the method feels more like an incremental transfer of classical redescending M-estimators to DPO than a fundamentally new paradigm, so its novelty and generality are limited.
2. Although the experiments are generally thorough, there are a few gaps. The robustness is primarily demonstrated on simulated noise (random label flips injected into datasets). While the authors did apply Hölder-DPO to a real dataset (Anthropic HH anad Reddit TL;DR) for noise estimation and cleaning, the alignment performance on real data with unknown noise was not directly evaluated without synthetic corruption.
3. Could the author consider extend the experiments on the more well-established benchmark like arena hard to verify its effectiveness further?

---

> ### Author Rebuttal · Authors · 2025-07-29
>
> Thank you very much for the positive feedback and the helpful comments. In our revision, we have improved the clarity and readability by incorporating your suggestions. The following are our detailed responses.
>
> # Q1: The method's novelty appears limited, feeling like an incremental transfer of classical M-estimators to DPO, and its theoretical guarantees are confined to i.i.d. label flips.
>
> We address this concern in three parts:
>
> ## Q1.1: On the assumption of small contamination.
> ### R1.1: **We respectfully disagree.**
>
> Our method is explicitly designed to handle **heavy contamination**, as stated in Assumption 1. Unlike classical influence function (IF) theory based on infinitesimal contamination ($\epsilon \approx 0$), we adopt a tail contamination model, allowing substantial noise (e.g., $\epsilon = 0.4$). Our experiments confirm that Hölder-DPO remains robust even under 40% label noise.
>
> ## Q1.2: On the noise model (i.i.d. flips) and failure cases (near 50% or systematic noise).
> ### R1.2: **We acknowledge this limitation but clarify its scope.**
>
> You are correct that our theoretical framework currently focuses on the i.i.d. label flip model, and that identifying mislabeled data becomes fundamentally challenging near a 50% noise ratio. However, this is not a limitation of our method per se, but a theoretical identifiability barrier: when noise and clean data are equally represented, distinguishing the two becomes statistically impossible without a clean reference set.
> Our approach assumes a bimodal distribution where one mode (typically the majority) corresponds to clean data. If the noise ratio is extreme (e.g., 90%), Hölder-DPO can still isolate the minority clean subset (10%) as outliers. In such cases, the roles can be reversed—the identified minority is treated as clean, and the rest discarded. This reversal can be verified with minimal human effort by inspecting a few top-ranked samples flagged by our method, if human-in-the-loop workflows are permissible.
> Thus, Hölder-DPO not only enables principled mislabel detection under reasonable assumptions, but also offers a practical, interpretable, and efficient workflow for data inspection with minimal manual oversight.
>
> Table 1: Noise identification conditions
> | Noise regime | Identifiable? | Condition to be identifiable |
> |---|---|---|
> | 0 - 40% | Yes | Unconditional under Assumption 1 |
> | 40 - 60% | No | Requires access to a clean validation dataset |
> | 60 - 100% | Yes | Via manual inspection of a few samples |
>
> Notably, prior methods cannot identify mislabeled data in arbitrary noise regimes, even with clean labels or human inspection (see Theorem 3). Our i.i.d. flip model is standard [24, 104], and extending to adversarial/systematic noise is a compelling future direction.
>
> ### Q1.3: On the novelty and generality of the method.
> ### R1.3: **We respectfully disagree.**
>
> While the redescending property is indeed a classical concept, theoretical novelty often lies not in the conceptual idea itself, but in the rigor, adaptation, and extension to new domains. If formal IF analysis had already been applied to modern LLM alignment methods like DPO, our contribution might be considered a straightforward reuse. However, to the best of our knowledge, no prior work has done so. Thus, we had to lay the theoretical groundwork. As classical redescending estimators are typically studied in much simpler settings (e.g., linear regression), and adapting them to the DPO presents substantial methodological and analytical challenges. Our contributions go beyond a straightforward transfer of classical ideas. Specifically, we introduce three key innovations:
> - **The First Formal IF Analysis for the DPO Family:** Our work is the first to **define the redescending property tailored to the DPO context**, and provide a formal IF analysis for DPO-style objectives, rigorously proving that no existing methods satisfy the redescending property.
> - **A Novel, Compatible Objective Function:** We designed Hölder-DPO, a new objective that is compatible with the unique log-ratio structure of DPO while **provably satisfying the redescending property**, a non-trivial theoretical adaptation.
> - **Gradient-Free Data Valuation:** Our most significant practical contribution is the novel connection between this robust loss and a principled, gradient-free data valuation method, enabling scalable mislabel detection—a new capability in this domain. While the nature of contamination ratio estimation is inherited via Hölder-divergence, it is non-trivial that this merit successfully works in LLM fine tuning. Consequently, we show that some special weighting is needed to construct the valid ratio estimator (see Proposition 1 and Appendix F.3).
>
> By laying the groundwork for integrating classical robust tools into modern machine learning systems, we enable plug-and-play extensions to other domains such as diffusion models. In this sense, our contribution is not merely an application of old ideas, but a significant theoretical advancement that modernizes and operationalizes them in previously unexplored contexts.
>
> In the revision, we will clarify this positioning more explicitly in the introduction and related work.
> We agree that handling structured or adversarial noise remains an important challenge, and we have added a discussion of this as future work in Section 6.
>
> # Q2: The alignment performance on real data with unknown noise was not directly evaluated without synthetic corruption. The robustness is primarily demonstrated on simulated noise.
>
> ## R2: We have already done this.
> In Section 5.3, we evaluate Hölder-DPO on the original, unmodified Anthropic HH training set, which is widely recognized to contain substantial real-world label noise. As shown in Figure 5, Hölder-DPO achieves a GPT-4 win rate of \~0.62, outperforming Dr. DPO (\~0.59), R-DPO (\~0.58), and vanilla DPO (\~0.57).
> This result provides direct empirical evidence that Hölder-DPO’s robustness extends beyond synthetic corruption, yielding improved alignment performance on a real-world benchmark with naturally occurring noise.
>
> We acknowledge that this important result was not sufficiently emphasized. In the revision, we will revise Section 5.3 to explicitly highlight this as a real-world robustness evaluation, underscoring the practical effectiveness of our approach.
>
> # Q3: Additional experiments on a more well-established benchmark like Arena-Hard
>
> ## R3: Yes, it is indeed nice to have, but our experiments are already large as a theoretical paper.
>
> We agree that evaluating on a more challenging benchmark like Arena-Hard would further strengthen our empirical validation. We attempted this, but encountered practical limitations due to computational constraints—specifically, our setup (Google Colaboratory with a single A100 40GB GPU) could not support the larger models and parallel runs required to complete the benchmark within the tight rebuttal period. We plan to include additional benchmarks in the camera-ready version.
>
> That said, we would like to clarify that our current experimental evaluation is comparatively extensive for a theory-oriented paper. We summarize the scope of our setup below for context:
>
> Table 1: Comparison of experimental validataion setups with accepted papers and ours
>
> | Method | Number of benchmarks | Number of baselines | Number of LLM models | Largest model size |
> | --- | --- | --- | --- | --- |
> | R-DPO (ICML 2024; [20]) | 1 | 4 | 2 | 7B |
> | Dr. DPO (ICLR 2025; [105]) | 2 | 4 | 3 | 13B |
> | Hölder-DPO (ours) | 4 | 5 | 5 | 12B |
>
> While our setup may be smaller than that of purely experimental works, our primary contribution is theoretical—we provide the first formal guarantees for redescending and noise identifiability properties in DPO. Our empirical results serve to verify and complement these theoretical findings, and we believe they are appropriate in scope for a theory-driven submission.
>
> Thank you again for the helpful suggestions. We have incorporated them into our revision and will continue to strengthen the experimental section in the final version. We hope our rebuttal clarifies our points.
>
> Many thanks,
>
> All the authors

---

> > ### Author Response · Authors · 2025-08-07
> > **Any further questions or suggestions?**
> >
> > Dear reviewer qBds
> >
> > Your review and comments have helped us improve the quality of the manuscript a lot. And we hope our rebuttal and further responses have addressed your concerns adequately. Otherwise, please let us know if you have any further questions or suggestions. We are more than happy to provide further responses.
> >
> > Best,
> >
> > All the authors.

---

### Note · Authors · 2025-08-12

We sincerely thank all involved in the review process for the highly constructive discussion. We are pleased to confirm that we have addressed all points raised, resulting in a strengthened manuscript.

To be concise within the character limit, we will highlight the core improvements that address the most critical feedback:

## **1. Theoretical Clarity has been Enhanced (addressing concerns from Rv kngT)**
- We have rigorously clarified the definitions and derivations of the Influence Function (IF), ensuring the distinction between the general definition (Eq. 4) and its specific analytical form for each method is unambiguous.
- We have added context to the standard assumptions used in IF analysis (e.g., the local positive definite Hessian), citing seminal related works to show this is standard practice.
- The "redescending property" has been emphasized  not as an assumption, but as a principled design choice for the loss function, explaining its intuitive mechanism as a "volume knob" for outliers. A comprehensive table comparing the "IF Weights" of all methods is now included to make this core concept clear.

## **2. The Discussion on Novelty and Scope has been Deepened (addressing concerns from Rv qBds, buZZ)**
- We have sharpened our claims regarding novelty, highlighting the specific technical challenges (e.g., compatibility with DPO's log-ratio structure) that make our work a non-trivial adaptation of classical robust statistics.
- We have added a more concrete discussion on the limitations of our noise model. This discussion now clarifies that our core "tail contamination" assumption may not hold for certain structured noise (e.g., topic-specific noise), and frames the extension of our method to such scenarios that do not manifest as outliers in the tail of the distribution as a vital direction for future work.

## **3. Practical Guidance has been Provided (addressing concerns from Rv FZwu, buZZ)**
- A discussion has been added to the appendix that proposes a potential human-in-the-loop process for selecting $\gamma$ as a promising avenue for future research.
- We have explicitly discussed the dual role of Hölder-DPO as both a direct training objective and a preprocessing tool, providing data-dependent deployment scenarios for practitioners.


We hope these revisions, directly guided by the invaluable review process, have elevated the manuscript to meet the high standards of NeurIPS.

Warm regards,

--All the authors

---

### Decision · Program_Chairs · 2025-09-17

**Decision:**

Accept (poster)

**Comment:**

This paper tackles the challenge of noisy human feedback in aligning large language models by introducing Hölder-DPO, a novel alignment loss with a provable redescending property. Unlike existing DPO variants, Hölder-DPO ensures that the influence of extreme mislabeled points vanishes, enabling robust learning even under heavy label noise. The authors provide theoretical guarantees that no existing alignment methods satisfy this property, and show that Hölder-DPO enables principled mislabel detection and dataset valuation. Empirical results demonstrate both improved robustness to synthetic and naturally occurring noise (e.g., Anthropic HH, Reddit TL;DR) and the ability to identify and filter mislabeled data, which in turn boosts downstream alignment performance.

Reviewers agreed on the importance of the problem and recognized the technical novelty of formally adapting the redescending property to DPO. Strengths highlighted include the rigorous theoretical analysis, the novel connection between robust statistics and preference optimization, and empirical results showing consistent gains under contamination. Concerns focused mainly on (i) the reliance on i.i.d. label flip or tail-contamination assumptions, with limited discussion of more structured or adversarial noise (R-qBds, R-buZZ); (ii) sensitivity to the hyperparameter $\gamma$ that governs robustness, with requests for more practical guidance (R-FZwu, R-buZZ); and (iii) notational clarity in influence-function definitions (R-kngT). In rebuttal, the authors clarified theoretical assumptions, emphasized that Hölder-DPO handles heavy contamination more broadly than infinitesimal-noise models, highlighted direct robustness results on real datasets without synthetic corruption, and provided additional discussion on $\gamma$, including human-in-the-loop tuning strategies.

Overall, I recommend acceptance. The paper introduces a technically principled method that strengthens robustness in alignment while offering new capabilities for data valuation and mislabel detection. While open questions remain regarding hyperparameter sensitivity and robustness under structured noise, there remain important directions for follow-up work. Please incorporate the final suggestions of the reviewers in the camera-ready version as promised.